# LINEARLY INDEPENDENT FEATURE EXTRACTION

## ABSTRACT

We argue that domain invariance is fundamentally limiting as an objective for out-of-domain generalization (OOD) and propose a more nuanced alternative: Modeling a full spectrum of dependence on the state of the world. We make this objective tractable by developing a spectral theory, grounded in a novel operator algebra, that is formally equivalent to information-theoretic measures of dependence. The culmination is `Linearly Independent Feature Extraction` (LIFE): An algorithm for learning representations with controllable state-dependence, implemented using a simple eigensolver. Analytical evaluation on known data-generating processes demonstrates that `LIFE` recovers oracle-level features. Empirically, on linear hypothesis `LIFE` outperforms current gold standards and, on some datasets, even surpasses deep invariant models. A broadly applicable dynamic theory of state-dependence emerges.

## 1 INTRODUCTION

This is a work in representation learning (Bengio et al., 2013). The subject study is the adaptation of learned representations to large shifts in the state of the world. The objects of analysis are *stateful input streams*; Sequences of inputs whose distribution depends on a discrete random variable, hereafter *the state*.

The payoff is twofold: (i) a reframing of OOD generalization as the control and decomposition of state-dependence, and (ii) a novel spectral theory reducing this complex adaptation objective to a standard symmetric eigenproblem.

The life cycle of a learning machine follows a binary rhythm of learning and inference. During the learning phase, models are trained on *learning streams*, which are often scarce and costly to acquire. This scarcity amplifies the core challenge of out-of-domain generalization: Performing reliably on new *inference streams* while allowing for large distributional shifts.

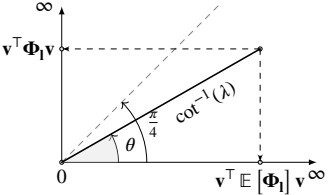

Domain invariant learning espouses a dichotomic classification of features into domain invariant and spurious. However, persistent failures (Gulrajani & Lopez-Paz, 2020) require a direct re-evaluation of invariance itself as a primary objective for out-of-domain adaptation.

Figure 1: Spectral trigonometry of state-dependence. The dependence between the $\mathbf{\Phi}$-embedding of an input stream, $\mathbf{\Phi_l}$, and the state $\mathbf{l}$ increases as the eigenvalue the ratio of the embedding of stream to that of its expectation with respect to the state deviates from 1.

We argue for a more granular control of representations: A continuous spectrum of state-dependence that asks, how much does a feature depend on the state of the world? For this spectrum to be practical, it must be interpretable, computationally tractable, and provide sufficient conditions for alignment with the learning phase objectives during inference.

**Our contribution** is to construct a Gaussian theory of dependence on the state rooted in information theory and accounting for regularity on held out states, readily implemented using an eigensolver, thus offering both the necessary language and algorithmic underpinning to rigorously identify and address the fundamental challenge of out-of-domain generalization. We proceed in three steps.

**First, a spectral theory of state dependence** is constructed by developing a novel operator algebra, $\mathcal{Q}$, which unifies the treatment of both conditional and unconditional dependence and allows for

the decomposition of representations into subspaces ranked by the strength of their coupling on the state via a standard trace-minimization argument (Fan, 1949). This will yield a simple, well defined, optimization objective for learning robust representations solvable using a single call to an eigensolver.

**Second, regularity of inference** is analyzed by modeling inference streams as **global perturbations** of the learning ones. The problem of ensuring *faithful processing*, the trustworthy application of learned representations to held-out inference streams, reduces to the stability of a structured symmetric eigenproblem (Demmel & Kågström, 1987) in $\mathcal{Q}$. Sufficient conditions are expressed in terms of the eigengap of learned representations.

**Third, out-of-domain generalization** is reframed. We shift the primary objective from the idealized pursuit of perfect invariance to the more practical goal of learning representations whose dependence on the state is stable across domain shifts. This provides a crucial property for deployed learning systems: Predictable out-of-domain performance that aligns with the intent established during the learning phase.

We analytically and empirically evaluate `LIFE` as a theory of out-of-domain generalization. Analytically, on established OOD data generating processes (Rosenfeld et al., 2020), `LIFE` attains oracle-level performance in both supervised and unsupervised tasks. Empirically, on linear hypothesis `LIFE` outperforms Invariant Risk Minimization (Arjovsky et al., 2019) while offering a simpler and stabler optimization problem, exceeds group Distributionally Robust Optimization on worst-group generalization (Sagawa et al., 2019), and, on some datasets, even outperforms deep invariant learning methods despite its linearity.

## 2 Spectral Decomposition for State-Dependence

This section builds a spectral theory for measuring and decomposing dependence on the state. Section 2.1 introduce the fundamental data-generation and acquisition processes under variable states. Section 2.2 constructs an operator division algebra allowing for algebraic probabilistic and information theoretic calculus, leading to spectral expression of symmetrized information theoretic measure in section 2.3 and subsequently the formulation of a simple optimization objective and an algorithm for learning representations with a complete spectrum of dependence on the state in section 2.4.

First, we establish our notation as follows. We use = for equality and := for assignment. Let $(\mathcal{X}, \langle \cdot, \cdot \rangle)$ be a finite dimensional inner product space. We adopt Householder's notation (Goodfellow et al., 2017); $x$ denotes a scalar, $\mathbf{x} = [x_1, \ldots, x_m]^\top$ a vector, and $\mathbf{X} = [\mathbf{x}_1 | \ldots | \mathbf{x}_n]$ a matrix.

### 2.1 Stateful Input Streams

A *state* is a random vector on a discrete *state space*. A *stateful input stream* is a random variable $\mathbf{x}$ on $\mathcal{X}$ following a Gaussian distribution indexed by a state $\mathbf{s}$ on a state space $\mathcal{S}$, $\mathbf{x} \sim \mathcal{N}(\boldsymbol{\mu}_\mathbf{s}, \boldsymbol{\Sigma}_\mathbf{s})$. A *stateful processing rule* is a conditionally Gaussian linear model (Bishop, 2011) from $\mathcal{X}$ to $\mathcal{Y}$, indexed by a state $\mathbf{t}$ on a state space $\mathcal{T}$, $\mathbf{x} \mapsto \mathbf{r}(\mathbf{x}) := \mathbf{A}_\mathbf{t}\mathbf{x} + \mathbf{b}_\mathbf{t} + \sqrt{\boldsymbol{\Gamma}_\mathbf{t}}\boldsymbol{\varepsilon}$ where $\boldsymbol{\varepsilon} \sim \mathcal{N}(\mathbf{0}, \mathbf{I})$.

The composition of stateful input streams and processing rules defines *jointly stateful joint input streams* as defined below in DGP-0.

**DGP-0** (Gaussian Interstate). Let $\mathbf{x}$ be stateful input stream, $\mathbf{r}$ a stateful processing rule, define $\mathbf{y} := \mathbf{r}(\mathbf{x})$. Then, $(\mathbf{x}, \mathbf{y}) \sim \mathcal{N}(\boldsymbol{\mu}_{[\mathbf{s}, \mathbf{t}]}, \boldsymbol{\Sigma}_{[\mathbf{s}, \mathbf{t}]})$ with,

$$\boldsymbol{\mu}_{[\mathbf{s}, \mathbf{t}]} := \begin{bmatrix} \boldsymbol{\mu}_\mathbf{s} \\ \mathbf{A}_\mathbf{t}\boldsymbol{\mu}_\mathbf{s} + \mathbf{b}_\mathbf{t} \end{bmatrix} \text{ and, } \boldsymbol{\Sigma}_{[\mathbf{s}, \mathbf{t}]} := \begin{bmatrix} \boldsymbol{\Sigma}_\mathbf{s} & \boldsymbol{\Sigma}_\mathbf{s}\mathbf{A}_\mathbf{t}^\top \\ \mathbf{A}_\mathbf{t}\boldsymbol{\Sigma}_\mathbf{s} & \mathbf{A}_\mathbf{t}\boldsymbol{\Sigma}_\mathbf{s}\mathbf{A}_\mathbf{t}^\top + \boldsymbol{\Gamma}_\mathbf{t} \end{bmatrix}.$$

While shifts in joint input streams are function of the joint state $[\mathbf{s}, \mathbf{t}]$ during data-generation. The *data acquisition process* does not observe the individual substates while being able to separate joint states. This means that there exists a state $\mathbf{l}$, a state space $\mathcal{L}$, and some unobserved bijection $g$ such that $\mathbf{l} = g(\mathbf{s}, \mathbf{t})$. Hence, the data acquisition process stores $(\mathbf{l}, \mathbf{x}, \mathbf{y})$, see figure 2.

Figure 2: Probabilistic graphical for DGP-0 and its data acquisition process.

Our goal is to construct an interpretable and practical spectral theory leading to measuring input streams dependence on the state and decomposing them into a discrete set of components ranked by

their dependence on the state. This dependence can be unconditional by considering an input stream or joint input stream as a unit, or conditional, by partitioning a joint input stream in two, in which case the measured dependence is that of a partition on the state given the remaining partition.

In order for the theory to be interpretable we desire it to mirror its information theoretic counterpart. Information theory of dependence is built on convex functionals of likelihood ratios and their factorizations. For the mirroring to work, we must give an operative spectral meaning to not only input streams but also operations on input streams, in particular, their ratio and expectation with respect to the state.

In order for theory to be practical it should readily lead to learning representations with controllable dependence on the state through a direct and clearly stated optimization problem.

The foundation of our approach is to embed DGP-0 into a division algebra of positive operators, $\mathcal{Q}$. This will lead to sizable conceptual and computational simplifications.

**Conceptually**, the algebra will allow for a unified algebraic probabilistic and information theoretic calculus; For instance, unconditional, conditional, and jointly Gaussian distributions, as well as their ratios and expectations, are all represented as single element of the operator algebra. The unification leads to disaggregation of the dependence on the state in terms of unconditional/conditional components, and subsequently, to unsupervised/supervised feature extractors.

**Computationally**, simultaneous diagonalization of collections of non-commuting matrices is notoriously difficult. The algebra folds collections of non-commuting matrices into one symmetric eigenproblem, thus sidestepping the lack of Schur decomposition for more than pairs of matrices (Kressner, 2005), the simultaneous diagonalization by congruence problem (Bustamante et al., 2020), or the polynomial eigenvalue problem (Gohberg et al., 2009). Measuring dependence on the state and learning state controlled representations are reduced to standard symmetric eigenvalue problem.

### 2.2 Smooth Operator Division Algebra

The fundamental building block of our approach is the $\boldsymbol{\Phi}$-map. This map generalizes an idea initially introduced by Siegel (1943) in the context of symplectic geometry and more recently, used in information geometry (Calvo & Oller, 1990) and manifold optimization (Hosseini & Sra, 2015). Before stating the formal definition, let's establish some intuition. A Gaussian distribution is fully described by its mean and covariance. In its simplest form, the $\boldsymbol{\Phi}$-map does embeds them jointly in a structured subset of the cone of positive definite symmetric matrix. This block matrix structure is carefully designed to turn probabilistic operations into algebraic ones. Examples of $\boldsymbol{\Phi}$-embeddings for two state bivariate Gaussian distributions are shown in figure 3.

More formally, writing $\mathcal{M}(m, n) = (\mathcal{M}_{m \times n}(\mathbb{R}), \langle \cdot, \cdot \rangle_F)$ the inner product space of rectangular matrices under the Frobenius norm induced by the Euclidean inner product on $\mathcal{X}$, $\mathcal{S}(m)$ the subspace of symmetric matrices, and $\mathcal{S}_\succ(m)$ the cone of positive definite matrices in $\mathcal{S}(m)$.

**Definition 1** ($\boldsymbol{\Phi}$-map)**.** Given $m \geq n$ positive integers, the $\boldsymbol{\Phi}$-map is the operator valued function,

$$
\begin{cases}
\mathcal{S}_\succ(m) \times \mathcal{M}(m, n) & \longrightarrow & \mathcal{Q}(m, n) \subset \mathcal{S}_\succ(m + n), \\
(\mathbf{H}, \mathbf{G}) & \longmapsto & \boldsymbol{\Phi}(\mathbf{H}, \mathbf{G}) := \begin{bmatrix} \mathbf{H} + \mathbf{G}\mathbf{G}^\top & \mathbf{G} \\ \mathbf{G}^\top & \mathbf{I}_n \end{bmatrix}.
\end{cases}
$$

$$
\boldsymbol{\Phi} \left( \quad \right) = \begin{bmatrix} 1 & 0 & 0 \\ 0 & 1 & 0 \\ 0 & 0 & 1 \end{bmatrix}
$$

(a) State 1

$$
\boldsymbol{\Phi} \left( \quad \right) = \begin{bmatrix} 1+\mu^2 & \rho & \mu \\ \rho & 1 & 0 \\ \mu & 0 & 1 \end{bmatrix}
$$

(b) State 2

Figure 3: Examples of two states $\boldsymbol{\Phi}$-embedding. (a) embedding of standard Gaussian. (b) embedding of $\mathcal{N}\left( \begin{bmatrix} \mu \\ 0 \end{bmatrix}, \begin{bmatrix} 1 & \rho \\ \rho & 1 \end{bmatrix} \right)$, rendered with $\mu = 1.2$ and $\rho = 0.6$.

Recall the direct sum (Singh et al., 2005) of two matrices $\mathbf{X}$ and $\mathbf{Y}$ forms the block diagonal matrix, $\mathbf{X} \oplus \mathbf{Y} = \begin{bmatrix} \mathbf{X} & \mathbf{0} \\ \mathbf{0} & \mathbf{Y} \end{bmatrix}$. Define the function $\mathbf{X} \mapsto \mathbf{U}(\mathbf{X}) := \begin{bmatrix} \mathbf{I}_m & \mathbf{X} \\ \mathbf{0} & \mathbf{I}_n \end{bmatrix}$. Using block Gauss-Jordan reduction (Dym, 2023), any $\boldsymbol{\Phi}$-embedding $\boldsymbol{\Phi}_{\mathbf{l}'} := \boldsymbol{\Phi}(\mathbf{H}_{\mathbf{l}'}, \mathbf{G}_{\mathbf{l}'})$ can be decomposed as the conjugation of a block diagonal matrix by an upper triangular one; $\boldsymbol{\Phi}_{\mathbf{l}'} = \mathbf{U}(\mathbf{G}_{\mathbf{l}'}) \left( \mathbf{H}_{\mathbf{l}'} \oplus \mathbf{I}_n \right) \mathbf{U}(\mathbf{G}_{\mathbf{l}'})^\top$. This decomposition allows us to define a ratio operation on $\mathcal{Q}$ turning it into a non-commutative operator valued analog of division on the positive reals. Moreover, $\mathcal{Q}$ is convex and hence algebraically closed under expectation with respect to the state. This will yield considerable simplification in the upcoming analysis. Intuitively, our ratio operation can be thought of as a centering

followed by whitening. If the two distributions are identical, this ratio is simply the identity matrix. Defining the variance of a random matrix (Muirhead, 1982) $\mathbf{X}$ as $\mathbb{V}[\mathbf{X}] = \mathbb{E}[\mathbf{X}\mathbf{X}^{\top}] - \mathbb{E}[\mathbf{X}]\,\mathbb{E}[\mathbf{X}]^{\top}$, the discussion above is encapsulated in theorem 1.

**Theorem 1** (Operator division algebra). *Let* $(\mathbf{H_l}, \mathbf{G_l}) \in \mathcal{S}_{\succ}(m) \times \mathcal{M}_{m \times n}(\mathbb{R})$, *and* $\mathbf{\Phi_l} := \mathbf{\Phi}(\mathbf{H_l}, \mathbf{G_l})$. *Then* $\mathcal{Q}(m, n)$ *is convex and closed under the binary ratio operations:*

$$(\mathbf{\Phi_l}, \mathbf{\Phi_{l'}}) \mapsto \mathbf{\Phi_l} : \mathbf{\Phi_{l'}} := \left( \mathbf{H_{l'}}^{-\frac{1}{2}} \oplus \mathbf{I}_n \right) \mathbf{U}(-\mathbf{G_{l'}}) \mathbf{\Phi_l} \mathbf{U}(-\mathbf{G_{l'}})^{\top} \left( \mathbf{H_{l'}}^{-\frac{1}{2}} \oplus \mathbf{I}_n \right). \text{ Moreover,}$$

(i) $\mathbf{\Phi_l} : \mathbf{I} = \mathbf{\Phi_l}$,

(ii) $\mathbf{\Phi_l} : \mathbf{\Phi_{l'}} = \mathbf{\Phi}\left( \mathbf{Q_{ll'}}, \mathbf{\Delta_{ll'}} \right)$, *with* $\mathbf{Q_{ll'}} := \mathbf{H_{l'}}^{-\frac{1}{2}} \mathbf{H_l} \mathbf{H_{l'}}^{-\frac{1}{2}}$ *and* $\mathbf{\Delta_{ll'}} := \mathbf{H_{l'}}^{-\frac{1}{2}}(\mathbf{G_l} - \mathbf{G_{l'}})$,

(iii) $\mathbb{E}\left[ \mathbf{\Phi_l} : \mathbf{\Phi_{l'}} \right] = \mathbf{\Phi}\left( \mathbb{E}\left[ \mathbf{Q_{ll'}} \right] + \mathbb{V}\left[ \mathbf{\Delta_{ll'}} \right], \mathbb{E}\left[ \mathbf{\Delta_{ll'}} \right] \right).$

In the following $\lambda(\mathbf{X})$ denotes the set of eigenvalues of $\mathbf{X}$, $\lambda^{\uparrow}(\mathbf{X})$ the vector of eigenvalues, including multiplicity, arrayed in non-decreasing order.

$\mathcal{Q}$ is a subset of the space of symmetric matrices; The spectral theorem applies (Hilbert, 1989). Hence, by theorem 1 $\mathcal{Q}$ carries a structured eigenvalue problem inherited by all of its elements. In fact, each element of $\mathcal{Q}$ associates a symmetric eigenvalue problem to a rational eigenvalue problem (Xi & Saad, 2015; Betcke et al., 2013); The spectrum of the former is equal to the union of the spectrum and the poles of the latter. Thankfully, it is generally simpler to solve a symmetric eigenproblem on $\mathcal{Q}$ than a rational eigenproblem (Su & Bai, 2011). By expressing the eigenvectors/spectrum of a typical element of $\mathcal{Q}$ as the critical points/values of the Rayleigh quotient (Sun, 1991), block coordinates optimization (Lange, 2016) explicitly characterizes this association.

**Lemma 1.** *Let* $\mathbf{H} \in \mathcal{S}_{\succ}(m)$, $\mathbf{G} \in \mathcal{M}_{m \times n}(\mathbb{R})$, $\mathbf{\Phi} := \mathbf{\Phi}(\mathbf{H}, \mathbf{G}) \in \mathcal{Q}(m, n)$, *and* $\mathbf{w}^{\top} = \begin{bmatrix} \mathbf{w}_1^{\top} & \mathbf{w}_2^{\top} \end{bmatrix}$ *be a unit normalized eigenvector of* $\mathbf{\Phi}$. *Then any eigenpair* $(\lambda, \mathbf{W})$ *of* $\mathbf{\Phi}$ *verifies,*

(i) *If* $\lambda = 1$, $\mathbf{G}^{\top}\mathbf{w}_1 = 0$ *and* $(\mathbf{H} - \mathbf{I})\mathbf{w}_1 = -\mathbf{G}\mathbf{w}_2$.

(ii) *If* $\lambda \neq 1$, $(\lambda, \mathbf{w})$ *is an eigenpair of* $\mathbf{\Phi}$ *if and only* $(\lambda, \mathbf{w}_1)$ *is an eigenpair of the rational eigenvalue problem,* $\mathbf{H}\mathbf{w}_1 = \lambda \left( \mathbf{I} + \frac{1}{1-\lambda} \mathbf{G}\mathbf{G}^{\top} \right) \mathbf{w}_1$, *and* $\mathbf{G}^{\top}\mathbf{w}_1 = (1 - \lambda)\mathbf{w}_2$.

An immediate application is that lemma 1 provides a unified operative interpretation for the canonical embeddings of definition 2, as well as their ratios, and expected ratios by considering them as an element of $\mathcal{Q}$. The spectrum of $\mathbf{\Phi}$-embeddings of a Gaussian distribution can be viewed as an absolute, or magnitude, spectrum; that of a ratio as a relative, or phase, spectrum. In particular, for a the embedding of Gaussian, $\mathbf{\Phi} := \mathbf{\Phi}(\mathbf{\Sigma}, \boldsymbol{\mu})$ any eigenpair $\left( \lambda, \mathbf{w}^{\top} := \begin{bmatrix} \mathbf{w}_1^{\top} & w_2 \end{bmatrix} \right)$ with $\|\mathbf{w}\|_2 = 1$ verifies $\lambda(\mathbf{\Phi}) = \mathbf{w}_1^{\top}\mathbf{\Sigma}\mathbf{w}_1 + \left( \langle \mathbf{w}_1, \boldsymbol{\mu} \rangle + w_2 \right)^2$. When $\boldsymbol{\mu} = \mathbf{0}$ any eigenpair of $\mathbf{\Sigma}$ is one of $\mathbf{\Phi}$. When $\boldsymbol{\mu} \neq \mathbf{0}$ the spectrum of $\mathbf{\Phi}$ differ from that of $\mathbf{\Sigma}$ through the addition of a quadratic term accounting for the mean. In contrast to the spectrum of $\mathbf{\Sigma}$, that of $\mathbf{\Phi}$ is mean aware. For a ratio of unconditional canonical embeddings, any such eigenpair verifies $\lambda(\mathbf{\Phi}_1 : \mathbf{\Phi}_2) = \mathbf{w}_1^{\top}(\mathbf{\Sigma}_2^{-\frac{1}{2}}\mathbf{\Sigma}_1\mathbf{\Sigma}_2^{-\frac{1}{2}})\mathbf{w}_1 + \left( \left\langle \mathbf{w}_1, \mathbf{\Sigma}_2^{\frac{1}{2}} \left( \boldsymbol{\mu}_1 - \boldsymbol{\mu}_2 \right) \right\rangle + w_2 \right)^2$.

Each eigenvalue separates into two different components: One accounting for variations of the means and another for that of the covariances. The spectrum is said to be relative as it quantifies how $\mathbf{\Phi}_1$ and $\mathbf{\Phi}_2$ differ. This "phase" information is exactly what we need to measure dependence on the state.

## 2.3 A SPECTRAL THEORY OF STATE-DEPENDENCE

In this section we spectrally express symmetric information-theoretic measure of dependence on the state by writing them as the trace of single element of $\mathcal{Q}$; State dependence is read from the spectrum of a single symmetric operator, preparing the optimization in section 2.4.

The mutual and conditional mutual information (Shannon, 1948) exhibit properties that render them well-suited for assessing dependence: They vanish if and only if the variables are independent and adhere to a chain rule (Polyanskiy & Wu, 2024) that disaggregates joint dependence into marginal and conditional components. When the joint distribution is absolutely continuous with respect to the product of marginals (Belghazi et al., 2018), the mutual information corresponds to the Kullback–Leibler divergence (Kullback, 1968).

In this study, we employ the symmetrized version of the KL divergence to define symmetrized mutual information. Recall that if $\mathbb{P}$ is absolutely continuous with respect to $\mathbb{Q}$, the Jeffreys divergence (Csiszár, 1972) is defined as $D_J(\mathbb{P} \parallel \mathbb{Q}) := KL(\mathbb{P} \parallel \mathbb{Q}) + KL(\mathbb{Q} \parallel \mathbb{P})$. Unlike the asymmetric KL divergence, Jeffreys' divergence does not include a log-partition function; This will enable exact spectral expressions over $\mathbb{Q}$.

In the probabilistic graphical model of figure 2, we define the *unconditional dependence on the state* as $S(\mathbf{x}; \mathbf{l}) := D_J\left(\mathbb{P}_{\mathbf{xl}} \parallel \mathbb{P}_{\mathbf{x}} \otimes \mathbb{P}_{\mathbf{l}}\right)$, and the *conditional dependence on the state* $S(\mathbf{y}; \mathbf{l} \mid \mathbf{x}) := D_J\left(\mathbb{P}_{\mathbf{y}|\mathbf{x}} \parallel \mathbb{P}_{\mathbf{y}|\mathbf{x}} \otimes \mathbb{P}_{\mathbf{l}|\mathbf{x}} \mid \mathbb{P}_{\mathbf{x}}\right)$. First, we must verify that the symmetrized mutual information are as equally suited to measuring dependence as their unsymmetric counterparts.

**Proposition 1** (Behaviour after measurement). *If $(\mathbf{l}, \mathbf{x}, \mathbf{y})$ are distributed according to figure 2. Then the following properties hold,*

CALIBRATED INDEPENDENCE: $S(\mathbf{x} \mid \mathbf{l}) = 0 \iff \mathbf{x} \perp\!\!\!\perp \mathbf{l}$ *and* $S(\mathbf{y}; \mathbf{l} \mid \mathbf{x}) = 0 \iff \mathbf{y} \mid \mathbf{x} \perp\!\!\!\perp \mathbf{l} \mid \mathbf{x}$.
CHAIN RULE OF INFORMATION: $S(\mathbf{x}; \mathbf{y} \mid \mathbf{l}) = S(\mathbf{x}; \mathbf{l}) + S(\mathbf{y}; \mathbf{l} \mid \mathbf{x})$.
STATE INDEPENDENT BAYES: $S(\mathbf{x}; \mathbf{l}) = S(\mathbf{y}; \mathbf{l} \mid \mathbf{x}) = 0 \implies S(\mathbf{y}; \mathbf{l}) = S(\mathbf{x}; \mathbf{l} \mid \mathbf{y}) = 0$.

We now leverage the algebraic structure offered by $\mathbb{Q}$ to uniformly handle unconditional and conditional dependence on the state. As information measure of dependence can be expressed as functionals of likelihood ratios. It is enough for us to represent marginal, conditional and joint distribution as element of $\mathbb{Q}$; Their ratios, and expectations, will be automatically handled by the ratio operation on $\mathbb{Q}$. This leads to the definition of the following canonical elements.

**Definition 2** (Canonical $\mathbf{\Phi}$-embeddings). In the notation of DGP-0, a $\mathbf{\Phi}$-embedding $\mathbf{\Phi}(\mathbf{H_l}, \mathbf{G_l})$ is called *canonical* of the *unconditional kind* if, $\mathbf{H_l} = \mathbf{\Sigma_l}$ and $\mathbf{G_l} = \mathbf{\mu_l}$, or of the *conditional kind* if

$$\mathbf{H_l} = \mathbf{\Gamma_l} \text{ and } \mathbf{G_l} = \begin{bmatrix} \mathbf{A_l} & \mathbf{b_l} \end{bmatrix} \mathbb{E}\left[\mathbf{\Phi}\left(\mathbf{\Sigma_l}, \mathbf{\mu_l}\right)\right]^{\frac{1}{2}}.$$

Note that, the conditional canonical $\mathbf{\Phi}$-embedding demonstrates that $\mathbb{Q}$ can represent more than distributions and their ratios.

Now that we have our algebraic tools, we can define our measure of state-dependence. The core idea is to measure how much each state-specific distribution $\mathbf{\Phi_l}$ deviates from a single, fixed reference element of $\mathbb{Q}$. The most natural reference is the expectation with respect to the state, the center of mass $\mathbb{E}[\mathbf{\Phi_l}]$. The relative spectrum of the random ratio $\mathbb{E}[\mathbf{\Phi_l}] : \mathbf{\Phi_l}$ quantities how much $\mathbf{\Phi_l}$ departs from the state independent $\mathbb{E}[\mathbf{\Phi_l}]$, see figure 1. Just as the aggregation of the log likelihood ratio of the joint to the product of the marginal characterizes the mutual information, $\mathbb{E}\left[\mathbb{E}[\mathbf{\Phi_l}] : \mathbf{\Phi_l}\right]$ characterize the dependence on the state. More formally,

**Theorem 2** (Spectral dependence). $\frac{1}{2}\text{Tr}\left[\mathbb{E}\left[\mathbb{E}\left[\mathbf{\Phi_l}\right] : \mathbf{\Phi_l}\right] - \mathbf{I}\right]$ *is equal to* $S(\mathbf{x}; \mathbf{l})$, *if* $\mathbf{\Phi}$ *is a canonical* $\mathbf{\Phi}$*-embedding of the unconditional kind, or* $S(\mathbf{y}; \mathbf{l} \mid \mathbf{x})$, *if* $\mathbf{\Phi}$ *is of the conditional kind.*

## 2.4 LINEARLY INDEPENDENT FEATURE EXTRACTION

We aim to identify subspaces over which state dependences decompose. Our objective is to determine the optimal rank-$k$ subspace that minimizes these dependences. Given the symmetry of $\mathbb{E}\left[\mathbb{E}\left[\mathbf{\Phi_l}\right] : \mathbf{\Phi_l}\right]$, the Ky-Fan trace minimization principle (Fan, 1949) readily applies. Writing $\text{Eig}(\mathbf{X}, \lambda) = \text{span}\{\mathbf{w} : \mathbf{Xw} = \lambda\mathbf{w}\}$ as the eigenspace of $\mathbf{X}$ associated to $\lambda$, we have,

**Corollary 1.** *Let* $\mathbf{\Phi_l}$ *be a state index canonical* $\mathbf{\Phi}$*-embedding. Then,*

$$\mathbb{R}^{m+n} = \bigoplus_{\lambda \in \lambda\left(\mathbb{E}\left[\mathbb{E}[\mathbf{\Phi_l}] : \mathbf{\Phi_l}\right]\right)} \text{Eig}\left(\mathbb{E}\left[\mathbb{E}[\mathbf{\Phi_l}] : \mathbf{\Phi_l}\right], \lambda\right)$$

$$\min_{\mathbf{W}^\top\mathbf{W}=\mathbf{I}_k} \text{Tr}\left[\mathbf{W}^\top \mathbb{E}\left[\mathbb{E}\left[\mathbf{\Phi_l}\right] : \mathbf{\Phi_l}\right]\mathbf{W}\right] = \sum_{i=1}^{k} \lambda_i^\uparrow\left(\mathbb{E}\left[\mathbb{E}\left[\mathbf{\Phi_l}\right] : \mathbf{\Phi_l}\right]\right).$$

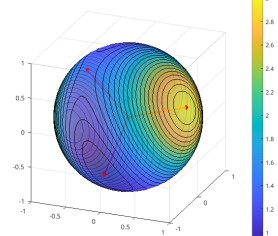

Figure 4: Contour of the expected ratio of the two state model of figure 3. Eigenvectors and eigenvalues of the value function function defined in corollary 1 are show in red.

Corollary 1 defines two fundamental algebraically invariant subspaces. 1) **The state independent**

**Algorithm 1** LIFE: A procedure

**Require:** $\mathcal{D} = [\mathcal{D}_1, \dots, \mathcal{D}_{|\mathcal{L}|}]$      # Learning data.
**Require:** $\mathbf{p} = [p_1, \dots, p_{|\mathcal{L}|}]$      # priors.
**Require:** $k \in \mathbb{N}$      # Number of eigenvectors.
**Require:** mode $\in \{$uncond,cond$\}$.
**Ensure:** A feature extractor $\mathbf{W}$.

```
 1  procedure LIFE(𝒟, p, k, mode)
 2    for l ← 1 to |ℒ| do
 3      Φ[l] ← EstimatePhi(𝒟[l], mode)
 4    end for
 5    Φ ← FormExpectedRatio(Φ, p)
 6    (Λ, W) ← Eig(Φ)
 7    return W[:, : k],          # k-smallest eigenvectors.
 8  end procedure
```

```
 1  function FormExpectedRatio(Φ, p)
 2    Φ, Ψ ← 0;
 3    for i ← 1 to |ℒ| do
 4      Ψ += p[i]Φ[i]
 5    end for
 6    for i ← 1 to |ℒ| do
 7      Φ += p[i]Φ[i]^{-1/2}ΨΦ[i]^{-1/2}
 8    end for
 9    return Φ
10  end function
```

eigenspace, $\mathcal{E}_\perp := \mathrm{Eig}\left(\mathbb{E}\left[\mathbb{E}\left[\mathbf{\Phi_l}\right] : \mathbf{\Phi_l}\right], 1\right)$, if it exists, corresponds to the space of zero dependence on state. 2) **The rank-$k$ subspace of minimal dependence** defined as the range of the spectral projector, $\mathcal{E}_k := \mathcal{R}(\mathbf{W}\mathbf{W}^\top)$.

# 3 REGULARITY OF INFERENCE

This section considers question of inference pertaining to dependence on the state. We establish sufficient conditions under which state-independent and rank-$k$ minimal dependence subspaces can faithfully process inference streams. Representation spaces hold semantic meaning (Zeiler & Fergus, 2013); Thus, it is crucial to ensure that this meaning aligns consistently with the inference streams.

We first show how the expected ratio of inference streams can be represented as **global** perturbations of the learning one. Next, we derive sufficient conditions for faithful processing in terms of the magnitude of this perturbation. Finally, we demonstrate that, under these conditions, the dependence on the state remains regular and predictable throughout inference.

## 3.1 INFERENCE STREAMS AS GLOBAL PERTURBATION

We start by partitioning the state space into disjoint learning and inference states $\mathcal{L} = \underline{\mathcal{L}} \sqcup \overline{\mathcal{L}}$. Let $\underline{\mathbf{\Phi}} := \mathbb{E}\left[\mathbb{E}\left[\mathbf{\Phi_l} \mid \underline{\mathcal{L}}\right] : \mathbf{\Phi_l} \mid \underline{\mathcal{L}}\right]$ be the expected ratio over the learning states. Similarly, let $\overline{\mathbf{\Phi}} = \mathbb{E}\left[\mathbb{E}[\mathbf{\Phi_l} \mid \overline{\mathcal{L}}] : \mathbf{\Phi_l} \mid \overline{\mathcal{L}}\right]$, be the expected ratio of inference streams which we can also write as $\overline{\mathbf{\Phi}} = \underline{\mathbf{\Phi}} + \mathbf{E}$. The fundamental question is: When is an algebraically invariant space of the expected ratio of the learning streams also one for inference streams? We will reason on the norm of the perturbation as it can cause eigenvalue coalescence (Demmel, 1986); Where at least one eigenvalue shifts from a chosen eigencluster to its spectrum's complement.

## 3.2 FAITHFUL PROCESSING AND REGULARITY OF DEPENDENCE

According to the Schur decomposition theorem (Mayers et al., 1986) , identifying conditions for the spectral resolution of the perturbation to be block upper triangular is sufficient (Kressner, 2005). This occurs if and only if an algebraic Riccati matrix equation (Lancaster & Rodman, 1995) has a solution. Conditions ensuring this solution are based on norms of perturbation blocks and spectrum eigencluster separation (Stewart et al., 1990). Specifically, if the algebraically invariant subspaces correspond to the first $k$ smallest eigenvalues without multiplicity, the separation from the spectrum is the eigengap, $\lambda_{k+1}^\uparrow(\underline{\mathbf{\Phi}}) - \lambda_k^\uparrow(\underline{\mathbf{\Phi}})$, as detailed in the following proposition.

**Theorem 3** (Conditions for faithful processing). *Let $\mathcal{E}_k$ be the rank-$k$ subspace of minimal dependence of $\underline{\mathbf{\Phi}}$. Let $\overline{\mathbf{\Phi}} = \underline{\mathbf{\Phi}} + \mathbf{E}$. If $\|\mathbf{E}\|_F < \frac{1}{2}\left(\lambda_{k+1}^\uparrow(\underline{\mathbf{\Phi}}) - \lambda_k^\uparrow(\underline{\mathbf{\Phi}})\right)$, then $\mathcal{E}_k$ is an algebraically invariant subspace of $\overline{\mathbf{\Phi}}$.*

Now that we have conditions for faithful processing we can control the regularity of the dependence on the state.

**Corollary 2** (Regularity of dependence on the state). *Let $\mathbf{l}'$ be an independent copy of $\mathbf{l}$, $p :=$ $\mathbb{P}\left((\mathbf{l}, \mathbf{l}') \in \underline{\mathscr{L}} \times \underline{\mathscr{L}}\right)$, $\mathbf{W}_k$ be a matrix with $k$ orthonormal columns such that $\mathscr{R}(\mathbf{W}_k) = \mathscr{E}_k$. Then, under conditions for faithful processing,*

$$\text{Tr}\left[\mathbf{W}_k^\top \mathbb{E}\left[\mathbb{E}[\boldsymbol{\Phi}_{\mathbf{l}}] : \boldsymbol{\Phi}_{\mathbf{l}}\right]\mathbf{W}_k\right] \leq \text{Tr}[\mathbf{W}_k^\top \underline{\boldsymbol{\Phi}}\mathbf{W}_k] + \frac{(1-p)}{2}\left(\lambda_{k+1}^\uparrow(\boldsymbol{\Phi}) - \lambda_k^\uparrow(\boldsymbol{\Phi})\right).$$

## 4 RELATED WORK

**Modal decompositions** are foundational to correspondence analysis (Hirschfeld, 1935) and Lancaster's distribution theory (Lancaster, 1958), finding use in strong data processing inequalities (Polyanskiy & Wu, 2016; Raginsky, 2014) and representation learning (Huang et al., 2024). They achieve a spectral decomposition of bivariate distribution dependencies (Lancaster, 1958) via singular value decomposition (Schmidt, 1907) (SVD) of the conditional expectation operator (Makur, 2019). However, the inherent dependence on SVD limits their usefulness as the number of component quantity is at most equal to the number of states which makes them unsuited for our setting.

**Relative Matrix Decomposition** The Generalized Singular Value Decomposition (GSVD) (Van Loan, 1976; Paige & Saunders, 1981) jointly decomposes two matrices with identical column counts into three bases: two orthonormal bases specific to each matrix and one shared basis. The GSVD finds applications in prenatal EKG (Callaerts et al., 1990) and genetics (Aiello et al., 2018). Unfortunately, the GSVD operates only on two matrices and not at the distributional level. Extensions allowing decomposition of more than two matrices have been proposed (De Lathauwer, 2011; Ponnapalli et al., 2011; Khamidullina et al., 2020), however these approach lose either the orthogonality of the subspaces or the exactness of the decomposition (Ponnapalli et al., 2011).

**Spectral Perspectives on Representation Learning** A longstanding approach interprets neural representations spectrally. Roweis & Brody (1999) examines learned linear neural representations in regression via the SVD. Linsker (1988); Deco & Obradovic (1996) relate InfoMAX to PCA via Hebbian learning. Baldi & Hornik (1989); Zhou & Liang (2017) analyze linear neural networks critical points and, using centered Gaussian distributions, and express them as permutation of principal singular vectors. Recent work extends these approach to self-supervised learning (Balestriero & LeCun, 2022) and infinite width Gaussian processes approximation of neural networks (Jacot et al., 2018; Yang, 2019).

**Domain invariant learning for Out-of-Domain Generalization** aims to learn domain-invariant predictors (Arjovsky et al., 2019; Mahajan et al., 2020; Krueger et al., 2020; Ye et al., 2021; Lin et al., 2022). However, empirical performance has often fallen short (Gulrajani & Lopez-Paz, 2020). Several studies (Rosenfeld et al., 2020; Kaur et al., 2022; Ahuja et al., 2020b) propose structural data-generating processes for learning invariant representations. Linear analyses of invariance are discussed in Wang & Veitch (2022); Zhang et al. (2025). In Krueger et al. (2020), invariance denotes statistical relationships conserved across domains, while in Li et al. (2020), invariant representations are defined as statistically independent of the domain. This dependency is analyzed in terms of unconditional/conditional mutual information in Tachet et al. (2020); Dong et al. (2024). Invariant Risk Minimization (IRM) (Arjovsky et al., 2019; Ahuja et al., 2020a; Lin et al., 2022; Ahuja et al., 2021; Zhou et al., 2022), which claims to estimates invariant causal predictors, remains the prevalent approach.

## 5 EVALUATION AS A THEORY OF OUT-OF-DOMAIN GENERALIZATION

We begin by highlighting methodological limitations in the dominant paradigm: estimating invariant causal predictors via Invariant Risk Minimization (IRM). Then we introduce LIFE as an objective for out-of-domain generalization and evaluate it both analytically and empirically.

## 5.1 Re-evaluating Invariance as a primary objective for out-of-domain generalization

The dominant out-of-distribution paradigm fixates on estimating perfectly invariant features, be it causal (Arjovsky et al., 2019) or statistical (Krueger et al., 2020), and face both methodological and practical limits. Methodologically, it treats the existence of invariant features as a precondition for learning, rather than as a falsifiable hypothesis to be tested. Practically, even when they exist and are successfully estimated, these features may lack sufficient predictive power—a failure mode known as excessive invariance (Jacobsen et al., 2018; Geirhos et al., 2020). This work argues that the fundamental challenge is one of predictable performance, where a deployed model's behavior aligns with expectations established during learning. Therefore, **to ask only if a feature is invariant is to mistake a practical problem of degree for a philosophical problem of kind**.

## 5.2 Analytical Evaluation

In this section we will demonstrate, at least in the verifiable setting established by influential OOD data-generating processes (Rosenfeld et al., 2020), that the search for causal invariant predictors reduces to identification of the state independent subspace. The appeal to the language of causality, in these analyzable settings, is an unnecessary complication. The features sought correspond precisely to a single point on our spectrum. Following, and slightly extending (Rosenfeld et al., 2020; Zhang et al., 2025), we will define DGPs at three level: 1) The topology of the conditional probabilistic graphical model, 2) The functional dependence of the nodes conditional moments on their parents, and 3) The distributions of the functional form of the conditional distribution of the nodes on their parents. The most fundamental assumption behind domain invariance is one of separability; The moments of $\mathbf{x}$ can be written as function of 1) An *invariant* or *common* random variable $\mathbf{c}$ that is independent of $\mathbf{l}$, and 2) a *spurious* or *peculiar* random variable $\mathbf{s}$ that is dependent on $\mathbf{l}$, We illustrate the decomposition in figure 5. Taking $\mathbf{A} \in \mathbb{R}^{d_c \times d}$, $\mathbf{B} \in \mathbb{R}^{d_l \times d}$ and $1 \leq d_l, d_c \leq d$. Writing, $\mathbf{z}^\top = \begin{bmatrix} \mathbf{c}^\top & \mathbf{s}^\top \end{bmatrix}$ and $\mathbf{F}^\top = \begin{bmatrix} \mathbf{A}^\top & \mathbf{B}^\top \end{bmatrix}$. Define, $\mathbf{x} = \mathbf{F}^\top \mathbf{z}$. Following (Rosenfeld et al., 2020), we assume that the latent factors of variations $\mathbf{z}$ can be recovered after mixing; Or more succinctly $\mathbf{F}$ to be injective.

**DGP-U (Unsupervised).**

$$\mathbf{c} \sim \mathcal{N}(\boldsymbol{\mu}, \boldsymbol{\Sigma}) \text{ and,} \quad \mathbf{s} \mid \mathbf{l} \sim \mathcal{N}\left(\boldsymbol{\mu}_\mathbf{l}, \boldsymbol{\Sigma}_\mathbf{l}\right),$$

$$\mathbf{x} \mid \mathbf{l} \sim \mathcal{N}\left(\mathbf{A}^\top \boldsymbol{\mu} + \mathbf{B}^\top \boldsymbol{\mu}_\mathbf{l} \mid \mathbf{A}^\top \boldsymbol{\Sigma} \mathbf{A} + \mathbf{B}^\top \boldsymbol{\Sigma}_\mathbf{l} \mathbf{B}\right).$$

**DGP-S (Supervised).**

$$\mathbf{c} \mid y \sim \mathcal{N}(y\boldsymbol{\mu}, \boldsymbol{\Sigma}) \text{ and,} \quad \mathbf{s} \mid y, \mathbf{l} \sim \mathcal{N}\left(y\boldsymbol{\mu}_\mathbf{l}, \boldsymbol{\Sigma}_\mathbf{l}\right),$$

$$\mathbf{x} \mid y, \mathbf{l} \sim \mathcal{N}\left(y(\mathbf{A}^\top \boldsymbol{\mu} + \mathbf{B}^\top \boldsymbol{\mu}_\mathbf{l}), \mathbf{A}^\top \boldsymbol{\Sigma} \mathbf{A} + \mathbf{B}^\top \boldsymbol{\Sigma}_\mathbf{l} \mathbf{B}\right).$$

For both data-generating processes the oracle distribution is defined by setting the matrix $\mathbf{B}$ to zero, the learning states are set the states 1 and 2. For DGP-S, we take $y \sim$ Rademecher ($\frac{1}{2}$).

We analyze DGP-U and DGP-S through their $\boldsymbol{\Phi}$-embeddings. We form their respective canonical expected ratio; Unconditional for the former and conditional for the latter. Analysis of the spectra of the respective expected ratios reveals the existence of state independent subspaces. Moreover, the range of the spectral projector associated with the state independent subspace is precisely equal to the feature space spanned by the invariant, or oracle, component of the data-generating processes and annihilates the spurious component.

**Proposition 2.** *The expected ratio* $\mathbb{E}\left[\mathbb{E}[\boldsymbol{\Phi}_\mathbf{l}] : \boldsymbol{\Phi}_\mathbf{l}\right]$ *of both DGP-U and DGP-S admit a state independent subspace* $\mathscr{E}_\perp$ *of dimension* $d_c$. *Moreover, a matrix* $\mathbf{W}_1^\top = \begin{bmatrix} \mathbf{W}_{11}^\top & \mathbf{W}_{12}^\top \end{bmatrix}$ *such that* $\mathscr{R}(\mathbf{W}_{11}) = \mathscr{E}_\perp$. *Then the orthogonal projection* $\mathbf{P}$ *onto* $\mathscr{R}(\mathbf{W}_{11})$ *verifies,*

*(i)* $\mathbf{BP} = \mathbf{0}$ *(Annihilation of state dependent component),*
*(ii)* $\mathscr{R}(\mathbf{A}^\top) = \mathscr{R}(\mathbf{P})$ *(Preservation of oracle component),*
*(iii)* $\mathbf{Px} = \mathbf{A}^\top \mathbf{c}$ *for DGP-U, and,* $\mathbf{Px} = y\mathbf{A}^\top \mathbf{c}$ *for DGP-S.*

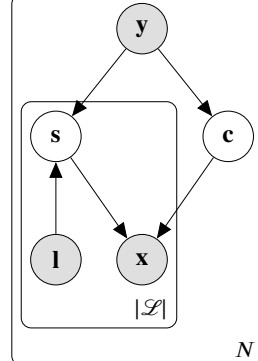

Figure 5: Probabilistic graphical model for DGP-S.

## 5.3 Empirical Evaluation

Our empirical evaluation assesses the practical utility of LIFE across a suite of standard out-of-domain (OOD) benchmarks. We aim to determine if the extracted features (i) generalize out-of-domain, (ii) achieve robust worst-group generalization, and (iii) mitigate shortcut learning, while remaining

competitive against existing methods. For each experiment, we apply LIFE to extract a rank-k subspace of minimal state dependence by selecting the eigenvectors corresponding to the smallest eigenvalues of the expected ratio matrix, as per corollary 1 .

For all experiments on non-synthetic dataset we filter the inputs using best $k$-rank approximation using randomized singular value decomposition (Halko et al., 2009; Martinsson et al., 2011). Classification tasks are resolved using a logistic regression (Cox, 1958) trained using L-BFGS (Liu & Nocedal, 1989).

**Out-of-domain generalization**: ColorMNIST modifies MNIST (LeCun, 1998) by introducing color as an additional feature. Digits less than 5 are assigned to class 0; digits 5 and above to class 1. The environment variable sets the proportion of each class rendered in red or blue, creating strong dependencies between color and label that does not carry across environments. Following the exact protocol of (Arjovsky et al., 2019), results are presented in table 1.

**Worst-group accuracy**: Waterbirds evaluates robustness to background-label associations, with images of waterbirds and land birds placed on both matching and mismatched (rare) backgrounds. Worst-group accuracy measures performance on these challenging mismatched cases, following (Sagawa et al., 2019), with results in table 2.

**Shortcut learning**: MNIST-CIFAR (Shah et al., 2020) forms two classes by vertically concatenating MNIST digits and CIFAR-10 (Hinton, 2007) images: class 0 pairs digit zero with automobiles; class 1 pairs digit one with trucks. In this setting, MNIST digits provide a shortcut signal that is easier to classify than the associated CIFAR images. We follow the protocol of (Shah et al., 2020), with results summarized in table 3.

## 6 Conclusion and Perspectives

We introduced a spectral theory of state-dependence and its algorithmic realization in LIFE, demonstrating its effectiveness on the challenge of out-of-distribution generalization. The primary contribution is the computational theory itself; OOD being its first application. It offers a principled, tractable lens for analyzing learning systems subject to changing states of the world, e.g. continual learning, federated learning, and anomaly detection. These are not extensions by analogy but consequences of the same formulation.

Table 1: On linear hypothesis, LIFE outperforms established baselines in generalizing to an environment with a shifted color-label correlation.

| Algorithm | Accuracy |
| --- | --- |
| IRMv1 | 64.85 |
| Spectral Decoupling | 63.67 |
| LIFE | **65.68** |
| Oracle | 67.44 |

Table 2: By operating on ResNet-18 features, LIFE improves worst-group accuracy on the Waterbirds dataset by effectively isolating bird features from spurious background signals

| Algorithm | Avg | Worst |
| --- | --- | --- |
| GroupDRO | 91.13 | 77.57 |
| ISR-Cov | $90.46 \pm 0.80$ | $82.46 \pm 0.55$ |
| LIFE | $89.38 \pm 0.33$ | $\mathbf{84.17 \pm 0.67}$ |

Table 3: Even when constrained to linear features, LIFE approaches oracle-level performance on MNIST-CIFAR, successfully ignoring the shortcut feature and outperforming several deep invariant learning methods.

| Algorithm | Accuracy |
| --- | --- |
| ERM | $39.5 \pm 0.4$ |
| IRMGame | $46.7 \pm 2.1$ |
| DILU | $50.2 \pm 1.7$ |
| IRMv1 | $51.3 \pm 3.0$ |
| REx | $50.1 \pm 2.2$ |
| InvRat | $52.3 \pm 0.9$ |
| BIRM (ResNet-18) | $59.3 \pm 2.3$ |
| Oracle (ResNet-18) | $83.5 \pm 1.5$ |
| Oracle (Linear) | $63.72 \pm 0.3$ |
| LIFE (Linear) | $\mathbf{63.5 \pm 0.35}$ |

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

# Appendices

## Contents

## A   TWO STATES MODEL EXAMPLE

Consider a state space with two elements $\mathscr{L} = \{0, 1\}$. Let $\mathbf{l} \sim \text{Ber}(p)$. When $\mathbf{l} = 1$, $\mathbf{x} \sim \mathcal{N}\left(\begin{bmatrix} 0 \\ 0 \end{bmatrix}, \begin{bmatrix} 1 & 0 \\ 0 & 1 \end{bmatrix}\right)$. When $\mathbf{l} = 2$, $\mathbf{x} \sim \mathcal{N}\left(\begin{bmatrix} \mu \\ 0 \end{bmatrix}, \begin{bmatrix} 1 & \rho \\ \rho & 1 \end{bmatrix}\right)$, see figure 3. Using definition 1 with $\mathbf{H}$ equal to the covariance and $\mathbf{G}$ the mean vector, we see that the $\mathbf{\Phi}$-embedding of the input stream in the first state is simply the identity matrix, $\mathbf{\Phi}_1 = \mathbf{I}_3$, that of the second state is given by,

$$\mathbf{\Phi}_2 = \begin{bmatrix} 1 + \mu^2 & \rho & \mu \\ \rho & 1 & 0 \\ \mu & 0 & 1 \end{bmatrix}.$$

The expected state is given by,

$$\mathbb{E}[\mathbf{\Phi_l}] = \mathbb{P}(\mathbf{l} = 1)\mathbf{\Phi}_1 + \mathbb{P}(\mathbf{l} = 2)\mathbf{\Phi}_2,$$
$$= p\mathbf{\Phi}_1 + (1 - p)\mathbf{\Phi}_2,$$
$$= \begin{bmatrix} 1 + \mu^2(1 - p) & \rho(1 - p) & \mu(1 - p) \\ \rho(1 - p) & 1 & 0 \\ \mu(1 - p) & 0 & 1 \end{bmatrix}.$$

Let us now form the expected ratio,

$$\mathbb{E}[\mathbb{E}[\boldsymbol{\Phi}_l] : \boldsymbol{\Phi}_l] = p\left(\mathbb{E}[\boldsymbol{\Phi}_l] : \mathbf{I}\right) + (1-p)\left(\mathbb{E}[\boldsymbol{\Phi}_l] : \boldsymbol{\Phi}_2\right),$$
$$= p\,\mathbb{E}[\boldsymbol{\Phi}_l] + (1-p)\left(\mathbb{E}[\boldsymbol{\Phi}_l] : \boldsymbol{\Phi}_2\right).$$

We need to evaluate the ratio $E[\boldsymbol{\Phi}_l] : \boldsymbol{\Phi}_2$. We have,

$$\boldsymbol{\Phi}_2 = \mathbf{U}\left(\begin{bmatrix}\mu\\0\end{bmatrix}\right)\left(\begin{bmatrix}1&\rho\\\rho&1\end{bmatrix}\oplus 1\right)\mathbf{U}\left(\begin{bmatrix}\mu\\0\end{bmatrix}\right)^{\top},$$
$$= \begin{bmatrix}1&0&\mu\\0&1&0\\0&0&1\end{bmatrix}\begin{bmatrix}1&\rho&0\\\rho&1&0\\0&0&1\end{bmatrix}\begin{bmatrix}1&0&0\\0&1&0\\\mu&0&1\end{bmatrix}$$

Now by the spectral theorem,

$$\begin{bmatrix}1&\rho\\\rho&1\end{bmatrix} = \mathbf{V}\boldsymbol{\Lambda}\mathbf{V}^{\top},$$
$$= \begin{bmatrix}-1&1\\1&1\end{bmatrix}\begin{bmatrix}1-\rho&0\\0&1+\rho\end{bmatrix}\begin{bmatrix}-1&1\\1&1\end{bmatrix}$$

Hence,

$$\begin{bmatrix}1&\rho\\\rho&1\end{bmatrix}^{-\frac{1}{2}} = \begin{bmatrix}-1&1\\1&1\end{bmatrix}\begin{bmatrix}\frac{1}{\sqrt{1-\rho}}&0\\0&\frac{1}{\sqrt{1+\rho}}\end{bmatrix}\begin{bmatrix}-1&1\\1&1\end{bmatrix}$$

Using the definition of the ratio theorem 1 we compute,

$$\mathbb{E}[\boldsymbol{\Phi}_l] : \boldsymbol{\Phi}_2 = \begin{bmatrix}1 - \frac{p\left(\mu^2\sqrt{1-\rho^2}+\mu^2+2\rho^2\right)}{2(\rho^2-1)} & \frac{\left(\mu^2+2\right)p\rho}{2(\rho^2-1)} & -\frac{\mu p\left(\sqrt{1-\rho}+\sqrt{\rho+1}\right)}{2\sqrt{1-\rho^2}}\\[2mm] \frac{\left(\mu^2+2\right)p\rho}{2(\rho^2-1)} & \frac{\mu^2 p\sqrt{1-\rho^2}+\mu^2(-p)-2p\rho^2+2\rho^2-2}{2(\rho^2-1)} & -\frac{\mu p\left(\sqrt{1-\rho}-\sqrt{\rho+1}\right)}{2\sqrt{1-\rho^2}}\\[2mm] -\frac{\mu p\left(\sqrt{1-\rho}+\sqrt{\rho+1}\right)}{2\sqrt{1-\rho}\sqrt{\rho+1}} & -\frac{\mu p\left(\sqrt{1-\rho}-\sqrt{\rho+1}\right)}{2\sqrt{1-\rho}\sqrt{\rho+1}} & 1\end{bmatrix}.$$

We can then compute the dependence on the state $\mathbf{l}$. By theorem 2

$$S(\mathbf{x};\mathbf{l}) = \frac{1}{2}\,\text{Tr}\left[\mathbb{E}\left[\boldsymbol{\Phi}_l\right] : \boldsymbol{\Phi}_l\right],$$
$$= \frac{p(1-p)\left(\mu^2\rho^2 - 2\mu^2 - 2\rho^2\right)}{2\left(\rho^2-1\right)}.$$

## B  PARAMETRIZED SPECTRAL RATIO EXAMPLE

Consider two Gaussian distributions, $\mathbb{P}_1 = \mathcal{N}\left(\begin{bmatrix}1\\\sqrt{2}+\frac{\delta}{2}\end{bmatrix}, \begin{bmatrix}1-\delta^2&0\\0&1\end{bmatrix}\right)$, $\mathbb{P}_2 = \mathcal{N}\left(\begin{bmatrix}1\\\sqrt{2}-\frac{\delta}{2}\end{bmatrix}, \begin{bmatrix}1&0\\0&1+\delta^2\end{bmatrix}\right)$. These distributions exist as long as $\delta \in (-1,1)$. By lemma 1. Forming their respective canonical embeddings. We have, $\lambda(\boldsymbol{\Phi}_1 : \boldsymbol{\Phi}_2) = \left\{1-\delta^2, 1-\frac{|\delta|}{\sqrt{\delta^2+1}}, 1+\frac{|\delta|}{\sqrt{\delta^2+1}}\right\}$. Figure 6 shows the evolution of the spectrum of the ratio as a function of $\delta$.

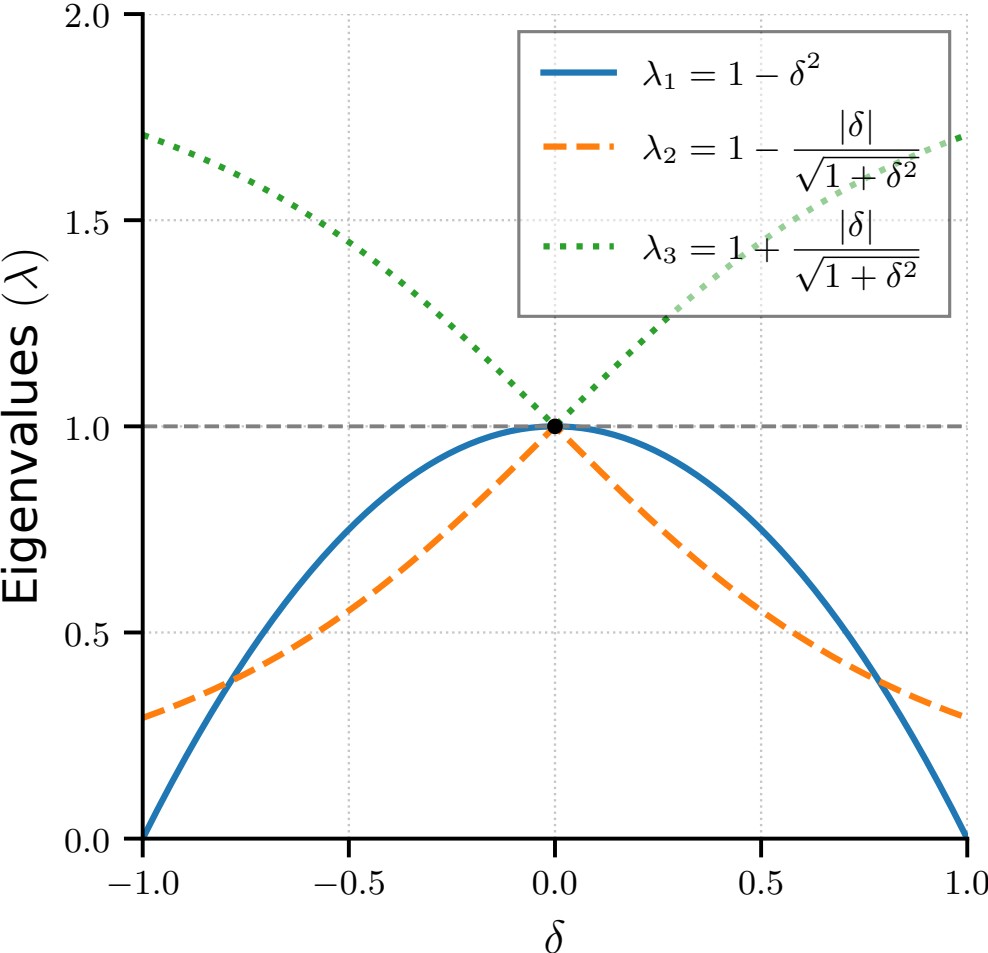

Figure 6: Spectrum of the parametrized ratio of **Φ**-embeddings.

## C CONSTRUCTING Q

### C.1 PHI-MAP IS INJECTIVE

**Proposition 3.** *The $\Phi$-map is injective.*

*Proof.* Indeed,

$$\begin{bmatrix} \mathbf{H}_1 + \mathbf{G}_1\mathbf{G}_1^\top & \mathbf{G}_1 \\ \mathbf{G}_1^\top & \mathbf{I}_n \end{bmatrix} = \begin{bmatrix} \mathbf{H}_2 + \mathbf{G}_2\mathbf{G}_2^\top & \mathbf{G}_2 \\ \mathbf{G}_2^\top & \mathbf{I}_n \end{bmatrix} \iff$$

$$\begin{bmatrix} \mathbf{H}_1 - \mathbf{H}_2 + \mathbf{G}_1\mathbf{G}_1^\top - \mathbf{G}_2\mathbf{G}_2^\top & \mathbf{G}_1 - \mathbf{G}_2 \\ \mathbf{G}_1^\top - \mathbf{G}_2^\top & \mathbf{0} \end{bmatrix} = \begin{bmatrix} \mathbf{0} & \mathbf{0} \\ \mathbf{0} & \mathbf{0} \end{bmatrix} \iff,$$

$$(\mathbf{G}_1, \mathbf{H}_1) = (\mathbf{G}_2, \mathbf{H}_2).$$

$\square$

Hence, the $\Phi$-map defines an embedding, $\mathcal{S}_{\succeq}(m) \times \mathcal{M}_{m \times n}(\mathbb{R}) \hookrightarrow \mathcal{S}_{\succeq}(m+n)$.

Next we define the $\Phi$-set.

**Definition 3** ($\mathcal{Q}$-set)**.** $\mathcal{Q}$ is the set $\mathcal{Q}(m,n) := \{\Phi(\mathbf{H}, \mathbf{G}) : \mathbf{H} \in \mathcal{S}_{\succ}(m), \mathbf{G} \in \mathcal{M}_{m \times n}(\mathbb{R})\}$.

### C.2 THE IMAGE OF THE PHI-MAP IS A SUBSET OF THE PD CONE

The next proposition show that the $\mathcal{Q}$ is indeed a subset of some cone of symmetric positive definite matrices.

**Proposition 4.** $\mathcal{Q}(m,n) \subset \mathcal{S}_{\succ}(m+n)$.

*Proof.* $\Phi := \Phi(\mathbf{H}, \mathbf{G}) \succ \mathbf{0} \iff \mathbf{I}_n \succ \mathbf{0}$ and $\Phi/\mathbf{I}_n = \mathbf{H} \succ \mathbf{0}$. $\square$

The next proposition shows that $\mathcal{Q}$ is convex.

### C.3 Q IS CONVEX IN THE PD CONE

**Proposition 5.** *Let $\boldsymbol{\pi} \in \Delta^{k-1}$, for all $i \in [k]$, $\mathbf{H}_i, \mathbf{G}_i \in \mathcal{S}_{\succ}(m) \times \mathcal{M}_{m \times n}(\mathbb{R})$. Define, $\bar{\mathbf{G}} = \sum_{i=1}^k \pi_i \mathbf{G}_i$.* $\bar{\mathbf{H}} = \sum_{i=1}^k \pi_i \left(\mathbf{H}_i + (\mathbf{G}_i - \bar{\mathbf{G}})(\mathbf{G}_i - \bar{\mathbf{G}})^\top\right)$. *Then, $\Phi(\bar{\mathbf{H}}, \bar{\mathbf{G}}) = \sum_{i=1} \pi_i \Phi(\mathbf{H}_i, \mathbf{G}_i)$.*

*Proof.* First, since $\forall i \in [k], (\mathbf{G}_i - \bar{\mathbf{G}})(\mathbf{G}_i - \bar{\mathbf{G}})^\top \succeq \mathbf{0}$ and $\mathbf{H}_i \succ \mathbf{0}$, we have that $\forall i \in [k], \pi_i \left(\mathbf{H}_i + (\mathbf{G}_i - \bar{\mathbf{G}})(\mathbf{G}_i - \bar{\mathbf{G}})^\top\right) \succ \mathbf{0}$ and hence, $\bar{\mathbf{H}} \succ \mathbf{0}$. Hence, $\Phi(\bar{\mathbf{H}}, \bar{\mathbf{G}}) \in \mathcal{Q}(m,n)$. Expanding $\bar{\Phi}$,

$$\begin{aligned} \Phi(\bar{\mathbf{H}}, \bar{\mathbf{G}}) &= \begin{bmatrix} \bar{\mathbf{H}} + \bar{\mathbf{G}}\bar{\mathbf{G}}^\top & \bar{\mathbf{G}} \\ \bar{\mathbf{G}}^\top & \mathbf{I} \end{bmatrix}, \\ &= \begin{bmatrix} \sum_{i=1}^k \pi_i \left(\mathbf{H}_i + (\mathbf{G}_i - \bar{\mathbf{G}})(\mathbf{G}_i - \bar{\mathbf{G}})^\top\right) + \bar{\mathbf{G}}\bar{\mathbf{G}}^\top & \bar{\mathbf{G}} \\ \bar{\mathbf{G}}^\top & \mathbf{I} \end{bmatrix}, \\ &= \begin{bmatrix} \sum_{i=1}^k \pi_i \left(\mathbf{H}_i + \mathbf{G}_i\mathbf{G}_i^\top\right) - \bar{\mathbf{G}}\bar{\mathbf{G}}^\top + \bar{\mathbf{G}}\bar{\mathbf{G}}^\top & \bar{\mathbf{G}} \\ \bar{\mathbf{G}}^\top & \mathbf{I} \end{bmatrix}, \\ &= \begin{bmatrix} \sum_{i=1}^k \pi_i \left(\mathbf{H}_i + \mathbf{G}_i\mathbf{G}_i^\top\right) & \bar{\mathbf{G}} \\ \bar{\mathbf{G}}^\top & \mathbf{I} \end{bmatrix}, \\ &= \sum_{i=1}^k \pi_k \begin{bmatrix} \mathbf{H}_i + \mathbf{G}_i\mathbf{G}_i^\top & \mathbf{G}_i \\ \mathbf{G}_i^\top & \mathbf{I} \end{bmatrix}, \\ &= \sum_{i=1}^k \pi_i \Phi(\mathbf{H}_i, \mathbf{G}_i). \end{aligned}$$

$\square$

**Corollary 3.** $\mathcal{Q}(m, n)$ *is a convex subset of* $\mathcal{S}_\succ(m + n)$

*Proof.* Follows from proposition 5. $\qquad\square$

## C.4 ALGEBRAIC STRUCTURE OF Q

The next lemma expresses the ratio of two element in $\mathcal{Q}$.

**Lemma 2.** *Let* $(\mathbf{H}_1, \mathbf{G}_1), (\mathbf{H}_2, \mathbf{G}_2) \in \mathcal{S}_\succ(m) \times \mathcal{M}_{m \times n}(\mathbb{R})$, *and define,* $\boldsymbol{\Phi}_1 := \boldsymbol{\Phi}(\mathbf{H}_1, \mathbf{G}_1)$, $\boldsymbol{\Phi}_2 := \boldsymbol{\Phi}(\mathbf{H}_2, \mathbf{G}_2)$. *Then,* $\boldsymbol{\Phi}_1 : \boldsymbol{\Phi}_2 = \boldsymbol{\Phi}(\mathbf{H}_2^{-\frac{1}{2}} \mathbf{H}_1 \mathbf{H}_2^{-\frac{1}{2}}, \mathbf{H}_2^{-\frac{1}{2}}(\mathbf{G}_1 - \mathbf{G}_2))$ *and,* $\boldsymbol{\Phi}_2 : \boldsymbol{\Phi}_1 = \boldsymbol{\Phi}(\mathbf{H}_1^{-\frac{1}{2}} \mathbf{H}_2 \mathbf{H}_1^{-\frac{1}{2}}, \mathbf{H}_1^{-\frac{1}{2}}(\mathbf{G}_2 - \mathbf{G}_1))$.

*Proof.* By definition of the $:$,

$$\boldsymbol{\Phi}_1 : \boldsymbol{\Phi}_2 = \begin{bmatrix} \mathbf{H}_2^{-\frac{1}{2}} \left( \mathbf{G}_1 \mathbf{G}_1^\top + \mathbf{G}_2 \mathbf{G}_2^\top + \mathbf{H}_1 - \mathbf{G}_2 \mathbf{G}_1^\top - \mathbf{G}_1 \mathbf{G}_2^\top \right) \mathbf{H}_2^{-\frac{1}{2}} & \mathbf{H}_2^{-\frac{1}{2}} \left( \mathbf{G}_1 - \mathbf{G}_2 \right) \\ \left( \mathbf{G}_1^\top - \mathbf{G}_2^\top \right) \mathbf{H}_2^{-\frac{1}{2}} & \mathbf{I}_n \end{bmatrix},$$

$$= \begin{bmatrix} \mathbf{H}_2^{-\frac{1}{2}}(\mathbf{H}_1 + (\mathbf{G}_1 - \mathbf{G}_2)(\mathbf{G}_1 - \mathbf{G}_2)^\top) \mathbf{H}_2^{-\frac{1}{2}} & \mathbf{H}_2^{-\frac{1}{2}}(\mathbf{G}_1 - \mathbf{G}_2) \\ (\mathbf{G}_1 - \mathbf{G}_2)^\top \mathbf{H}_2^{-\frac{1}{2}} & \mathbf{I}_n \end{bmatrix},$$

$$= \boldsymbol{\Phi}(\mathbf{H}_2^{-\frac{1}{2}} \mathbf{H}_1 \mathbf{H}_2^{-\frac{1}{2}}, \mathbf{H}_2^{-\frac{1}{2}}(\mathbf{G}_1 - \mathbf{G}_2)).$$

The expression for $\boldsymbol{\Phi}_2 : \boldsymbol{\Phi}_1$ follows by symmetry. $\qquad\square$

Note that, akin to division on positive reals, $:$ is not commutative.

We now start to endowing $\mathcal{Q}$ with algebraic structure. First recall,

**Definition 4** (Magma)**.** A magma, $(\mathcal{Q}, :)$ is a set $\mathcal{Q}$ closed under a binary operation $:$.

We show that $\mathcal{Q}$ is a magma,

**Corollary 4** ($\mathcal{Q}$ is a magma.)**.** $\mathcal{Q}(m, n)$ *is closed under* $:$.

*Proof.* Let $\boldsymbol{\Phi}_1, \boldsymbol{\Phi}_2 \in \mathcal{Q}(m, n)$. By proposition 3 there exists, $(\mathbf{H}_1, \mathbf{G}_1), (\mathbf{H}_2, \mathbf{G}_2) \in \mathcal{S}_\succ(m) \times \mathcal{M}_{m \times n}(\mathbb{R})$, such that $\boldsymbol{\Phi}_1 = \boldsymbol{\Phi}(\mathbf{H}_1, \mathbf{G}_1), \boldsymbol{\Phi}_2 = \boldsymbol{\Phi}(\mathbf{H}_2, \mathbf{G}_2)$. The conclusion follows from Lemma 2. $\qquad\square$

## C.5 TRACE AND DETERMINANTS OF RATIO OF PHI-EMBEDDING

Let's express the determinant and trace of ratios of $\boldsymbol{\Phi}$-embeddings.

**Proposition 6.** *Let,* $\boldsymbol{\Phi}_i := \boldsymbol{\Phi}(\mathbf{H}_i, \mathbf{G}_i)$, *for* $i \in \{1, 2\}$. *The*

(i) $\det \left[ \boldsymbol{\Phi}_1 : \boldsymbol{\Phi}_2 \right] = \det \left[ \mathbf{H}_2^{-1} \mathbf{H}_1 \right]$,

(ii) $\mathrm{Tr}[\boldsymbol{\Phi}_1 : \boldsymbol{\Phi}_2] = \mathrm{Tr}[\mathbf{H}_2^{-1} \mathbf{H}_1] + \mathrm{Tr}[(\mathbf{G}_2 - \mathbf{G}_1)^\top \mathbf{H}_2^{-1}(\mathbf{G}_2 - \mathbf{G}_1)] + n$.

*Proof.* (i) By proposition 26 $\lambda(\boldsymbol{\Phi}_1 : \boldsymbol{\Phi}_2) = \lambda(\boldsymbol{\Phi}_1, \boldsymbol{\Phi}_2)$. Moreover, $\lambda(\boldsymbol{\Phi}_1, \boldsymbol{\Phi}_2) = \lambda(\boldsymbol{\Phi}_2^{-1} \boldsymbol{\Phi}_1)$. Since the determinant is equal to the product of the eigenvalues we have, $\det \left[ \boldsymbol{\Phi}_1 : P\hat{f}_2 \right] = \det \left[ \boldsymbol{\Phi}_2^{-1} \boldsymbol{\Phi}_1 \right] = \det \left[ \boldsymbol{\Phi}_2^{-1} \right] \det \left[ \boldsymbol{\Phi}_1 \right] = \det \left[ \boldsymbol{\Phi}_2 \right]^{-1} \det \left[ \boldsymbol{\Phi}_1 \right]$, but $\det \left[ \boldsymbol{\Phi}_i \right] = \det \left[ \mathbf{H}_i \right]$, hence $\det \left[ \boldsymbol{\Phi}_1 : \boldsymbol{\Phi}_2 \right] = \det \left[ \mathbf{H}_2^{-1} \mathbf{H}_1 \right]$. (ii) Recalling that the trace of block diagonal matrix is equal to sum of the traces of the blocks and that the second block is the identity and its trace is equal to $n$. It is enough to compute the upper left hand block of $\boldsymbol{\Phi}_1 : \boldsymbol{\Phi}_2$. By lemma 2,

$$(\boldsymbol{\Phi}_1 : \boldsymbol{\Phi}_2)[: n, : n] = \mathbf{H}_2^{-\frac{1}{2}} \mathbf{H}_1 \mathbf{H}_2^{-\frac{1}{2}} + \mathbf{H}_2^{-\frac{1}{2}} \mathbf{G}_1 \mathbf{G}_1^\top \mathbf{H}_2^{-\frac{1}{2}} + \mathbf{H}_2^{-\frac{1}{2}} \mathbf{G}_2 \mathbf{G}_2^\top \mathbf{H}_2^{-\frac{1}{2}}$$

$$- \mathbf{H}_2^{-\frac{1}{2}} \mathbf{G}_2 \mathbf{G}_1^\top \mathbf{H}_2^{-\frac{1}{2}} - \mathbf{H}_2^{-\frac{1}{2}} \mathbf{G}_1 \mathbf{G}_2^\top \mathbf{H}_2^{-\frac{1}{2}}$$

$$= \mathbf{H}_2^{-\frac{1}{2}}(\mathbf{H}_1 + (\mathbf{G}_2 - \mathbf{G}_1)(\mathbf{G}_2 - \mathbf{G}_1)^\top) \mathbf{H}_2^{-\frac{1}{2}}.$$

Now, by proposition 26, $\text{Tr}[\boldsymbol{\Phi}_1 : \boldsymbol{\Phi}_2] - \text{Tr}[\mathbf{I}_n] = \text{Tr}[\mathbf{H}_2^{-\frac{1}{2}}\mathbf{H}_1\mathbf{H}_2^{-\frac{1}{2}}] + \text{Tr}[\mathbf{H}_2^{-1}(\mathbf{G}_2 - \mathbf{G}_1)(\mathbf{G}_2 - \mathbf{G}_1)^\top] = \text{Tr}[\mathbf{H}_2^{-1}\mathbf{H}_1] + \text{Tr}[(\mathbf{G}_2 - \mathbf{G}_1)^\top\mathbf{H}_2^{-1}(\mathbf{G}_2 - \mathbf{G}_1)]$. $\qquad\square$

# D  PROOFS

## D.1  THEOREM 1

**Theorem 1** (Operator division algebra). *Let* $(\mathbf{H}_l, \mathbf{G}_l) \in \mathcal{S}_{\succ}(m) \times \mathcal{M}_{m \times n}(\mathbb{R})$, *and* $\boldsymbol{\Phi}_l := \boldsymbol{\Phi}(\mathbf{H}_l, \mathbf{G}_l)$. *Then* $\mathbb{Q}(m, n)$ *is convex and closed under the binary ratio operations:*

$$(\boldsymbol{\Phi}_l, \boldsymbol{\Phi}_{l'}) \mapsto \boldsymbol{\Phi}_l : \boldsymbol{\Phi}_{l'} := \left( \mathbf{H}_{l'}^{-\frac{1}{2}} \oplus \mathbf{I}_n \right) \mathbf{U}(-\mathbf{G}_{l'})\boldsymbol{\Phi}_l\mathbf{U}(-\mathbf{G}_{l'})^\top \left( \mathbf{H}_{l'}^{-\frac{1}{2}} \oplus \mathbf{I}_n \right). \text{ Moreover,}$$

    (i) $\boldsymbol{\Phi}_l : \mathbf{I} = \boldsymbol{\Phi}_l$,

    (ii) $\boldsymbol{\Phi}_l : \boldsymbol{\Phi}_{l'} = \boldsymbol{\Phi}\left( \mathbf{Q}_{ll'}, \boldsymbol{\Delta}_{ll'} \right)$, *with* $\mathbf{Q}_{ll'} := \mathbf{H}_{l'}^{-\frac{1}{2}}\mathbf{H}_l\mathbf{H}_{l'}^{-\frac{1}{2}}$ *and* $\boldsymbol{\Delta}_{ll'} := \mathbf{H}_{l'}^{-\frac{1}{2}}(\mathbf{G}_l - \mathbf{G}_{l'})$,

    (iii) $\mathbb{E}\left[ \boldsymbol{\Phi}_l : \boldsymbol{\Phi}_{l'} \right] = \boldsymbol{\Phi}\left( \mathbb{E}\left[ \mathbf{Q}_{ll'} \right] + \mathbb{V}\left[ \boldsymbol{\Delta}_{ll'} \right], \mathbb{E}\left[ \boldsymbol{\Delta}_{ll'} \right] \right)$.

*Proof.* The convexity follows from corollary 3. Closure under : follows from corollary 4. (*i*) and (*ii*) follow from lemma 2. (*iii*) follows from (*ii*) and proposition 5. $\qquad\square$

## D.2  LEMMA 1

**Lemma 1.** *Let* $\mathbf{H} \in \mathcal{S}_{\succ}(m)$, $\mathbf{G} \in \mathcal{M}_{m \times n}(\mathbb{R})$, $\boldsymbol{\Phi} := \boldsymbol{\Phi}(\mathbf{H}, \mathbf{G}) \in \mathbb{Q}(m, n)$, *and* $\mathbf{w}^\top = \begin{bmatrix} \mathbf{w}_1^\top & \mathbf{w}_2^\top \end{bmatrix}$ *be a unit normalized eigenvector of* $\boldsymbol{\Phi}$. *Then any eigenpair* $(\lambda, \mathbf{W})$ *of* $\boldsymbol{\Phi}$ *verifies,*

    (i) *If* $\lambda = 1$, $\mathbf{G}^\top\mathbf{w}_1 = 0$ *and* $(\mathbf{H} - \mathbf{I})\mathbf{w}_1 = -\mathbf{G}\mathbf{w}_2$.

    (ii) *If* $\lambda \neq 1$, $(\lambda, \mathbf{w})$ *is an eigenpair of* $\boldsymbol{\Phi}$ *if and only if* $(\lambda, \mathbf{w}_1)$ *is an eigenpair of the rational eigenvalue problem,* $\mathbf{H}\mathbf{w}_1 = \lambda \left( \mathbf{I} + \frac{1}{1-\lambda}\mathbf{G}\mathbf{G}^\top \right) \mathbf{w}_1$, *and* $\mathbf{G}^\top\mathbf{w}_1 = (1 - \lambda)\mathbf{w}_2$.

*Proof.* Form the Rayleigh ratio, $\mathbf{w} \mapsto R[\mathbf{w}] := \frac{\mathbf{w}^\top\boldsymbol{\Phi}\mathbf{w}}{\mathbf{w}^\top\mathbf{w}}$. The critical points(values) of $R$ are the eigenvector (eigenvalues) of $\boldsymbol{\Phi}$. Partition, $\mathbf{w} = \begin{bmatrix} \mathbf{w}_1 \\ \mathbf{w}_2 \end{bmatrix}$. We'll proceed by block ascent over the eigenblocks $\mathbf{w}_2$ and $\mathbf{w}_1$. Recall, $\boldsymbol{\Phi} = \begin{bmatrix} \mathbf{H} + \mathbf{G}\mathbf{G}^\top & \mathbf{G} \\ \mathbf{G}^\top & \mathbf{I}_n \end{bmatrix}$. Maximization of $R$ over $\{\mathbf{w} \neq \mathbf{0} \mid \mathbf{w} \in \mathbb{R}^{m+n}\}$, is equivalent that of the following program,

$$\min_{(\mathbf{w}_1, \mathbf{w}_2) \in \mathbb{R}^m \times \mathbb{R}^n} - \mathbf{w}^\top\boldsymbol{\Phi}\mathbf{w},$$

$$\text{Such that, } \mathbf{w}^\top\mathbf{w} = 1.$$

We have, $\mathbf{w}^\top\boldsymbol{\Phi}\mathbf{w} = \mathbf{w}_1^\top\mathbf{H}\mathbf{w}_1 + \mathbf{w}_1^\top\mathbf{G}\mathbf{G}^\top\mathbf{w}_1 + \mathbf{w}_2^\top\mathbf{G}^\top\mathbf{w}_1 + \mathbf{w}_1^\top\mathbf{G}\mathbf{w}_2 + \mathbf{w}_2^\top\mathbf{w}_2 = \mathbf{w}_1^\top\mathbf{H}\mathbf{w}_1 + (\mathbf{w}_2 + \mathbf{G}^\top\mathbf{w}_1)^\top(\mathbf{w}_2 + \mathbf{G}^\top\mathbf{w}_1)$. Form the Lagrangian (Lagrange, 2009),

$$L[\mathbf{w}_1, \mathbf{w}_2; \lambda] = -\mathbf{w}_1^\top\mathbf{H}\mathbf{w}_1 - (\mathbf{w}_2 + \mathbf{G}^\top\mathbf{w}_1)^\top(\mathbf{w}_2 + \mathbf{G}^\top\mathbf{w}_1) + \lambda(\mathbf{w}_1^\top\mathbf{w}_1 + \mathbf{w}_2^\top\mathbf{w}_2 - 1). \quad (1)$$

We will proceed by block coordinate descent starting with $\mathbf{w}_2$. Taking the first differential, $dL(\mathbf{w}_2; d\mathbf{w}_2) = 2\,\text{Tr}\left[ \left( -(\mathbf{w}_1^\top\mathbf{G} + \mathbf{w}_2^\top) + \lambda\mathbf{w}_2^\top \right) d\mathbf{w}_2 \right]$. The first order condition holds if and only if, $\mathbf{w}_2 + \mathbf{G}^\top\mathbf{w}_1 = \lambda\mathbf{w}_2 \iff \mathbf{G}^\top\mathbf{w}_1 = (\lambda - 1)\mathbf{w}_2$. Taking first order differential of equation (1) with respect to $\mathbf{w}_1$,

$$dL(\mathbf{w}_1, d\mathbf{w}_1) = -2(\mathbf{w}_1^\top\mathbf{H} + \mathbf{w}_1^\top\mathbf{G}\mathbf{G}^\top + \mathbf{w}_2^\top\mathbf{G}^\top - \lambda\mathbf{w}_1^\top)\,d\mathbf{w}_1. \quad (2)$$

We will consider two separate cases: $\lambda = 1$ and $\lambda \neq 1$.

If $\lambda = 1$, then $\mathbf{G}^\top\mathbf{w}_1 = 0$. Hence, $dL(\mathbf{w}_1; d\mathbf{w}_1) = 0 \iff (\mathbf{H} - \mathbf{I})\mathbf{w}_1 = -\mathbf{G}\mathbf{w}_2$.

If $\lambda \neq 1$, then $\mathbf{w}_2 = \frac{1}{\lambda-1}\mathbf{G}^\top\mathbf{w}_1$. Substituting in equation (2),

$$dL(\mathbf{w}_1, d\mathbf{w}_1) = -2(\mathbf{w}_1^\top\mathbf{H} + \mathbf{w}_1^\top\mathbf{G}\mathbf{G}^\top + \frac{1}{\lambda - 1}\mathbf{w}_1^\top\mathbf{G}\mathbf{G}^\top - \lambda\mathbf{w}_1^\top)\,d\mathbf{w}_1.$$

The first order holds if and only if, $\left( \mathbf{H} + \frac{\lambda}{\lambda-1}\mathbf{G}\mathbf{G}^\top \right) \mathbf{w}_1 = \lambda\mathbf{w}_1$. $\qquad\square$

### D.3 Proposition 1

**Proposition 1** (Behaviour after measurement)**.** *If* $(\mathbf{l}, \mathbf{x}, \mathbf{y})$ *are distributed according to figure 2. Then the following properties hold,*

CALIBRATED INDEPENDENCE: $S(\mathbf{x} \mid \mathbf{l}) = 0 \iff \mathbf{x} \perp\!\!\!\perp \mathbf{l}$ *and* $S(\mathbf{y}; \mathbf{l} \mid \mathbf{x}) = 0 \iff \mathbf{y} \mid \mathbf{x} \perp\!\!\!\perp \mathbf{l} \mid \mathbf{x}.$
CHAIN RULE OF INFORMATION: $S(\mathbf{x}; \mathbf{y} \mid \mathbf{l}) = S(\mathbf{x}; \mathbf{l}) + S(\mathbf{y}; \mathbf{l} \mid \mathbf{x}).$
STATE INDEPENDENT BAYES: $S(\mathbf{x}; \mathbf{l}) = S(\mathbf{y}; \mathbf{l} \mid \mathbf{x}) = 0 \implies S(\mathbf{y}; \mathbf{l}) = S(\mathbf{x}; \mathbf{l} \mid \mathbf{y}) = 0.$

*Proof.* (*i*) Follows from Jensen's inequality. (*ii*) from theorem 4. (*iii*) from proposition 10. $\qquad\square$

#### D.3.1 Symmetric dependence on the state

**Definition and construction**

UNCONDITIONAL DEPENDENCE TO STATES    Two random variables are independent if and only if their joint distribution factors into the product of their marginals. This, in conjonction with definition 9 motivates the following definition,

**Definition 5** (Unconditional f-dependence to state)**.** The *unconditional f-dependence to state* is defined as,

$$I_f(\mathbf{x}; \mathbf{l}) := D_f\left(\mathbb{P}(\mathbf{x}, \mathbf{l}) \;||\; \mathbb{P}(\mathbf{x})\mathbb{P}(\mathbf{l})\right).$$

The next proposition expresses the unconditional $f$-dependence to states,

**Proposition 7.** *The unconditional f-dependence to state is equal to,*

$$I_f(\mathbf{x}; \mathbf{l}) = \mathbb{E}\left[ f\left( \frac{\mathbb{P}(\mathbf{x} \mid \mathbf{l})}{\mathbb{P}(\mathbf{x})} \right) \mathbb{P}(\mathbf{x} \mid \mathbf{l}') \right].$$

*Where* $\mathbf{l}'$ *is an independent copy of* $\mathbf{l}$.

*Proof.* From definition 5,

$$I_f = D_f\left(\mathbb{P}(\mathbf{x}, \mathbf{l}) \;||\; \mathbb{P}(\mathbf{x})\mathbb{P}(\mathbf{l})\right),$$

$$= \sum_{i=1}^{|\mathscr{L}|} \left( \int f\left( \frac{\mathbb{P}(\mathbf{x}, \mathbf{l}=i)}{\mathbb{P}(\mathbf{x})\mathbb{P}(\mathbf{l}=i)} \right) \mathbb{P}(\mathbf{x})\, d\mathbf{x} \right) \mathbb{P}(\mathbf{l}=i),$$

$$= \sum_{i=1}^{|\mathscr{L}|} \left( \int f\left( \frac{\mathbb{P}(\mathbf{x}, \mathbf{l}=i)}{\mathbb{P}(\mathbf{x})\mathbb{P}(\mathbf{l}=i)} \right) \sum_{j=1}^{|\mathscr{L}|} \mathbb{P}(\mathbf{x} \mid \mathbf{l}=j)\, d\mathbf{x} \right) \mathbb{P}(\mathbf{l}=i),$$

$$= \sum_{1 \le i,j \le |\mathscr{L}|} \mathbb{P}(\mathbf{l}=j)\mathbb{P}(\mathbf{l}=i) \left( \int f\left( \frac{\mathbb{P}(\mathbf{x}, \mathbf{l}=i)}{\mathbb{P}(\mathbf{x})\mathbb{P}(\mathbf{l}=i)} \right) \mathbb{P}(\mathbf{x} \mid \mathbf{l}=j)\, d\mathbf{x} \right),$$

$$= \sum_{1 \le i,j \le |\mathscr{L}|} \mathbb{P}(\mathbf{l}=j)\mathbb{P}(\mathbf{l}=i) \left( \int f\left( \frac{\mathbb{P}(\mathbf{x} \mid \mathbf{l}=i)}{\mathbb{P}(\mathbf{x})} \right) \mathbb{P}(\mathbf{x} \mid \mathbf{l}=j)\, d\mathbf{x} \right).$$

$\qquad\square$

The uncondtional $f$-dependence is determined by convex combinations of terms of the form, $G_f(i,j) := \int f\left( \frac{\mathbb{P}(\mathbf{x} \mid \mathbf{l}=i)}{\mathbb{P}(\mathbf{x})} \right) \mathbb{P}(\mathbf{x} \mid \mathbf{l}=j)\, d\mathbf{x}.$. If we consider the f-dependence with generator $t \mapsto f(t) = t \ln t$. The unconditional dependence to state is exactly the mutual information of $\mathbf{x}$ and $\mathbf{l}$.

**Corollary 5.** *Let* $f(t) = t \ln t$, *the Kullblack-Leibler divergence generator. Then,*

$$I_{KL}(\mathbf{x}; \mathbf{l}) = I(\mathbf{x}; \mathbf{l}) = H[\mathbb{P}(\mathbf{x})] - \mathbb{E}\left[ H[\mathbb{P}(\mathbf{x} \mid \mathbf{l})] \right].$$

*The mutual information of* $\mathbf{x}$ *and* $\mathbf{l}$.

*Proof.* Indeed in this setting,

$$G_f(i,j) = \left( \frac{\mathbb{P}(\mathbf{x} \mid \mathbf{l}=i)}{\mathbb{P}(\mathbf{x})} \right) \ln \left( \frac{\mathbb{P}(\mathbf{x} \mid \mathbf{l}=i)}{\mathbb{P}(\mathbf{x})} \right) \mathbb{P}(\mathbf{x} \mid \mathbf{l}=j),$$

$$= \left( \frac{\mathbb{P}(\mathbf{x} \mid \mathbf{l}=i)}{\mathbb{P}(\mathbf{x})} \right) (\ln \mathbb{P}(\mathbf{x} \mid \mathbf{l}=i) - \ln \mathbb{P}(\mathbf{x})) \, \mathbb{P}(\mathbf{x} \mid \mathbf{l}=j),$$

$$= \left( \frac{\mathbb{P}(\mathbf{x} \mid \mathbf{l}=i)\mathbb{P}(\mathbf{x} \mid \mathbf{l}=j)}{\mathbb{P}(\mathbf{x})} \right) (\ln \mathbb{P}(\mathbf{x} \mid \mathbf{l}=i) - \ln \mathbb{P}(\mathbf{x})),$$

$$= \left( \ln \mathbb{P}(\mathbf{x} \mid \mathbf{l}=i) \left( \frac{\mathbb{P}(\mathbf{x} \mid \mathbf{l}=i)\mathbb{P}(\mathbf{x} \mid \mathbf{l}=j)}{\mathbb{P}(\mathbf{x})} \right) - \ln \mathbb{P}(\mathbf{x}) \left( \frac{\mathbb{P}(\mathbf{x} \mid \mathbf{l}=i)\mathbb{P}(\mathbf{x} \mid \mathbf{l}=j)}{\mathbb{P}(\mathbf{x})} \right) \right).$$

Summing over $j$ with weights $\mathbb{P}(\mathbf{l}=j)$,

$$\sum_{j=1}^{|\mathscr{L}|} \mathbb{P}(\mathbf{l}=j) G_f(i,j) = \ln \mathbb{P}(\mathbf{x} \mid \mathbf{l}=i) \left( \frac{\mathbb{P}(\mathbf{x} \mid \mathbf{l}=i) \sum_{j=1}^{|\mathscr{L}|} \mathbb{P}(\mathbf{l}=j)\mathbb{P}(\mathbf{x} \mid \mathbf{l}=j)}{\mathbb{P}(\mathbf{x})} \right)$$

$$- \ln \mathbb{P}(\mathbf{x}) \left( \frac{\mathbb{P}(\mathbf{x} \mid \mathbf{l}=i) \sum_{j=1}^{|\mathscr{L}|} \mathbb{P}(\mathbf{l}=j)\mathbb{P}(\mathbf{x} \mid \mathbf{l}=j)}{\mathbb{P}(\mathbf{x})} \right),$$

$$= \ln \mathbb{P}(\mathbf{x} \mid \mathbf{l}=i) \left( \frac{\mathbb{P}(\mathbf{x} \mid \mathbf{l}=i)\mathbb{P}(\mathbf{x})}{\mathbb{P}(\mathbf{x})} \right) - \ln \mathbb{P}(\mathbf{x}) \left( \frac{\mathbb{P}(\mathbf{x} \mid \mathbf{l}=i)\mathbb{P}(\mathbf{x})}{\mathbb{P}(\mathbf{x})} \right),$$

$$= \ln \mathbb{P}(\mathbf{x} \mid \mathbf{l}=i)\mathbb{P}(\mathbf{x} \mid \mathbf{l}=i) - \ln \mathbb{P}(\mathbf{x})\mathbb{P}(\mathbf{x} \mid \mathbf{l}=i).$$

Summing over $i$ with weights $\mathbb{P}(\mathbf{l}=i)$,

$$\sum_{1 \leq i,j \leq |\mathscr{L}|} \mathbb{P}(\mathbf{l}=j) G_f(i,j) = \sum_{i=1}^{|\mathscr{L}|} \mathbb{P}(\mathbf{l}=i) \ln \mathbb{P}(\mathbf{x} \mid \mathbf{l}=i)\mathbb{P}(\mathbf{x} \mid \mathbf{l}=i) - \ln \mathbb{P}(\mathbf{x})\mathbb{P}(\mathbf{x}).$$

Integrating with respect to $\mathbb{P}(\mathbf{x})$,

$$U_f = \sum_{i=1}^{|\mathscr{L}|} \mathbb{P}(\mathbf{l}=i) \int \ln \mathbb{P}(\mathbf{x} \mid \mathbf{l}=i)\mathbb{P}(\mathbf{x} \mid \mathbf{l}=i) \, d\mathbf{x} - \int \ln \mathbb{P}(\mathbf{x})\mathbb{P}(\mathbf{x}) \, d\mathbf{x},$$

$$= H[\mathbb{P}(\mathbf{x})] - \sum_{i=1}^{|\mathscr{L}|} \mathbb{P}(\mathbf{l}=i) H[\mathbb{P}(\mathbf{x} \mid \mathbf{l}=i)] = H[\mathbb{P}(\mathbf{x})] - \mathbb{E}\left[ H[\mathbb{P}(\mathbf{x} \mid \mathbf{l})] \right] = I(\mathbf{x}; \mathbf{l}).$$

$\square$

Now we consider the reverse Kullblack-Leibler divergence.

**Corollary 6.** *Let $f(t) = -\ln t$, reverse Kullblack-Leibler divergence generator. Then,*

$$I_{LK}(\mathbf{x}; \mathbf{l}) = \mathbb{E}[H[\mathbb{P}(\mathbf{x} \mid \mathbf{l}), \mathbb{P}(\mathbf{x} \mid \mathbf{l}')] - H[\mathbb{P}(\mathbf{x})].$$

*Where $\mathbf{l}'$ is independent copy of $\mathbf{l}$.*

*Proof.* Indeed in this setting, $-G_f(i,j) = \ln \left( \frac{\mathbb{P}(\mathbf{x}|\mathbf{l}=i)}{\mathbb{P}(\mathbf{x})} \right) \mathbb{P}(\mathbf{x} \mid \mathbf{l}=j) = (\ln \mathbb{P}(\mathbf{x} \mid \mathbf{l}=i) - \ln \mathbb{P}(\mathbf{x})) \, \mathbb{P}(\mathbf{x} \mid \mathbf{l}=j)$. Summing over $j$ with weights $\mathbb{P}(\mathbf{l}=j)$,

$$- \sum_{j=1}^{|\mathscr{L}|} \mathbb{P}(\mathbf{l}=j) G_f(i,j) = \sum_{j=1}^{|\mathscr{L}|} \mathbb{P}(\mathbf{l}=j) \ln \mathbb{P}(\mathbf{x} \mid \mathbf{l}=i)\mathbb{P}(\mathbf{x} \mid \mathbf{l}=j) - \ln \mathbb{P}(\mathbf{x})\mathbb{P}(\mathbf{x}).$$

Integrating with respect to $\mathbf{x}$, and summing over $i$ with weights $\mathbb{P}(\mathbf{l}=i)$,

$$-U_f(\mathbf{x}; \mathbf{l}) = \sum_{1 \leq i,j \leq |\mathscr{L}|} \mathbb{P}(\mathbf{l}=i)\mathbb{P}(\mathbf{l}=j) \int \ln \mathbb{P}(\mathbf{x} \mid \mathbf{l}=i)\mathbb{P}(\mathbf{x} \mid \mathbf{l}=j) \, d\mathbf{x} - \int \ln \mathbb{P}(\mathbf{x})\mathbb{P}(\mathbf{x}) \, d\mathbf{x},$$

$$= H[\mathbb{P}(\mathbf{x})] - \sum_{1 \leq i,j \leq |\mathscr{L}|} \mathbb{P}(\mathbf{l}=i)\mathbb{P}(\mathbf{l}=j) H[\mathbb{P}(\mathbf{x} \mid \mathbf{l}=i), \mathbb{P}(\mathbf{x} \mid \mathbf{l}=j)],$$

$$= H[\mathbb{P}(\mathbf{x})] - \mathbb{E}[H[\mathbb{P}(\mathbf{x} \mid \mathbf{l}), \mathbb{P}(\mathbf{x} \mid \mathbf{l}')].$$

And hence, $U_f(\mathbf{x}; \mathbf{l}) = \mathbb{E}[H[\mathbb{P}(\mathbf{x} \mid \mathbf{l}), \mathbb{P}(\mathbf{x} \mid \mathbf{l}')] - H[\mathbb{P}(\mathbf{x})].$ $\square$

By exploiting exploiting the properties of $f$-divergences we can readily express the analogue for the symmetrized Kullblack-Leilber divergence,

**Corollary 7.** *When $f(t) = (t-1) \ln t$, the Jeffrey's divergence generator,*

$$I_J(\mathbf{x}; \mathbf{l}) = \mathbb{E}\left[KL\left(\mathbb{P}(\mathbf{x} \mid \mathbf{l}) \mid\mid \mathbb{P}(\mathbf{x} \mid \mathbf{l}')\right)\right].$$

*Where $\mathbf{l}'$ is an independent copy of $\mathbf{l}$.*

*Proof.* Let $g(t) = t \ln t$ and $h(t) = -\ln t$ denote the respective generators of the Kullblack-Leibler and reverse Kullblack-Leibler divergences. We can write, $f(t) = g(t) + h(t)$. Hence by Proposition 5, $U_f = S_g + S_h$. By 5 and 6,

$$I_J(\mathbf{x}; \mathbf{l}) = \mathbb{E}[H[\mathbb{P}(\mathbf{x} \mid \mathbf{l}), \mathbb{P}(\mathbf{x} \mid \mathbf{l}')] - H[\mathbb{P}(\mathbf{x})] + H[\mathbb{P}(\mathbf{x})] - \mathbb{E}[H[\mathbb{P}(\mathbf{x} \mid \mathbf{l})]],$$

$$= \mathbb{E}[H[\mathbb{P}(\mathbf{x} \mid \mathbf{l}), \mathbb{P}(\mathbf{x} \mid \mathbf{l}')] - \mathbb{E}[H[\mathbb{P}(\mathbf{x} \mid \mathbf{l})]] = \mathbb{E}\left[KL\left(\mathbb{P}(\mathbf{x} \mid \mathbf{l}) \mid\mid \mathbb{P}(\mathbf{x} \mid \mathbf{l}')\right)\right].$$

$\square$

Which we can express in terms of symmetrized Kullblack-Leibler divergences pairs sampled without replacement

**Corollary 8.**

$$I_J(\mathbf{x}; \mathbf{l}) = \frac{1}{2} \mathbb{E}\left[D_J\left(\mathbb{P}(\mathbf{x} \mid \mathbf{l}) \mid\mid \mathbb{P}(\mathbf{x} \mid \mathbf{l}')\right)\right].$$

*Where $\mathbf{l}$ is an independent copy of $\mathbf{l}'$.*

*Proof.* Follows from the fact that Jeffreys' divergence and symmetric and corollary 7. $\square$

CONDITIONAL   We can now move to characterizing the conditional dependence. $\mathbf{y}$ and $\mathbf{l}$ are independent given $\mathbf{x}$ if and and only, $\mathbb{P}(\mathbf{y}, \mathbf{l} \mid \mathbf{x}) = \mathbb{P}(\mathbf{y} \mid \mathbf{x})\mathbb{P}(\mathbf{l} \mid \mathbf{x})$. This prompts the following definition.

**Definition 6** (conditional f-dependence to state). $I_f(\mathbf{y}; \mathbf{l} \mid \mathbf{x}) := D_f(\mathbb{P}(\mathbf{y}, \mathbf{l} \mid \mathbf{x}) \mid\mid \mathbb{P}(\mathbf{y} \mid \mathbf{x})\mathbb{P}(\mathbf{l} \mid \mathbf{x}) \mid \mathbb{P}(\mathbf{x}))$.

The following proposition yields a handful of useful expression for the conditional dependence.

**Corollary 9.** *Let $\mathbf{x} \mapsto U_f(\mathbf{x}) = D_f(\mathbb{P}(\mathbf{y}, \mathbf{l} \mid \mathbf{x}) \mid\mid \mathbb{P}(\mathbf{y} \mid \mathbf{x})\mathbb{P}(\mathbf{l} \mid \mathbf{x}))$. The following are all equal to $I_f(\mathbf{y}; \mathbf{l} \mid \mathbf{x})$.*

(i) $\int U_f(\mathbf{x})\mathbb{P}(\mathbf{x}) \, d\mathbf{x}$,

(ii) $\int \left(\sum_{i=1}^{|\mathscr{L}|} \int f\left(\frac{\mathbb{P}(\mathbf{y}, \mathbf{l}=i \mid \mathbf{x})}{\mathbb{P}(\mathbf{y}|\mathbf{x})\mathbb{P}(\mathbf{l}=i|\mathbf{x})}\right) \mathbb{P}(\mathbf{y} \mid \mathbf{x})\mathbb{P}(\mathbf{l}=i \mid \mathbf{x}) \, d\mathbf{y}\right) \mathbb{P}(\mathbf{x}) \, d\mathbf{x}$,

(iii) $\int \sum_{1 \leq i, j \leq |\mathscr{L}|} \mathbb{P}(\mathbf{x})\mathbb{P}(\mathbf{l}=i \mid \mathbf{x})\mathbb{P}(\mathbf{l}=j \mid \mathbf{x}) \left(\int f\left(\frac{\mathbb{P}(\mathbf{y}|\mathbf{l}=i,\mathbf{x})}{\mathbb{P}(\mathbf{y}|\mathbf{x})}\right) \mathbb{P}(\mathbf{y} \mid \mathbf{l}=j, \mathbf{x}) \, d\mathbf{y}\right) d\mathbf{x}$,

(iv) $\sum_{1 \leq i,j,k \leq |\mathscr{L}|} \mathbb{P}(\mathbf{l}=k) \int \mathbb{P}(\mathbf{l}=i \mid \mathbf{x})\mathbb{P}(\mathbf{l}=j \mid \mathbf{x}) \left(\int f\left(\frac{\mathbb{P}(\mathbf{y}|\mathbf{l}=i,\mathbf{x})}{\mathbb{P}(\mathbf{y}|\mathbf{x})}\right) \mathbb{P}(\mathbf{y} \mid \mathbf{l}=j, \mathbf{x}) \, d\mathbf{y}\right) \mathbb{P}(\mathbf{x} \mid \mathbf{l}=k) \, d\mathbf{x}$.

*Proof.* This in turn can be expressed as,

$$I_f(\mathbf{y}; \mathbf{l} \mid \mathbf{x}) = \int \sum_{1 \leq i,j \leq |\mathscr{L}|} \mathbb{P}(\mathbf{l}=i \mid \mathbf{x})\mathbb{P}(\mathbf{l}=j \mid \mathbf{x}) \left(\int f\left(\frac{\mathbb{P}(\mathbf{y} \mid \mathbf{l}=i, \mathbf{x})}{\mathbb{P}(\mathbf{y} \mid \mathbf{x})}\right) \mathbb{P}(\mathbf{y} \mid \mathbf{l}=j, \mathbf{x}) \, d\mathbf{y}\right) \mathbb{P}(\mathbf{x}) \, d\mathbf{x},$$

$$= \int \sum_{1 \leq i,j \leq |\mathscr{L}|} \mathbb{P}(\mathbf{x})\mathbb{P}(\mathbf{l}=i \mid \mathbf{x})\mathbb{P}(\mathbf{l}=j \mid \mathbf{x}) \left(\int f\left(\frac{\mathbb{P}(\mathbf{y} \mid \mathbf{l}=i, \mathbf{x})}{\mathbb{P}(\mathbf{y} \mid \mathbf{x})}\right) \mathbb{P}(\mathbf{y} \mid \mathbf{l}=j, \mathbf{x}) \, d\mathbf{y}\right) d\mathbf{x}, \quad (iii).$$

By Bayes theorem we have,

$$\mathbb{P}(\mathbf{l}=i \mid \mathbf{x}) = \frac{\mathbb{P}(\mathbf{l}=i)\mathbb{P}(\mathbf{x} \mid \mathbf{l}=i)}{\mathbb{P}(\mathbf{x})}.$$

Hence,

$$I_f(\mathbf{y}; \mathbf{l} \mid \mathbf{x}) = \sum_{1 \le i,j \le |\mathscr{L}|} \mathbb{P}(\mathbf{l} = i) \int \mathbb{P}(\mathbf{x} \mid \mathbf{l} = i)\mathbb{P}(\mathbf{l} = j \mid \mathbf{x}) \left( \int f\left( \frac{\mathbb{P}(\mathbf{y} \mid \mathbf{l} = i, \mathbf{x})}{\mathbb{P}(\mathbf{y} \mid \mathbf{x})} \right) \mathbb{P}(\mathbf{y} \mid \mathbf{l} = j, \mathbf{x}) \, d\mathbf{y} \right) d\mathbf{x}.$$

Moreover,

$$I_f(\mathbf{y}; \mathbf{l} \mid \mathbf{x}) = \int \sum_{1 \le i,j \le |\mathscr{L}|} \mathbb{P}(\mathbf{l} = i \mid \mathbf{x})\mathbb{P}(\mathbf{l} = j \mid \mathbf{x}) \left( \int f\left( \frac{\mathbb{P}(\mathbf{y} \mid \mathbf{l} = i, \mathbf{x})}{\mathbb{P}(\mathbf{y} \mid \mathbf{x})} \right) \mathbb{P}(\mathbf{y} \mid \mathbf{l} = j, \mathbf{x}) \, d\mathbf{y} \right) \mathbb{P}(\mathbf{x}) \, d\mathbf{x},$$

$$= \sum_{1 \le i,j,k \le |\mathscr{L}|} \mathbb{P}(\mathbf{l} = k) \int \mathbb{P}(\mathbf{l} = i \mid \mathbf{x})\mathbb{P}(\mathbf{l} = j \mid \mathbf{x}) \left( \int f\left( \frac{\mathbb{P}(\mathbf{y} \mid \mathbf{l} = i, \mathbf{x})}{\mathbb{P}(\mathbf{y} \mid \mathbf{x})} \right) \mathbb{P}(\mathbf{y} \mid \mathbf{l} = j, \mathbf{x}) \, d\mathbf{y} \right) \mathbb{P}(\mathbf{x} \mid \mathbf{l} = k) \, d\mathbf{x}. \quad (iv)$$

$\square$

We can now readily express the $LK$, $KL$ and $J$ conditional sensitivies,

**Corollary 10.** $I_{KL}(\mathbf{y}; \mathbf{l} \mid \mathbf{x}) = H(\mathbf{y} \mid \mathbf{x}) - \mathbb{E}\left[ H\left[ \mathbb{P}(\mathbf{y} \mid \mathbf{l}, \mathbf{x}) \right] \right].$

*Proof.* By corollary 5, $U_{KL}(\mathbf{x}) = H[\mathbb{P}(\mathbf{y} \mid \mathbf{x})] - \mathbb{E}\left[ H\left[ \mathbb{P}(\mathbf{y} \mid \mathbf{l}, \mathbf{x}) \right] \mid \mathbf{x} \right]$. Hence,

$$I_{KL}(\mathbf{y}; \mathbf{l} \mid \mathbf{x}) = \mathbb{E}\left[ H\left[ \mathbb{P}(\mathbf{y} \mid \mathbf{x}) \right] \right] - \mathbb{E}\left[ \mathbb{E}\left[ H\left[ \mathbb{P}(\mathbf{y} \mid \mathbf{l}, \mathbf{x}) \right] \mid \mathbf{x} \right] \right],$$
$$= H(\mathbf{y} \mid \mathbf{x}) - \mathbb{E}\left[ H\left[ \mathbb{P}(\mathbf{y} \mid \mathbf{l}, \mathbf{x}) \right] \right].$$

$\square$

**Proposition 8.** $I_{LK}(\mathbf{y}; \mathbf{l} \mid \mathbf{x}) = \mathbb{E}[H[\mathbb{P}(\mathbf{y} \mid \mathbf{l}, \mathbf{x}), \mathbb{P}(\mathbf{y} \mid \mathbf{l}', \mathbf{x}) \mid \mathbf{x}] - \mathbb{E}[H[\mathbb{P}(\mathbf{y} \mid \mathbf{x})]].$

*Proof.* By corollary 5 $U_{LK}(\mathbf{x}) = \mathbb{E}[H[\mathbb{P}(\mathbf{y} \mid \mathbf{l}, \mathbf{x}), \mathbb{P}(\mathbf{y} \mid \mathbf{l}', \mathbf{x}) \mid \mathbf{x}] - H[\mathbb{P}(\mathbf{y} \mid \mathbf{x})]$. Hence,

$$I_{LK}(\mathbf{y}; \mathbf{l} \mid \mathbf{x}) = \mathbb{E}[\mathbb{E}[H[\mathbb{P}(\mathbf{y} \mid \mathbf{l}, \mathbf{x}), \mathbb{P}(\mathbf{y} \mid \mathbf{l}', \mathbf{x}) \mid \mathbf{x}]] - \mathbb{E}[H[\mathbb{P}(\mathbf{y} \mid \mathbf{x})]],$$
$$= \mathbb{E}[H[\mathbb{P}(\mathbf{y} \mid \mathbf{l}, \mathbf{x}), \mathbb{P}(\mathbf{y} \mid \mathbf{l}', \mathbf{x}) \mid \mathbf{x}] - \mathbb{E}[H[\mathbb{P}(\mathbf{y} \mid \mathbf{x})]].$$

$\square$

**Corollary 11.** $I_J(\mathbf{y}; \mathbf{l} \mid \mathbf{x}) = \mathbb{E}\left[ KL\left( \mathbb{P}(\mathbf{y} \mid \mathbf{l}, \mathbf{x}) \| \mathbb{P}(\mathbf{y} \mid \mathbf{l}', \mathbf{x}) \mid \mathbb{P}(\mathbf{x}) \right) \right]$, *Where $\mathbf{l}'$ is independent copy of $\mathbf{l}$.*

As in corollary 7 we can express the conditional dependence to state with respect to the symmetrized Kullback-Leibler divergence.

**Corollary 12.** $I_J(\mathbf{y}; \mathbf{l} \mid \mathbf{x}) = \frac{1}{2} \mathbb{E}\left[ D_J\left( \mathbb{P}(\mathbf{y} \mid \mathbf{x}, \mathbf{l}) \| \mathbb{P}(\mathbf{y} \mid \mathbf{x}, \mathbf{l}') \mid \mathbb{P}(\mathbf{x}) \right) \right]$ *., Where $\mathbf{l}'$ is an independent copy of $\mathbf{l}$.*

**Chain rule of information** We can now establish the information Bayes theorem for the $I_J$. First we need to establish the chain rule for the Kullback-Leibler divergence

**Lemma 3** (Chain rule for the Kullback-Leibler divergence). $KL\left( \mathbb{P}_{\mathbf{xy}} \| \mathbb{Q}_{\mathbf{xy}} \right) = KL\left( \mathbb{P}_{\mathbf{y}|\mathbf{x}} \| \mathbb{Q}_{\mathbf{y}|\mathbf{x}} \mid \mathbb{P}_{\mathbf{x}} \right) + KL\left( \mathbb{P}_{\mathbf{x}} \| \mathbb{Q}_{\mathbf{x}} \right).$

*Proof.* (Polyanskiy & Wu, 2024) $\square$

**Proposition 9.**

*Proof.* Start by decomposing the likelihood ratio,

$$\frac{\mathbb{P}_{\mathbf{xyl}}}{\mathbb{P}_{\mathbf{xy}}\mathbb{P}_{\mathbf{l}}} = \frac{\mathbb{P}_{\mathbf{yl}|\mathbf{x}}}{\mathbb{P}_{\mathbf{y}|\mathbf{x}}\mathbb{P}_{\mathbf{l}|\mathbf{x}}} \frac{\mathbb{P}_{\mathbf{xl}}}{\mathbb{P}_{\mathbf{x}}\mathbb{P}_{\mathbf{l}}}. \tag{3}$$

Applying the function $t \mapsto t \log t$ on each side of equation (3),

$$\frac{\mathbb{P}_{\mathbf{xyl}}}{\mathbb{P}_{\mathbf{xy}}\mathbb{P}_{\mathbf{l}}} \log \left( \frac{\mathbb{P}_{\mathbf{xyl}}}{\mathbb{P}_{\mathbf{xy}}\mathbb{P}_{\mathbf{l}}} \right) = \frac{\mathbb{P}_{\mathbf{yl|x}}}{\mathbb{P}_{\mathbf{y|x}}\mathbb{P}_{\mathbf{l|x}}} \frac{\mathbb{P}_{\mathbf{xl}}}{\mathbb{P}_{\mathbf{x}}\mathbb{P}_{\mathbf{l}}} \log \left( \frac{\mathbb{P}_{\mathbf{yl|x}}}{\mathbb{P}_{\mathbf{y|x}}\mathbb{P}_{\mathbf{l|x}}} \frac{\mathbb{P}_{\mathbf{xl}}}{\mathbb{P}_{\mathbf{x}}\mathbb{P}_{\mathbf{l}}} \right),$$

$$= \frac{\mathbb{P}_{\mathbf{yl|x}}}{\mathbb{P}_{\mathbf{y|x}}\mathbb{P}_{\mathbf{l|x}}} \frac{\mathbb{P}_{\mathbf{xl}}}{\mathbb{P}_{\mathbf{x}}\mathbb{P}_{\mathbf{l}}} \left( \log \left( \frac{\mathbb{P}_{\mathbf{yl|x}}}{\mathbb{P}_{\mathbf{y|x}}\mathbb{P}_{\mathbf{l|x}}} \right) + \log \left( \frac{\mathbb{P}_{\mathbf{xl}}}{\mathbb{P}_{\mathbf{x}}\mathbb{P}_{\mathbf{l}}} \right) \right).$$

Taking the Expectation with respect to $\mathbb{P}_{\mathbf{xy}}\mathbb{P}_{\mathbf{l}}$ the left hand side is equal to,

$$\mathbb{E}_{\mathbb{P}_{\mathbf{xy}}\mathbb{P}_{\mathbf{l}}} \left[ \frac{\mathbb{P}_{\mathbf{xyl}}}{\mathbb{P}_{\mathbf{xy}}\mathbb{P}_{\mathbf{l}}} \log \left( \frac{\mathbb{P}_{\mathbf{xyl}}}{\mathbb{P}_{\mathbf{xy}}\mathbb{P}_{\mathbf{l}}} \right) \right] = \mathbb{E}_{\mathbb{P}_{\mathbf{xyl}}} \left[ \log \left( \frac{\mathbb{P}_{\mathbf{xyl}}}{\mathbb{P}_{\mathbf{xy}}\mathbb{P}_{\mathbf{l}}} \right) \right] = I_{KL}([\mathbf{x\,y}]; \mathbf{l}).$$

The integration measure on the right hand side simplifies to,

$$\mathbb{P}_{\mathbf{xy}}\mathbb{P}_{\mathbf{l}} \frac{\mathbb{P}_{\mathbf{yl|x}}}{\mathbb{P}_{\mathbf{y|x}}\mathbb{P}_{\mathbf{l|x}}} \frac{\mathbb{P}_{\mathbf{xl}}}{\mathbb{P}_{\mathbf{x}}\mathbb{P}_{\mathbf{l}}} = \frac{\mathbb{P}_{\mathbf{yl|x}}}{\mathbb{P}_{\mathbf{l|x}}} \mathbb{P}_{\mathbf{xl}} = \frac{\mathbb{P}_{\mathbf{yl|x}}}{\mathbb{P}_{\mathbf{l|x}}} \mathbb{P}_{\mathbf{l|x}} \mathbb{P}_{\mathbf{x}} = \mathbb{P}_{\mathbf{xyl}}.$$

Distributing over the logarithms, the right hand side is equal, $I_{KL}(\mathbf{y}; \mathbf{l} \mid \mathbf{x}) + I(\mathbf{x}; \mathbf{l})$. Similarly by flipping $\mathbf{x}$ and $\mathbf{y}$, $I_{KL}([\mathbf{x\,y}]; \mathbf{l}) = I_{KL}(\mathbf{x}; \mathbf{l} \mid \mathbf{y}) + I(\mathbf{y}; \mathbf{l})$. Hence, $I_{KL}(\mathbf{y}; \mathbf{l} \mid \mathbf{x}) + I_{KL}(\mathbf{x}; \mathbf{l}) = I_{KL}(\mathbf{x}; \mathbf{l} \mid \mathbf{y}) + I_{KL}(\mathbf{y}; \mathbf{l})$. $\qquad\square$

We can now prove the assertion that it is enough for use to consider unconditional and conditional sensitivities to states.

**Theorem 4.**

$$S(\mathbf{x}; \mathbf{l} \mid \mathbf{y}) + S(\mathbf{y} \mid \mathbf{l}) = S([\mathbf{x\,y}] \mid \mathbf{l}) = S([\mathbf{y\,x}]\mathbf{l}) = S(\mathbf{y}; \mathbf{l} \mid \mathbf{x}) + S(\mathbf{x} \mid \mathbf{l})$$

*Proof.* By corollary 7 , $S([\mathbf{x\,y}] \mid \mathbf{l}) = \mathbb{E}[KL\left(\mathbb{P}_{\mathbf{xy|l}} \parallel \mathbb{P}_{\mathbf{xy|l'}}\right)]$. Where $\mathbf{l'}$ is an independent copy of $\mathbf{l}$. By lemma 3 , $KL\left(\mathbb{P}_{\mathbf{xy|l}} \parallel \mathbb{P}_{\mathbf{xy|l'}}\right) = KL\left(\mathbb{P}_{\mathbf{y|x,l}} \parallel \mathbb{P}_{\mathbf{y|x,l'}} \mid \mathbb{P}_{\mathbf{x}}\right) + KL\left(\mathbb{P}_{\mathbf{x|l}} \parallel \mathbb{P}_{\mathbf{x|l'}}\right)$. Taking the expectation with respect to $(\mathbf{l}, \mathbf{l'})$, $S([\mathbf{x\,y}] \mid \mathbf{l}) = \mathbb{E}\left[KL\left(\mathbb{P}_{\mathbf{y|x,l}} \parallel \mathbb{P}_{\mathbf{y|x,l'}} \mid \mathbb{P}_{\mathbf{x}}\right)\right] + \mathbb{E}\left[KL\left(\mathbb{P}_{\mathbf{x|l}} \parallel \mathbb{P}_{\mathbf{x|l'}}\right)\right]$. Hence, $S([\mathbf{x\,y}] \mid \mathbf{l}) = S(\mathbf{y}; \mathbf{l} \mid \mathbf{x}) + S(\mathbf{x}; \mid \mathbf{l})$. Switching the roles of $\mathbf{x}$ and $\mathbf{y}$, by symmetry we establish the information Bayes theorem, $S(\mathbf{x}; \mathbf{l} \mid \mathbf{y}) + S(\mathbf{y}; \mid \mathbf{l}) = S([\mathbf{x\,y}] \mid \mathbf{l}) = S([\mathbf{y\,x}] \mid \mathbf{l}) = S(\mathbf{y}; \mathbf{l} \mid \mathbf{x}) + S(\mathbf{x}; \mid \mathbf{l})$. $\qquad\square$

**State Independent Bayes**

**Proposition 10** (State Independent Bayes). $\qquad$ (i) $S(\mathbf{y}; \mathbf{l}) = 0 \wedge S(\mathbf{x}; \mathbf{l} \mid \mathbf{y}) = 0 \implies S(\mathbf{y}; \mathbf{l} \mid \mathbf{x}) = 0$ *(Bayes)*.
$\qquad$ (ii) $S(\mathbf{y}; \mathbf{l}) = 0 \wedge S(\mathbf{x}; \mathbf{l} \mid \mathbf{y}) = 0 \implies S(\mathbf{x}; \mathbf{l}) = 0$ *(Marginalization)*

*Proof.* $S(\mathbf{y}; \mathbf{l} \mid \mathbf{x}) = 0 \wedge S(\mathbf{x}; \mathbf{l}) = 0 \implies S(\mathbf{y}; \mathbf{l}) = 0 \wedge S(\mathbf{x}; \mathbf{l} \mid \mathbf{y}) = 0$

$$S(\mathbf{x}; \mathbf{l}) + S(\mathbf{y}; \mathbf{l} \mid \mathbf{x}) = S(\mathbf{y}; \mathbf{l}) + S(\mathbf{x}; \mathbf{l} \mid \mathbf{y}),$$

$S(\mathbf{y}; \mathbf{l} \mid \mathbf{x}) = 0 \implies S(\mathbf{x}; \mathbf{l}) = S(\mathbf{y}; \mathbf{l}) + S(\mathbf{x}; \mathbf{l} \mid \mathbf{y})$. $S(\mathbf{x}; \mathbf{l}) = 0 \implies -S(\mathbf{x}; \mathbf{l} \mid \mathbf{y}) = S(\mathbf{y}; \mathbf{l})$. Which only holds if $S(\mathbf{x}; \mathbf{l} \mid \mathbf{y}) = S(\mathbf{y}; \mathbf{l}) = 0$.

Similarly, if $S(\mathbf{x}; \mathbf{l} \mid \mathbf{y}) = 0$, then,

$$S(\mathbf{y}; \mathbf{l} \mid \mathbf{x}) = -S(\mathbf{x}; \mathbf{l}) + S(\mathbf{y}; \mathbf{l})$$

If $S(\mathbf{y}; \mathbf{l}) = 0$ then we are done. $\qquad\square$

D.4    THEOREM 2

**Theorem 2** (Spectral dependence). $\frac{1}{2} \operatorname{Tr}\left[\mathbb{E}\left[\mathbb{E}\left[\boldsymbol{\Phi}_{\mathbf{l}}\right] : \boldsymbol{\Phi}_{\mathbf{l}}\right] - \mathbf{I}\right]$ *is equal to* $S(\mathbf{x}; \mathbf{l})$*, if* $\boldsymbol{\Phi}$ *is a canonical* $\boldsymbol{\Phi}$-*embedding of the unconditional kind, or* $S(\mathbf{y}; \mathbf{l} \mid \mathbf{x})$*, if* $\boldsymbol{\Phi}$ *is of the conditional kind.*

*Proof.* By corollary 8, $I_J(\mathbf{x};\mathbf{l}) = \frac{1}{2}\mathbb{E}\left[D_J\left(\mathbb{P}(\mathbf{x}\mid\mathbf{l}) \parallel \mathbb{P}(\mathbf{x}\mid\mathbf{l}')\right)\right]$. Now, $D_J\left(\mathbb{P}_{\mathbf{x}\mid\mathbf{l}} \parallel \mathbb{P}_{\mathbf{x}\mid\mathbf{l}'}\right) = KL\left(\mathbb{P}_{\mathbf{x}\mid\mathbf{l}} \parallel \mathbb{P}_{\mathbf{x}\mid\mathbf{l}'}\right) + KL\left(\mathbb{P}_{\mathbf{x}\mid\mathbf{l}'} \parallel \mathbb{P}_{\mathbf{x}\mid\mathbf{l}}\right)$. By proposition 12

$$KL\left(\gamma_{\mu_1,\Sigma_1} \parallel \gamma_{\mu_2,\Sigma_2}\right) = \frac{1}{2}\mathbf{D}\left(\mathbf{\Phi}(\Sigma_1,\mu_1) \parallel \mathbf{\Phi}(\Sigma_2,\mu_2)\right).$$

By definition 2 $\mathbf{D}\left(\mathbf{\Phi}(\Sigma_1,\mu_1) \parallel \mathbf{\Phi}(\Sigma_2,\mu_2)\right) = \mathbf{D}\left(\mathbf{\Phi}_\mathbf{l} \parallel \mathbf{\Phi}_{\mathbf{l}'}\right)$. Now, $\mathbf{D}\left(\mathbf{\Phi}_\mathbf{l} \parallel \mathbf{\Phi}_{\mathbf{l}'}\right) + \mathbf{D}\left(\mathbf{\Phi}_{\mathbf{l}'} \parallel \mathbf{\Phi}_\mathbf{l}\right) = \mathrm{Tr}[\mathbf{\Phi}_\mathbf{l} : \mathbf{\Phi}_{\mathbf{l}'} + (\mathbf{\Phi}_\mathbf{l} : \mathbf{\Phi}_{\mathbf{l}'})^{-1} - 2\mathbf{I}]$. By proposition 27, $\lambda(\mathbf{\Phi}_\mathbf{l} : \mathbf{\Phi}_{\mathbf{l}'}^{-1}) = \lambda(\mathbf{\Phi}_{\mathbf{l}'} : \mathbf{\Phi}_\mathbf{l})$. Hence,

$$\mathbf{D}\left(\mathbf{\Phi}_\mathbf{l} \parallel \mathbf{\Phi}_{\mathbf{l}'}\right) + \mathbf{D}\left(\mathbf{\Phi}_{\mathbf{l}'} \parallel \mathbf{\Phi}_\mathbf{l}\right) = \mathrm{Tr}[\mathbf{\Phi}_\mathbf{l} : \mathbf{\Phi}_{\mathbf{l}'} + \mathbf{\Phi}_{\mathbf{l}'} : \mathbf{\Phi}_\mathbf{l} - 2\mathbf{I}].$$

Thus $D_J\left(\mathbb{P}_{\mathbf{x}\mid\mathbf{l}} \parallel \mathbb{P}_{\mathbf{x}\mid\mathbf{l}'}\right) = \frac{1}{2}\mathrm{Tr}[\mathbf{\Phi}_\mathbf{l} : \mathbf{\Phi}_{\mathbf{l}'} + \mathbf{\Phi}_{\mathbf{l}'} : \mathbf{\Phi}_\mathbf{l} - 2\mathbf{I}]$. Hence, $I_J(\mathbf{x};\mathbf{l}) = \frac{1}{4}\mathbb{E}\left[\mathrm{Tr}[\mathbf{\Phi}_\mathbf{l} : \mathbf{\Phi}_{\mathbf{l}'} + \mathbf{\Phi}_{\mathbf{l}'} : \mathbf{\Phi}_\mathbf{l} - 2\mathbf{I}]\right]$. By linearity of he trace and the expectation, $I_J(\mathbf{x};\mathbf{l}) = \frac{1}{4}\mathrm{Tr}\left[\mathbb{E}\left[\mathbf{\Phi}_\mathbf{l} : \mathbf{\Phi}_{\mathbf{l}'}\right] + \mathbb{E}\left[\mathbf{\Phi}_{\mathbf{l}'} : \mathbf{\Phi}_\mathbf{l}\right] - 2\mathbf{I}\right]$. $\mathbf{l}$ and $\mathbf{l}'$ being independent. $\mathbb{E}[\mathbf{\Phi}_\mathbf{l} : \mathbf{\Phi}_{\mathbf{l}'}] = \mathbb{E}[\mathbf{\Phi}_{\mathbf{l}'} : \mathbf{\Phi}_\mathbf{l}]$. Hence, $I_J(\mathbf{x};\mathbf{l}) = \frac{1}{4}\mathrm{Tr}\left[2\mathbb{E}\left[\mathbf{\Phi}_\mathbf{l} : \mathbf{\Phi}_{\mathbf{l}'}\right] - 2\mathbf{I}\right]$ and $I_J(\mathbf{x};\mathbf{l}) = \frac{1}{2}\mathrm{Tr}\left[\mathbb{E}\left[\mathbf{\Phi}_\mathbf{l} : \mathbf{\Phi}_{\mathbf{l}'}\right] - \mathbf{I}\right]$. This proves the statement for $\mathbf{\Phi}$-embeddings of the unconditional kind. The statement for $\mathbf{\Phi}$-embeddings of the conditional kind follow similarly by using proposition 13. $\qquad\square$

### D.4.1 SPECTRAL DIVERGENCES

Relative entropies by looking at first differential of the logarithm of the size of a solution. The log determinant is concave by 28. We take the logarithm of the determinant to inject convavity in the scoring of perturbation along covariates and pairs of covariates,$\log\det[\mathbf{S}] = \mathrm{Tr}[\mathbf{Log\,S}]$. Taking the first differential, $\mathrm{d}(\log\det[\mathbf{S}];\mathbf{S}) = \mathrm{Tr}[\mathbf{S}^{-1}\,\mathrm{d}\mathbf{S}]$. Take $\mathrm{d}\mathbf{S} \in \mathcal{S}_\succ(d)$, then $\mathrm{Tr}[\mathbf{S}^{-1}\,\mathrm{d}\mathbf{S}] = \mathrm{Tr}[\mathrm{d}\mathbf{S}\mathbf{S}^{-1}] = \mathrm{Tr}[\mathrm{d}\mathbf{S} : \mathbf{S}]$. Thus we have, $\mathrm{d}(\log\det[\mathbf{S}];\mathrm{d}\mathbf{S}) = \mathrm{Tr}[\mathrm{d}\mathbf{S} : \mathbf{S}]$. Using the pairing, $\langle\mathbf{S}_1,\mathbf{S}_2\rangle_F = \mathrm{Tr}[\mathbf{S}_1\mathbf{S}_2]$, the convex conjugate of the function $\mathbf{S} \mapsto F(\mathbf{S}) := -\log\det[\mathbf{S}]$, $\mathbf{Y} \mapsto F^*(\mathbf{Y}) = -\log\det[\mathbf{I} - \mathbf{Y}]$. This allows establishing, conditions on the size of solution using Fenchel-Young type inequalities. Moreover, comparison of candidates solution is readily available through Bregman Divergences. The function $F$ is precisely the negative Burg entropy (Burg, 1972). Turning $\mathcal{Q}$ into a Bregman manifold.

**Log-det divergence**
**Proposition 11** (Burg's entropy Bregman's divergence). *Let* $\mathbf{X} \mapsto F(\mathbf{X}) = -\log\det[\mathbf{X}]$, *the $\mathcal{Q}$-negentropy generator, Let* $\mathbf{\Phi}_1,\mathbf{\Phi}_2 \in \mathcal{Q}(d_1,d_2)$. *Then,* $B_F(\mathbf{\Phi}_1,\mathbf{\Phi}_2) = \mathrm{Tr}[\mathbf{\Phi}_1 : \mathbf{\Phi}_2 - \mathbf{I}] - \log\det\left[\mathbf{\Phi}_1 : \mathbf{\Phi}_2\right]$.

*Proof.* By proposition 28 $F$ is stricly convex, proper and lower semicontinuous on $\mathcal{S}_\succeq(d)$. Now $\mathrm{d}F(\mathbf{\Phi}_1,\mathrm{d}\mathbf{\Phi}_1) = -\mathrm{Tr}[\mathbf{\Phi}_1^{-1}\,\mathrm{d}\mathbf{\Phi}_1] = -\left\langle\mathbf{\Phi}_1^{-1},\mathrm{d}\mathbf{\Phi}_1\right\rangle_F$. Hence $B_F(\mathbf{\Phi}_1,\mathbf{\Phi}_2) = -\log\det\left[\mathbf{\Phi}_1\right] + \log\det\left[\mathbf{\Phi}_2\right] + \left\langle\mathbf{\Phi}_1 - \mathbf{\Phi}_2, \mathbf{\Phi}_2^{-1}\right\rangle_F, = \log\det\left[\mathbf{\Phi}_2\mathbf{\Phi}_1^{-1}\right] + \mathrm{Tr}[(\mathbf{\Phi}_1 - \mathbf{\Phi}_2)\mathbf{\Phi}_2^{-1}] = \mathrm{Tr}[\mathbf{\Phi}_2^{-1}\mathbf{\Phi}_1 - \mathbf{I}] - \log\det\left[\mathbf{\Phi}_2^{-1}\mathbf{\Phi}_1\right]$. By proposition 26 $\lambda(\mathbf{\Phi}_2^{-1}\mathbf{\Phi}_1) = \lambda(\mathbf{\Phi}_1 : \mathbf{\Phi}_2)$. By proposition 6 $B_F(\mathbf{\Phi}_1,\mathbf{\Phi}_2) = \mathrm{Tr}[\mathbf{\Phi}_1 : \mathbf{\Phi}_2 - \mathbf{I}] - \log\det\left[\mathbf{\Phi}_1 : \mathbf{\Phi}_2\right]$. $\qquad\square$

*Remark* 1. The $\mathcal{Q}$-negentropy Bregman's divergence betwenn $\mathbf{\Phi}_1$ and $\mathbf{\Phi}_2$ is the log-det divergence (Cichocki et al., 2015), $\mathbf{D}\left(\mathbf{\Phi}_1 \parallel \mathbf{P}_2\right)$.

This lead to the following spectral expressions,
**Corollary 13.** *Let* $\mathbf{\Lambda}$ *be the eigenvalues matrix of* $\mathbf{\Phi}_1 : \mathbf{\Phi}_2$. *Then,*

$$\mathbf{D}\left(\mathbf{\Phi}_1 \parallel \mathbf{\Phi}_2\right) = \mathrm{Tr}[\mathbf{\Lambda} - \mathbf{I}] - \log\det[\mathbf{\Lambda}] = \mathrm{Tr}[\mathbf{\Lambda} - \mathbf{Log\,\Lambda} - \mathbf{I}].$$

**Correspondance between log-det divergence Kullblack-Leibler divergence**   The next proposition shows that the KL-divergence of the distributions is precisely the log-det divergence of a $\mathbf{\Phi}$-embedding.
**Proposition 12.**

$$KL\left(\gamma_{\mu_1,\Sigma_1} \parallel \gamma_{\mu_2,\Sigma_2}\right) = \frac{1}{2}\mathbf{D}\left(\mathbf{\Phi}(\Sigma_1,\mu_1) \parallel \mathbf{\Phi}(\Sigma_2,\mu_2)\right).$$

*Proof.* By proposition 11 $\mathbf{D}\left(\Phi(\Sigma_1, \mu_1) \| \Phi(\Sigma_2, \mu_2)\right) = \mathrm{Tr}[\Phi_1 : \Phi_2 - \mathbf{I}] - \log \det[\Phi_1 : \Phi_2]$. By proposition 6, $\det[\Phi_1 : \Phi_2] = \det[\Sigma_2^{-1}\Sigma_1]$ and $\mathrm{Tr}[\Phi_1 : \Phi_2] = \mathrm{Tr}[\Sigma_2^{-1}\Sigma_1] + \mathrm{Tr}[(\mu_2 - \mu_1)^\top \Sigma_2^{-1}(\mu_2 - \mu_1)] + 1$. Hence $\mathbf{D}\left(\Phi_1 \| \Phi_2\right) = \log\left(\frac{\det[\Sigma_2]}{\det[\Sigma_1]}\right) + \mathrm{Tr}\left[\Sigma_2^{-1}\Sigma_1\right] - m + \mathrm{Tr}\left[(\mu_2 - \mu_1)^\top \Sigma_2^{-1}(\mu_2 - \mu_1)\right] = 2KL\left(\mathcal{N}(\mu_1, \Sigma_1) \| \mathcal{N}(\mu_2, \Sigma_2)\right)$. $\qquad\square$

**Symmetrized log-det divergence.** The log-det divergence is not symmetric, $\mathbf{D}\left(\Phi_1 \| \Phi_2\right) \neq \mathbf{D}\left(\Phi_2 \| \Phi_1\right)$. However we can symmetrize it, by defining,

**Definition 7** (J-divergence)**.**

$$\mathbf{D}_J\left(\Phi_1 \| \Phi_2\right) = \mathbf{D}\left(\Phi_1 \| \Phi_2\right) + \mathbf{D}\left(\Phi_2 \| \Phi_1\right)$$

**Proposition 13.**

$$\mathbf{D}_J\left(\Phi_1 \| \Phi_2\right) = \frac{1}{2}\mathrm{Tr}[\Phi_1 : \Phi_2 + (\Phi_1 : \Phi_2)^{-1} - 2\mathbf{I}].$$

*Proof.* Follows from definition 7, corollary 13 and proposition 26. $\qquad\square$

Note that from Corollary 13 we get a spectral expression in terms of the eigenvalues $\Phi_1 : \Phi_2$.

**Corollary 14** (J-Divergence spectral form)**.**

$$D_J\left(\Phi_1 \| \Phi_2\right) = \frac{1}{2}\mathrm{Tr}[\Lambda + \Lambda^{-1} - 2\mathbf{I}].$$

*Where $\Lambda$ is the eigenvalue matrix of $\Phi_1 : \Phi_2$.*

**Conditional Kullblack-Leibler divergence.** It is easier to understand in the conditional Kullblack-Leilber as functional of two markov kernels and a probability distributions. The next proposition shows that the conditional Kullblack-Leibler divergence can also be expressed in terms of the log-determinant divergence on $\mathcal{Q}$.

**Proposition 14.** *For $i \in \{1, 2\}$, Let $\mathbf{K}_i$ a Gaussian Markov kernel with source $\mathcal{X}$, destination $\mathscr{P}(\mathcal{Y})$, and $(\forall \mathbf{x} \in \mathcal{X})\ \mathbf{K}_i(\mathbf{x}) \sim \gamma_{\mu_i(\mathbf{x}), \Gamma_i}$, where $\mu_i(\mathbf{x}) := \mathbf{A}_i\mathbf{x} + \mathbf{b}_i$. Let $\Phi$ be the $\Phi$-embedding of some non-degenerate Gaussian measure $\gamma_{\mu, \Sigma}$. Define,*

$$\Delta := \begin{bmatrix} \mathbf{A}_2 - \mathbf{A}_1 & \mathbf{b}_2 - \mathbf{b}_1 \end{bmatrix} \Phi \begin{bmatrix} (\mathbf{A}_2 - \mathbf{A}_1)^\top \\ (\mathbf{b}_2 - \mathbf{b}_1)^\top \end{bmatrix}.$$

*Then,*

$$KL\left(\mathbf{K}_1 \| \mathbf{K}_2 \mid \gamma_{\mu, \Sigma}\right) = \frac{1}{2}\mathbf{D}\left(\Phi(\Gamma_1, \Delta) \| \Phi(\Gamma_2, \mathbf{0}_{d_y \times (d_x+1)})\right).$$

*Proof.* The conditional Kullblack-Leilber divergence of $\mathbf{K}_1$ and $\mathbf{K}_2$ given $\gamma_{\mu, \Sigma}$ is given by,

$$KL\left(\mathbf{K}_1 \| \mathbf{K}_2 \mid \gamma_{\mu, \Sigma}\right) = \mathbb{E}\left[KL\left(\mathbf{K}_1(\cdot, \mathbf{x}) \| \mathbf{K}_2(\cdot, \mathbf{x})\right)\right],$$

Where $\mathbf{x} \sim \gamma_{\mu, \Sigma}$. Let $\Phi_i : \mathcal{X} \to \mathcal{Q}(d_y + 1)$, $\mathbf{x} \mapsto \Phi_i(\mathbf{x}) = \mathbb{E}[q(\mathbf{y}) \mid \mathbf{x}]$. By proposition 12,

$$(\forall \mathbf{x} \in \mathcal{X})\, KL\left(\mathbf{K}_1(\cdot, \mathbf{x}) \| \mathbf{K}_2(\cdot, \mathbf{x})\right) = \frac{1}{2}\mathrm{Tr}\left[\Phi_1(\mathbf{x}) : \Phi_2(\mathbf{x}) - (\mathbf{I}_{d_y} + 1)\right] - \log \det\left[\Phi_1(\mathbf{x}) : \Phi_2(\mathbf{x})\right].$$

Now, proposition 6,

$$\mathrm{Tr}[\Phi_1(\mathbf{x}) : \Phi_2(\mathbf{x})] = \mathrm{Tr}[\Gamma_2^{-1}\Gamma_1] + \mathrm{Tr}[(\mu_2(\mathbf{x}) - \mu_1(\mathbf{x}))^\top \Gamma_2^{-1}(\mu_2(\mathbf{x}) - \mu_1(\mathbf{x}))] + 1.$$

By linearity the expectation and ciclycity of the trace,

$$\mathbb{E}\left[\mathrm{Tr}\left[\Phi_1(\mathbf{x}) : \Phi_2(\mathbf{x})\right]\right] = \mathrm{Tr}[\Gamma_2^{-1}\Gamma_1] + \mathrm{Tr}\left[\Gamma_2^{-1}\mathbb{E}\left[(\mu_2(\mathbf{x}) - \mu_1(\mathbf{x}))(\mu_2(\mathbf{x}) - \mu_1(\mathbf{x}))^\top\right]\right] + 1. \quad (4)$$

Writing $\mathbf{A} := \mathbf{A}_2 - \mathbf{A}_1$ and $\mathbf{b} := \mathbf{b}_2 - \mathbf{b}_1$. Expanding the quadratic product yields,

$$(\boldsymbol{\mu}_2(\mathbf{x}) - \boldsymbol{\mu}_1(\mathbf{x}))(\boldsymbol{\mu}_2(\mathbf{x}) - \boldsymbol{\mu}_1(\mathbf{x}))^\top = (\mathbf{A}\mathbf{x} + \mathbf{b})(\mathbf{A}\mathbf{x} + \mathbf{b})^\top,$$
$$= \mathbf{b}\mathbf{b}^\top + \mathbf{A}\mathbf{x}\mathbf{b}^\top + \mathbf{b}\mathbf{x}^\top\mathbf{A}^\top + \mathbf{A}\mathbf{x}\mathbf{x}^\top\mathbf{A}^\top.$$

Taking the expectation we have,

$$\mathbb{E}[(\boldsymbol{\mu}_2(\mathbf{x}) - \boldsymbol{\mu}_1(\mathbf{x}))(\boldsymbol{\mu}_2(\mathbf{x}) - \boldsymbol{\mu}_1(\mathbf{x}))^\top] = \mathbf{b}\mathbf{b}^\top + \mathbf{A}\boldsymbol{\mu}\mathbf{A}^\top + \mathbf{A}\boldsymbol{\mu}\mathbf{b}^\top + \mathbf{A}\boldsymbol{\Sigma}\mathbf{A}^\top + \mathbf{b}\boldsymbol{\mu}^\top\mathbf{A}^\top,$$
$$= \begin{bmatrix} \mathbf{A} & \mathbf{b} \end{bmatrix} \begin{bmatrix} \boldsymbol{\Sigma} + \boldsymbol{\mu}\boldsymbol{\mu}^\top & \boldsymbol{\mu} \\ \boldsymbol{\mu}^\top & 1 \end{bmatrix} \begin{bmatrix} \mathbf{A}^\top \\ \mathbf{b}^\top \end{bmatrix},$$
$$= \begin{bmatrix} \mathbf{A} & \mathbf{b} \end{bmatrix} \boldsymbol{\Phi}(\boldsymbol{\Sigma}, \boldsymbol{\mu}) \begin{bmatrix} \mathbf{A}^\top \\ \mathbf{b}^\top \end{bmatrix}.$$

Hence,

$$2KL\left(\mathbf{K}_1 \parallel \mathbf{K}_2 \mid \gamma_{\boldsymbol{\mu},\boldsymbol{\Sigma}}\right) = \mathrm{Tr}[\boldsymbol{\Gamma}_2^{-1}\boldsymbol{\Gamma}_1 - \mathbf{I}_{d_y}] - \log\det\left[\boldsymbol{\Gamma}_2^{-1}\boldsymbol{\Gamma}_1\right] + \mathrm{Tr}\left[\boldsymbol{\Gamma}_2^{-1} \begin{bmatrix} \mathbf{A} & \mathbf{b} \end{bmatrix} \boldsymbol{\Phi}(\boldsymbol{\Sigma}, \boldsymbol{\mu}) \begin{bmatrix} \mathbf{A}^\top \\ \mathbf{b}^\top \end{bmatrix}\right].$$

Writing, $\boldsymbol{\Delta} := \begin{bmatrix} \mathbf{A} & \mathbf{b} \end{bmatrix} \boldsymbol{\Phi}(\boldsymbol{\Sigma}, \boldsymbol{\mu}) \begin{bmatrix} \mathbf{A}^\top \\ \mathbf{b}^\top \end{bmatrix}$, $\mathbf{M}_1 = \boldsymbol{\Phi}(\boldsymbol{\Gamma}_1, \boldsymbol{\Delta}), \mathbf{M}_2 = \boldsymbol{\Phi}(\boldsymbol{\Gamma}_2, \mathbf{0}_{d_y \times (d_x+1)})$ By proposition 6,

$$\mathrm{Tr}\left[\boldsymbol{\Gamma}_2^{-1}\boldsymbol{\Delta}\right] = \mathrm{Tr}[\mathbf{M}_1 : \mathbf{M}_2] - \mathrm{Tr}[\boldsymbol{\Gamma}_2^{-1}\boldsymbol{\Gamma}_1] - (d_x + 1).$$

Moreover, by proposition 6, $\det\left[\boldsymbol{\Gamma}_2^{-1}\boldsymbol{\Gamma}_1\right] = \det\left[\mathbf{M}_1 : \mathbf{M}_2\right]$ Thus we conclude,

$$KL\left(\mathbf{K}_1 \parallel \mathbf{K}_2 \mid \gamma_{\boldsymbol{\mu},\boldsymbol{\Sigma}}\right) = \frac{1}{2}\mathbf{D}\left(\boldsymbol{\Phi}(\boldsymbol{\Gamma}_1, \boldsymbol{\Delta}) \parallel \boldsymbol{\Phi}(\boldsymbol{\Gamma}_2, \mathbf{0}_{d_y \times (d_x+1)})\right).$$

$\square$

### D.5 COROLLARY 1

**Corollary 1.** *Let $\boldsymbol{\Phi}_\mathbf{l}$ be a state index canonical $\boldsymbol{\Phi}$-embedding. Then,*

$$\mathbb{R}^{m+n} = \bigoplus_{\lambda \in \lambda(\mathbb{E}[\mathbb{E}[\boldsymbol{\Phi}_\mathbf{l}] : \boldsymbol{\Phi}_\mathbf{l}])} \mathrm{Eig}\left(\mathbb{E}\left[\mathbb{E}[\boldsymbol{\Phi}_\mathbf{l}] : \boldsymbol{\Phi}_\mathbf{l}\right], \lambda\right)$$

$$\min_{\mathbf{W}^\top\mathbf{W}=\mathbf{I}_k} \mathrm{Tr}\left[\mathbf{W}^\top \mathbb{E}\left[\mathbb{E}\left[\boldsymbol{\Phi}_\mathbf{l}\right] : \boldsymbol{\Phi}_\mathbf{l}\right]\mathbf{W}\right] = \sum_{i=1}^{k} \lambda_i^\uparrow\left(\mathbb{E}\left[\mathbb{E}\left[\boldsymbol{\Phi}_\mathbf{l}\right] : \boldsymbol{\Phi}_\mathbf{l}\right]\right).$$

*Proof.* Direct application of the Ky-Fan trace minimization principle (Fan, 1949). $\square$

### D.6 THEOREM 3

#### D.6.1 REDUCING STABILITY OF EIGENSPACES TO A MATRIX EQUATIONS

How to find the condition under which a simple invariant space of of hermitian matrix is stable? Consider a symmetric matrix in block form,

$$\mathbf{S} = \begin{bmatrix} \mathbf{S}_{11} & \mathbf{S}_{12} \\ \mathbf{S}_{12}^\top & \mathbf{S}_{22} \end{bmatrix}.$$

Let $\mathcal{X}$ be a simple invariant subspace of $\mathbf{S}$. Write $\mathbf{X}_1$ for the matrix whose columns form an orthonomal basis fo $\mathcal{X}$ and $\mathbf{Y}_2$ the matrix whose columns form an orthonomal basis for $\mathcal{Y}$. The matrix $\mathbf{W} = \begin{bmatrix} \mathbf{X}_1 & \mathbf{Y}_2 \end{bmatrix}$. $\mathbf{W}$ is unitary. Indeed,

$$\mathbf{W}^\top\mathbf{W} = \begin{bmatrix} \mathbf{X}_1^\top \\ \mathbf{Y}_2^\top \end{bmatrix} \begin{bmatrix} \mathbf{X}_1 & \mathbf{Y}_2 \end{bmatrix} = \begin{bmatrix} \mathbf{X}_1^\top\mathbf{X}_1 & \mathbf{X}_1^\top\mathbf{Y}_2 \\ \mathbf{Y}_2^\top\mathbf{X}_1 & \mathbf{Y}_2^\top\mathbf{Y}_2 \end{bmatrix} = \begin{bmatrix} \mathbf{I} & \mathbf{0} \\ \mathbf{0} & \mathbf{I} \end{bmatrix}.$$

Consider the *spectral resolution* of $\mathbf{S}$,

$$\mathbf{W}^\top\mathbf{S}\mathbf{W} = \begin{bmatrix} \mathbf{X}_1^\top\mathbf{S}\mathbf{X}_1 & \mathbf{X}_1^\top\mathbf{S}\mathbf{Y}_2 \\ \mathbf{Y}_2^\top\mathbf{S}\mathbf{X}_1 & \mathbf{Y}_2^\top\mathbf{S}\mathbf{Y}_2 \end{bmatrix}.$$

There exists diagonal matrices $\mathbf{\Lambda}_{11}$ and $\mathbf{\Lambda}_{22}$ such that $\mathbf{SX}_1 = \mathbf{X}_1\mathbf{\Lambda}_{11}$ and $\mathbf{SY}_2 = \mathbf{Y}_2\mathbf{\Lambda}_{22}$. We therefore have, $\mathbf{Y}_2^\top \mathbf{SX}_1 = \mathbf{Y}_2^\top \mathbf{X}_1\mathbf{\Lambda}_{11} = \mathbf{0}$, and hence,

$$\mathbf{W}^\top \mathbf{SW} = \begin{bmatrix} \mathbf{\Lambda}_{11} & \mathbf{0} \\ \mathbf{0} & \mathbf{\Lambda}_{22} \end{bmatrix}.$$

Now consider a perturbation of $\tilde{\mathbf{S}} = \mathbf{S} + \mathbf{E}$. Partition $\mathbf{E}$ conformably, and write,

$$\mathbf{W}^\top \mathbf{EW} = \begin{bmatrix} \mathbf{E}_{11} & \mathbf{E}_{12} \\ \mathbf{E}_{21} & \mathbf{E}_{22} \end{bmatrix}.$$

we want to find conditions under which the invariant space $\mathcal{X}$ of $\mathbf{S}$ is also an invariant space of $\tilde{\mathbf{S}}$. Form,

$$\hat{\mathbf{X}}_1 = (\mathbf{X}_1 + \mathbf{Y}_2\mathbf{P})(\mathbf{I} + \mathbf{P}^\top\mathbf{P})^{-\frac{1}{2}},$$

$$\hat{\mathbf{Y}}_2 = (\mathbf{Y}_2 - \mathbf{X}_1\mathbf{P}^\top)(\mathbf{I} + \mathbf{P}\mathbf{P}^\top)^{-\frac{1}{2}}.$$

Write, $\hat{\mathbf{W}} = \begin{bmatrix} \hat{\mathbf{X}}_1 & \hat{\mathbf{Y}}_2 \end{bmatrix}$. Notice that $\hat{\mathbf{W}}$ is unitary. Indeed,

$$\hat{\mathbf{W}}^\top\hat{\mathbf{W}} = \begin{bmatrix} (\mathbf{I} + \mathbf{P}^\top\mathbf{P})^{-1} + \sqrt{(\mathbf{I} + \mathbf{P}^\top\mathbf{P})^{-1}}\mathbf{P}^\top\mathbf{P}\sqrt{(\mathbf{I} + \mathbf{P}^\top\mathbf{P})^{-1}} & \mathbf{0} \\ \mathbf{0} & (\mathbf{I} + \mathbf{P}\mathbf{P}^\top)^{-1} + \sqrt{(\mathbf{I} + \mathbf{P}\mathbf{P}^\top)^{-1}}\mathbf{P}\mathbf{P}^\top\sqrt{(\mathbf{I} + \mathbf{P}\mathbf{P}^\top)^{-1}} \end{bmatrix}$$

$$= \begin{bmatrix} \sqrt{(\mathbf{I} + \mathbf{P}^\top\mathbf{P})^{-1}}(\mathbf{I} + \mathbf{P}^\top\mathbf{P})\sqrt{(\mathbf{I} + \mathbf{P}\mathbf{P}^\top)^{-1}} & \mathbf{0} \\ \mathbf{0} & \sqrt{(\mathbf{I} + \mathbf{P}\mathbf{P}^\top)^{-1}}(\mathbf{I} + \mathbf{P}\mathbf{P}^\top)\sqrt{(\mathbf{I} + \mathbf{P}\mathbf{P}^\top)^{-1}} \end{bmatrix},$$

$$= \begin{bmatrix} \mathbf{I} & \mathbf{0} \\ \mathbf{0} & \mathbf{I} \end{bmatrix}.$$

Forming the resolution of $\tilde{\mathbf{S}}$ with respect to $\hat{\mathbf{W}}$, the lower left block is equal to zero if and only if,

$$\mathbf{0} = -\left( \sqrt{(\mathbf{I} + \mathbf{P}\mathbf{P}^\top)^{-1}} \left( \mathbf{P}\mathbf{\Lambda}_{11} + \mathbf{P}\mathbf{X}_1^\top\mathbf{E}\mathbf{X}_1 + \mathbf{P}\mathbf{X}_1^\top\mathbf{E}\mathbf{Y}_2\mathbf{P} - \mathbf{Y}_2^\top\mathbf{E}\mathbf{Y}_2\mathbf{P} - \mathbf{Y}_2^\top\mathbf{E}\mathbf{X}_1 - \mathbf{\Lambda}_{22}\mathbf{P} \right) \sqrt{(\mathbf{I} + \mathbf{P}^\top\mathbf{P})^{-1}} \right), \iff$$

$$\mathbf{0} = \mathbf{P}\mathbf{\Lambda}_{11} + \mathbf{P}\mathbf{X}_1^\top\mathbf{E}\mathbf{X}_1 + \mathbf{P}\mathbf{X}_1^\top\mathbf{E}\mathbf{Y}_2\mathbf{P} - \mathbf{Y}_2^\top\mathbf{E}\mathbf{Y}_2\mathbf{P} - \mathbf{Y}_2^\top\mathbf{E}\mathbf{X}_1 - \mathbf{\Lambda}_{22}\mathbf{0}.$$

$$(\mathbf{\Lambda}_{11} + \mathbf{X}_1^\top\mathbf{E}\mathbf{X}_1)\mathbf{P} - \mathbf{P}(\mathbf{\Lambda}_{22} + \mathbf{Y}_2^\top\mathbf{E}\mathbf{Y}_2)\mathbf{P} = \mathbf{Y}_2\mathbf{E}\mathbf{X}_1 - \mathbf{P}\mathbf{X}_1^\top\mathbf{E}\mathbf{Y}_2\mathbf{P}. \tag{5}$$

Let $\mathbf{L}_1 = \mathbf{\Lambda}_{11} + \mathbf{E}_{11}$, $\mathbf{L}_2 = (\mathbf{\Lambda}_{22} + \mathbf{E}_{22})$, $\mathbf{H} := \mathbf{E}_{12} = \mathbf{X}_1^\top\mathbf{E}\mathbf{Y}_2$, and $\mathbf{G} := \mathbf{E}_{21} = \mathbf{Y}_2^\top\mathbf{E}\mathbf{X}_1$. We can rewrite equation (5) in terms of the Sylvester operator,

$$\mathbf{T} : \mathbf{P} \mapsto \mathbf{PL}_1 - \mathbf{L}_2\mathbf{P}, \tag{6}$$

yielding,

$$\mathbf{T}(\mathbf{P}) = \mathbf{G} - \mathbf{PHP}. \tag{7}$$

D.6.2   EXISTENCE OF A SOLUTION

**Proposition 15.** *The equation* (7) *admits a solution if,*

$$4\,\|\mathbf{G}\|\,\|\mathbf{H}\|\,\left\|\mathbf{T}^{-1}\right\|^2 < 1. \tag{8}$$

*Proof.* Formally, Inverting $\mathbf{T}$ in equation (7),

$$\mathbf{P} = \mathbf{T}^{-1}(\mathbf{G} - \mathbf{PHP}), \tag{9}$$

suggests that the solution to equation (7) should be a fixed point of equation (9). Form the sequence,

$$\begin{cases} \mathbf{P}_0 & = \mathbf{0}, \\ \mathbf{P}_{k+1} & = \mathbf{T}^{-1}(\mathbf{G} - \mathbf{P}_k\mathbf{H}\mathbf{P}_k), \quad \forall k \in \mathbb{N} \bigcap [0, \infty). \end{cases} \tag{10}$$

Consider set space of all matrices $\mathcal{M}_{m\times n}(\mathbb{R})$, endowed a metric derived from a unitarily invariant norm,

$$\mathcal{M}_{m\times n}(\mathbb{R}) \times \mathcal{M}_{m\times n}(\mathbb{R}) \to [0, \infty),$$

$$(\mathbf{A}, \mathbf{B}) \mapsto d(\mathbf{A}, \mathbf{B}) := \|\mathbf{A} - \mathbf{B}\|.$$

We need to establish when the sequence defined in equation (10):

1. Cauchy.
2. Converges to the fixed defined in equation (9).

First, we show that the sequence is bounded.

$$\left\|\mathbf{P}_{k+1}\right\| \le \left\|\mathbf{T}^{-1}\right\| \left\|\mathbf{G} - \mathbf{P}_k \mathbf{H} \mathbf{P}_k\right\|,$$
$$\le \left\|\mathbf{T}^{-1}\right\| \left(\|\mathbf{G}\| + \left\|\mathbf{P}_k \mathbf{H} \mathbf{P}_k\right\|\right),$$
$$\le \left\|\mathbf{T}^{-1}\right\| \left(\|\mathbf{G}\| + \|\mathbf{H}\| \left\|\mathbf{P}_k\right\|^2\right),$$
$$= \left\|\mathbf{T}^{-1}\right\| \|\mathbf{G}\| + \left\|\mathbf{T}^{-1}\right\| \|\mathbf{H}\| \left\|\mathbf{P}_k\right\|^2$$

Let $x_{k+1} = \left\|\mathbf{T}^{-1}\right\| \|\mathbf{G}\| + \left\|\mathbf{T}^{-1}\right\| \|\mathbf{H}\| \, x_k^2$ for $k \in \mathbb{N} \bigcap [0, \infty)$ and $x_0 = 0$. Then,

$$\forall k \in \mathbb{N} \bigcap [0, \infty), x_{k+1} \ge x_k.$$

Any accumulation point of $x_k$ must verify,

$$x = \left\|\mathbf{T}^{-1}\right\| \|\mathbf{G}\| + \left\|\mathbf{T}^{-1}\right\| \|\mathbf{H}\| \, x^2.$$

Formally, The roots of this quadratic polynomial is $x$ are given by,

$$x_{\pm} = \frac{1 \pm \sqrt{1 - 4 \|\mathbf{G}\| \|\mathbf{H}\| \left\|\mathbf{T}^{-1}\right\|^2}}{2 \|\mathbf{H}\| \left\|\mathbf{T}^{-1}\right\|}.$$

For the roots to be real we need to require,

$$\mathbf{\Delta} := 1 - 4 \|\mathbf{G}\| \|\mathbf{H}\| \left\|\mathbf{T}^{-1}\right\|^2 \ge 0 \tag{11}$$

Since we are interested in a least upper bound, we will only consider the accumulation point $\mathbf{x}_-$. Hence, we have,

$$\|\mathbf{P}\|_{k+1} \le x_- = \frac{1 - \sqrt{1 - 4 \|\mathbf{G}\| \|\mathbf{H}\| \left\|\mathbf{T}^{-1}\right\|^2}}{2 \|\mathbf{H}\| \left\|\mathbf{T}^{-1}\right\|}. \tag{12}$$

And hence, assuming $\mathbf{\Delta} > 0$, then,

$$\|\mathbf{P}\| \le 2 \|\mathbf{G}\| \left\|\mathbf{T}^{-1}\right\|. \tag{13}$$

The sequence in equation (10) is bounded and increasing.

$$\left\|\mathbf{P}_{k+1} - \mathbf{P}_k\right\| = \left\|\mathbf{T}^{-1}(\mathbf{G} - \mathbf{P}_k \mathbf{H} \mathbf{P}_k) - \mathbf{P}_k\right\|,$$
$$= \left\|\mathbf{T}^{-1}(\mathbf{G} - \mathbf{P}_k \mathbf{H} \mathbf{P}_k) - \left(\mathbf{T}^{-1}(\mathbf{G} - \mathbf{P}_{k-1} \mathbf{H} \mathbf{P}_{k-1})\right)\right\|,$$
$$\le \left\|\mathbf{T}^{-1}\right\| \left\|\mathbf{P}_k \mathbf{H} \mathbf{P}_k - \mathbf{P}_{k-1} \mathbf{H} \mathbf{P}_{k-1}\right\|.$$

Now,

$$\mathbf{P}_k \mathbf{H} \mathbf{P}_k - \mathbf{P}_{k-1} \mathbf{H} \mathbf{P}_{k-1} = \frac{1}{2} \left\{ \left(\mathbf{P}_k - \mathbf{P}_{k-1}\right) \mathbf{H} \left(\mathbf{P}_k + \mathbf{P}_{k-1}\right) \right\} + \frac{1}{2} \left\{ \left(\mathbf{P}_k + \mathbf{P}_{k-1}\right) \mathbf{H} \left(\mathbf{P}_k - \mathbf{P}_{k-1}\right) \right\}.$$

Taking the norm,

$$\left\|\mathbf{P}_k \mathbf{H} \mathbf{P}_k - \mathbf{P}_{k-1} \mathbf{H} \mathbf{P}_{k-1}\right\| = \left\|\tfrac{1}{2} \left(\left(\mathbf{P}_k - \mathbf{P}_{k-1}\right) \mathbf{H} \left(\mathbf{P}_k + \mathbf{P}_{k-1}\right)\right) + \tfrac{1}{2} \left(\left(\mathbf{P}_k + \mathbf{P}_{k-1}\right) \mathbf{H} \left(\mathbf{P}_k - \mathbf{P}_{k-1}\right)\right)\right\|,$$
$$\le \tfrac{1}{2} \left\{ \left\|\left(\mathbf{P}_k - \mathbf{P}_{k-1}\right) \mathbf{H} \left(\mathbf{P}_k + +\mathbf{P}_{k-1}\right)\right\| + \left\|\left(\mathbf{P}_k + \mathbf{P}_{k-1}\right) \mathbf{H} \left(\mathbf{P}_k - \mathbf{P}_{k-1}\right)\right\| \right\},$$
$$\le \tfrac{1}{2} \left\{ \left\|\mathbf{P}_k - \mathbf{P}_{k-1}\right\| \|\mathbf{H}\| \left\|\mathbf{P}_k + \mathbf{P}_{k-1}\right\| + \left\|\mathbf{P}_k - \mathbf{P}_{k-1}\right\| \|\mathbf{H}\| \left\|\mathbf{P}_k + \mathbf{P}_{k-1}\right\| \right\},$$
$$= \|\mathbf{H}\| \left\|\mathbf{P}_k + \mathbf{P}_{k-1}\right\| \left\|\mathbf{P}_k - \mathbf{P}_{k-1}\right\|,$$
$$\le 2 \max \left(\left\|\mathbf{P}_k\right\|, \left\|\mathbf{P}_{k-1}\right\|\right) \|\mathbf{H}\| \left\|\mathbf{P}_k - \mathbf{P}_{k-1}\right\|.$$

Substituting, we have,

$$\left\|\mathbf{P}_{k+1} - \mathbf{P}_k\right\| \le 2 \left\|\mathbf{T}^{-1}\right\| \max \left(\left\|\mathbf{P}_k\right\|, \left\|\mathbf{P}_{k-1}\right\|\right) \|\mathbf{H}\| \left\|\mathbf{P}_k - \mathbf{P}_{k-1}\right\|,$$
$$\le \left(2 \left\|\mathbf{T}^{-1}\right\| \|\mathbf{H}\| \frac{1 - \sqrt{\mathbf{\Delta}}}{2 \|\mathbf{H}\| \left\|\mathbf{T}^{-1}\right\|}\right) \left\|\mathbf{P}_k - \mathbf{P}_{k-1}\right\|,$$
$$= \left(1 - \sqrt{\mathbf{\Delta}}\right) \left\|\mathbf{P}_k - \mathbf{P}_{k-1}\right\|.$$

Unpacking the recurrence,

$$\left\| \mathbf{P}_{k+1} - \mathbf{P}_k \right\| \leq \left( 1 - \sqrt{\Delta} \right)^k \left\| \mathbf{P}_1 - \mathbf{P}_0 \right\|.$$

Thus as long as $\left( 1 - \sqrt{\Delta} \right) < 1$ which holds if and only if,

$$4 \left\| \mathbf{G} \right\| \left\| \mathbf{H} \right\| \left\| \mathbf{T}^{-1} \right\|^2 < 1, \tag{14}$$

the sequence in equation (10) is Cauchy. $\square$

### D.6.3   Theorem 3

**Theorem 3** (Conditions for faithful processing). *Let $\mathscr{E}_k$ be the rank-k subspace of minimal dependence of $\underline{\mathbf{\Phi}}$. Let $\overline{\mathbf{\Phi}} = \underline{\mathbf{\Phi}} + \mathbf{E}$. If $\left\| \mathbf{E} \right\|_F < \frac{1}{2} \left( \lambda_{k+1}^{\uparrow}(\mathbf{\Phi}) - \lambda_k^{\uparrow}(\mathbf{\Phi}) \right)$, then $\mathscr{E}_k$ is an algebraically invariant subspace of $\overline{\mathbf{\Phi}}$.*

*Proof.* Now that we have establish the conditions under which the matrix equation (7) we need to refine them. First let's express $\left\| \mathbf{T}^{-1} \right\|$. We re-express the sylvester operator. For any, $\mathbf{P}$ we have,

$$\mathbf{T}(\mathbf{P}) = \mathbf{P}\mathbf{L}_1 - \mathbf{L}_2\mathbf{P}, \iff$$
$$\mathbf{vec}(\mathbf{T}(\mathbf{P})) = \mathbf{vec}\left( \mathbf{P}\mathbf{L}_1 - \mathbf{L}_2\mathbf{P} \right),$$
$$= \left( \mathbf{I} \otimes \mathbf{L}_1 - \mathbf{L}_2 \otimes \mathbf{I} \right) \mathbf{vec}(\mathbf{P}).$$

And hence, by propostion todo,

$$\lambda_{ij}(\mathbf{I} \otimes \mathbf{L}_1 - \mathbf{L}_2^\top \otimes \mathbf{I}) = \lambda_i(\mathbf{L}_1) - \lambda_j(\mathbf{L}_2).$$

Indeed, by the spectral mapping theorem, (Kato, 1995), for any unitarily invariant norm, $\mathbf{X} \mapsto \left\| \mathbf{X} \right\|$ there exists symmetric Gauge function $g$ such that $\left\| \mathbf{X} \right\| = g(\sigma(\mathbf{X}))$.

$$\left\| \mathbf{T}^{-1} \right\| = g(\sigma(\mathbf{T}^{-1})),$$
$$= g(\sigma(\mathbf{T})^{-1}).$$

Thus if consider the spectral norm then,

$$\left\| \mathbf{T}^{-1} \right\| = \sigma_d(\mathbf{T}),$$

the seperation betwenn $\mathbf{L}_1$ and $\mathbf{L}_2$. Hence we have that,

$$\text{sep}(\mathbf{L}_1, \mathbf{L}_2)^{-1} = \inf_{\mathbf{P} \neq \mathbf{0}} \frac{\left\| \mathbf{T}(\mathbf{P}) \right\|_F}{\left\| \mathbf{P} \right\|_F},$$

and hence,
$$\text{sep}(\mathbf{L}_1, \mathbf{L}_2) = \min|\lambda(\mathbf{L}_{11}) - \lambda(\mathbf{L}_{22})|.$$

Condition (14) becomes,
$$4 \left\| \mathbf{G} \right\| \left\| \mathbf{H} \right\| \text{sep}(\mathbf{L}_1, \mathbf{L}_2)^{-2} < 1. \tag{15}$$

If we assume that $\mathbf{E} = \mathbf{E}^\top$ then $\mathbf{G}^\top = \mathbf{H}$ and hence, condition (15) becomes,
$$4 \left\| \mathbf{G} \right\|^2 \text{sep}(\mathbf{L}_1, \mathbf{L}_2)^{-2} < 1. \tag{16}$$

or equivalently,

$$4 \left( \frac{\left\| \mathbf{E}_{12} \right\|}{\text{sep}(\mathbf{L}_1, \mathbf{L}_2)} \right)^2 < 1.$$

which holds if and only if,

$$\left\| \mathbf{E}_{12} \right\| < \frac{\text{sep}(\mathbf{L}_1, \mathbf{L}_2)}{2}. \tag{17}$$

Here I need to explicitly show that if condition (17) is satisfied then the spectra of the diagonal blocks of the perturbed matrix are separated. In order to show the first point it is enough to establish that,

$$\text{sep}(\mathbf{L}_1, \mathbf{L}_2) > 0.$$

Indeed, we have,

$$\mathbf{T}(\mathbf{P}) = \mathbf{P}\mathbf{\Lambda}_{11} - \mathbf{\Lambda}_{22}\mathbf{P} + \left(\mathbf{P}\mathbf{X}_1^\top \mathbf{E}\mathbf{X}_1^\top - \mathbf{Y}_2^\top \mathbf{E}\mathbf{Y}_2\mathbf{P}\right),$$

$$\|\mathbf{T}(\mathbf{P})\| \geq |\left\|\mathbf{P}\mathbf{\Lambda}_{11} - \mathbf{\Lambda}_{22}\mathbf{P}\right\| - \left\|\mathbf{P}\mathbf{X}_1^\top \mathbf{E}\mathbf{X}_1^\top - \mathbf{Y}_2^\top \mathbf{E}\mathbf{Y}_2\mathbf{P}\right\||,$$

$$\geq |\left\|\mathbf{P}\mathbf{\Lambda}_{11} - \mathbf{\Lambda}_{22}\mathbf{P}\right\| - \left\|\mathbf{P}\mathbf{X}_1^\top \mathbf{E}\mathbf{X}_1\right\| - \left\|\mathbf{Y}_1^\top \mathbf{E}\mathbf{Y}_1\mathbf{P}\right\||,$$

$$\geq |\left\|\mathbf{P}\mathbf{\Lambda}_{11} - \mathbf{\Lambda}_{22}\mathbf{P}\right\| - \|\mathbf{P}\| \left\|\mathbf{X}_1^\top \mathbf{E}\mathbf{X}_1\right\| - \|\mathbf{P}\| \left\|\mathbf{Y}_2^\top \mathbf{E}\mathbf{Y}_2\right\||,$$

$$\geq |\left\|\mathbf{P}\mathbf{\Lambda}_{11} - \mathbf{\Lambda}_{22}\mathbf{P}\right\| - \|\mathbf{P}\| (\left\|\mathbf{X}_1^\top \mathbf{E}\mathbf{X}_1\right\| + \left\|\mathbf{Y}_2^\top \mathbf{E}\mathbf{Y}_2\right\|)|.$$

Thus taking the infinimum over the set of all $\{\mathbf{P} : \|\mathbf{F}\| = 1\}$,

$$\text{sep}(\mathbf{L}_1, \mathbf{L}_2) \geq \text{sep}(\mathbf{\Lambda}_{11}, \mathbf{\Lambda}_{22}) - (\left\|\mathbf{E}_{11}\right\| + \left\|\mathbf{E}_{22}\right\|).$$

Morover, by bounds (13) and (18), it is enoufh to require,

$$\|\mathbf{E}\| < \frac{\text{sep}(\mathbf{\Lambda}_{11}, \mathbf{\Lambda}_{22})}{2}, \tag{18}$$

If we choose the Froebenius norm, we have,

$$\text{sep}(\mathbf{\Lambda}_{11}, \mathbf{\Lambda}_{22}) = \min|\lambda(\mathbf{\Lambda}_{11}) - \lambda(\mathbf{\Lambda}_{22})|.$$

The condition reduces to,

$$\|\mathbf{E}\| < \frac{\min|\lambda(\mathbf{\Lambda}_{11}) - \lambda(\mathbf{\Lambda}_{22})|}{2}.$$

$\square$

### D.7 Proposition 2

#### D.7.1 Simplification

Two simplifications are helpful here. First, the analysis of DGP-S reduces to that of DGP-U. Second, it is enough to consider only one ratio which will allow us to make the ratio more tractable by resorting to the generalized eigenvalue problem.

**Equivalence between DGP-U and DGP-S** Indeed, $y \sim \text{Rademecher}(\frac{1}{2})$, hence $\mathbb{E}[y] = 0$ and $\mathbb{V}[y] = 1$. Hence, $\mathbf{\Phi}_y = \mathbf{I}_2$. $\mathbf{A}_\mathbf{l} = \text{Cov}[\mathbf{x}, y \mid \mathbf{l}] \mathbb{V}[y]^{-1} = \mathbb{E}[\mathbf{x}y \mid \mathbf{l}] - \mathbb{E}[\mathbf{x} \mid \mathbf{l}] \mathbb{E}[y] = \mathbb{E}[\mathbf{x}y \mid \mathbf{l}]$. Now,

$$\mathbb{E}[\mathbf{x}y \mid \mathbf{l}] = \frac{1}{2} \mathbb{E}[\mathbf{x}y \mid \mathbf{l}, y = 1] + \frac{1}{2} \mathbb{E}[\mathbf{x}y \mid \mathbf{l}, y = -1],$$

$$= \mathbf{A}^\top \boldsymbol{\mu} + \mathbf{B}^\top \boldsymbol{\mu}_\mathbf{l}.$$

Hence, $\mathbf{A}_\mathbf{l} = \mathbf{A}^\top \boldsymbol{\mu} + \mathbf{B}^\top \boldsymbol{\mu}_\mathbf{l}$. Following DGP-0, $\mathbf{x}_\mathbf{l} = \mathbf{A}_\mathbf{l}y + \mathbf{b}_\mathbf{l} + \sqrt{\mathbf{\Gamma}_\mathbf{l}}\boldsymbol{\varepsilon}$. Taking the expectation we have, $\mathbb{E}[\mathbf{x} \mid \mathbf{l}] = \mathbf{b}_\mathbf{l}$, and thus,

$$\mathbf{b}_\mathbf{l} = \mathbb{E}[\mathbf{x} \mid \mathbf{l}],$$

$$= \frac{1}{2} \mathbb{E}[\mathbf{A}^\top \boldsymbol{\mu} + \mathbf{B}^\top \boldsymbol{\mu}_\mathbf{l} \mid \mathbf{l}, y = 1] - \frac{1}{2} \mathbb{E}[\mathbf{A}^\top \boldsymbol{\mu} + \mathbf{B}^\top \boldsymbol{\mu}_\mathbf{l} \mid \mathbf{l}, y = -1],$$

$$= \mathbf{0}.$$

Hence the arguments to the fundamental $\mathbf{\Phi}$-embedding is given by,

$$\mathbf{X}_\mathbf{l} = \mathbf{A}^\top \mathbf{\Sigma} \mathbf{A} + \mathbf{B}^\top \mathbf{\Sigma}_\mathbf{l} \mathbf{B},$$

$$\mathbf{Y}_\mathbf{l} = [(\mathbf{A}^\top \boldsymbol{\mu} + \mathbf{B}^\top \boldsymbol{\mu}_\mathbf{l}) \, \mathbf{0}].$$

In this setting the fundamental $\mathbf{\Phi}$-embedding of DGP-U and DGP-S coincide.

**It is enough to consider a single ratio**   Consider the state dependence,

$$S = \frac{1}{2} \text{Tr}[\mathbb{E}\left[\mathbb{E}\left[\mathbf{\Phi_l}\right] \bullet \mathbf{\Phi_l}\right] - \mathbf{I}].$$

We have,

$$\text{Tr}\left[\mathbb{E}\left[\mathbb{E}\left[\mathbf{\Phi_l}\right] : \mathbf{\Phi_l}\right] - \mathbf{I}\right] = \text{Tr}\left[\mathbb{E}\left[\mathbf{\Phi_l} : \mathbf{\Phi_{l'}} - \mathbf{I}\right]\right],$$
$$= \mathbb{P}(\mathbf{l} = 1)\mathbb{P}(\mathbf{l'} = 2)\left(\text{Tr}\left[\mathbf{\Phi_1} : \mathbf{\Phi_2} + \mathbf{\Phi_2} : \mathbf{\Phi_1} - 2\mathbf{I}\right]\right).$$

By proposition 27 $\text{Tr}[\mathbf{\Phi_2} : \mathbf{\Phi_1}] = \text{Tr}[(\mathbf{\Phi_1} : \mathbf{\Phi_2})^{-1}]$. Thus it is enough to look for a state independent subspace in $\mathbf{\Phi_1} : \mathbf{\Phi_2}$. Diagonalizing $(\mathbf{\Phi_1}, \mathbf{\Phi_2})$ by congruence. There exists a non-singular matrix, $\mathbf{\Phi_2}$-orthonormal, $\mathbf{V}$ such that, $\mathbf{\Phi_2}^{-1}\mathbf{\Phi_1} = \mathbf{V}\mathbf{\Lambda}\mathbf{V}^{-1}$. Hence,

$$S = \frac{\mathbb{P}(\mathbf{l} = 1)\mathbb{P}(\mathbf{l} = 2)}{2}(\text{Tr}[\mathbf{\Lambda} + \mathbf{\Lambda}^{-1} - 2\mathbf{I}]).$$

Moreover, $\mathbf{W} = \mathbf{\Phi_2}^{\frac{1}{2}}\mathbf{V}$ is orthonormal and Diagonalizes $\mathbf{\Phi_1} : \mathbf{\Phi_2}$.

### D.7.2   Two useful lemmas

**Lemma 4.**
$$\mathbf{A}\mathbf{A}^\top - \mathbf{B}\mathbf{B}^\top = \frac{1}{2}\left(\mathbf{A} - \mathbf{B}\right)\left(\mathbf{A}^\top + \mathbf{B}^\top\right) + \frac{1}{2}\left(\mathbf{A} + \mathbf{B}\right)\left(\mathbf{A}^\top - \mathbf{B}^\top\right).$$

*Proof.* Direct computation. □

Under DGP-U,
**Lemma 5.**
$$\mathbf{\Phi_{x|l}} = (\mathbf{F} \oplus 1)^\top \mathbf{\Phi_{z|l}}(\mathbf{F} \oplus 1).$$

*Proof.* Direct computation. □

### D.7.3   DGP-U admits 1 as an eigenvalue

The next proposition shows that for any two different states $i$, $j$ the state conditional pencils of $\mathbf{\Phi}$-embeddings under DGP-U admit 1 as an eigenvalue with geometric multiplicity equal to $d_c$.

**Proposition 16** (1 as eigenvalue for DGP-U). *Let $i \neq j \in |\mathcal{L}|$. Then, $1 \in \lambda(\mathbf{\Phi_{x|l=i}}, \mathbf{\Phi_{x|l=j}})$ and $m(1) = d_c$.*

*Proof.* First we have, $\mathbf{\Phi} = (\mathbf{F} \oplus 1)^\top (\mathbf{\Phi}_i - \lambda\mathbf{\Phi}_j)(\mathbf{F} \oplus 1)$, $\mathbf{F} = \begin{bmatrix} \mathbf{A} \\ \mathbf{B} \end{bmatrix} \in \text{Hom}\left(\mathbb{R}^d, \mathbb{R}^{d_c+d_l}\right)$. (Rosenfeld et al., 2020) requires $\mathbf{F}^\top$ to be in injective; $\dim(\text{Ker}(\mathbf{F}^\top)) = 0$ and subsequently also surjective. Hence, the injectivity assumption on $\mathbf{F}$ requires,

$$d = d_c + d_l. \tag{19}$$

Moreover, $\text{rank } \mathbf{F} = \text{rank } \mathbf{A} + \text{rank } \mathbf{B} - \dim\left(\mathcal{R}(\mathbf{A}^\top) \cap \mathcal{R}(\mathbf{B}^\top)\right)$. We therefore rank $\mathbf{F} = d_c + d_l - \dim\left(\mathcal{R}(\mathbf{A}^\top) \cap \mathcal{R}(\mathbf{B}^\top)\right)$. By equation (19), $\dim\left(\mathcal{R}(\mathbf{A}^\top) \cap \mathcal{R}(\mathbf{B}^\top)\right) = 0$.

Form the characteristic polynomial for $(\mathbf{\Phi_{x|l=i}}, \mathbf{\Phi_{x|l=j}})$, $p(\lambda) := \det\left[\mathbf{\Phi_{x|l=i}} - \lambda\mathbf{\Phi_{x|l=j}}\right]$. Let $\mathbf{\Phi}_i = \mathbf{\Phi_{z|l=i}}$, from lemma 5 , $\mathbf{\Phi_{x|l=i}} - \lambda\mathbf{\Phi_{x|l=j}} = (\mathbf{F} \oplus 1)^\top (\mathbf{\Phi}_i - \lambda\mathbf{\Phi}_j)(\mathbf{F} \oplus 1)$. Expanding $\mathbf{\Phi}$-embedding for $\mathbf{\Phi}_i$,

$$\mathbf{\Phi}_i = \begin{bmatrix} \boldsymbol{\mu_c}\boldsymbol{\mu_c}^\top + \mathbf{\Sigma_c} & \boldsymbol{\mu_c}\boldsymbol{\mu_i}^\top & \boldsymbol{\mu_c} \\ \boldsymbol{\mu_i}\boldsymbol{\mu_c}^\top & \boldsymbol{\mu_i}\boldsymbol{\mu_i}^\top + \mathbf{\Sigma}_i & \boldsymbol{\mu_i} \\ \boldsymbol{\mu_c}^\top & \boldsymbol{\mu_i}^\top & 1 \end{bmatrix}.$$

This yields,

$$\boldsymbol{\Phi}_i - \lambda \boldsymbol{\Phi}_j = \begin{bmatrix} (1-\lambda)\left(\boldsymbol{\mu}_c \boldsymbol{\mu}_c^\top\right) + \boldsymbol{\Sigma}_c - \lambda \boldsymbol{\Sigma}_c & \boldsymbol{\mu}_c \left(\boldsymbol{\mu}_i^\top - \lambda \boldsymbol{\mu}_j^\top\right) & \boldsymbol{\mu}_c - \lambda \boldsymbol{\mu}_c \\ \left(\boldsymbol{\mu}_i - \lambda \boldsymbol{\mu}_j\right) \boldsymbol{\mu}_c^\top & \boldsymbol{\mu}_i \boldsymbol{\mu}_i^\top + \boldsymbol{\Sigma}_i - \lambda \boldsymbol{\Sigma}_j - \lambda \left(\boldsymbol{\mu}_j \boldsymbol{\mu}_j^\top\right) & \boldsymbol{\mu}_i - \lambda \boldsymbol{\mu}_j \\ \boldsymbol{\mu}_c^\top - \lambda \boldsymbol{\mu}_c^\top & \boldsymbol{\mu}_i^\top - \lambda \boldsymbol{\mu}_j^\top & 1 - \lambda \end{bmatrix}.$$

In particular if $\lambda = 1$,

$$\boldsymbol{\Phi}_i - \boldsymbol{\Phi}_j = \begin{bmatrix} \mathbf{0} & \boldsymbol{\mu}_c \left(\boldsymbol{\mu}_i^\top - \boldsymbol{\mu}_j^\top\right) & \mathbf{0} \\ \left(\boldsymbol{\mu}_i - \boldsymbol{\mu}_j\right) \boldsymbol{\mu}_c^\top & \boldsymbol{\mu}_i \boldsymbol{\mu}_i^\top + \boldsymbol{\Sigma}_i - \boldsymbol{\Sigma}_j - \boldsymbol{\mu}_j \boldsymbol{\mu}_j^\top & \boldsymbol{\mu}_i - \boldsymbol{\mu}_j \\ \mathbf{0} & \boldsymbol{\mu}_i^\top - \boldsymbol{\mu}_j^\top & \mathbf{0} \end{bmatrix}. \tag{20}$$

We have $\boldsymbol{\Phi} := \boldsymbol{\Phi}_{\mathbf{x}|\mathbf{l}=i} - \boldsymbol{\Phi}_{\mathbf{x}|\mathbf{l}=j} = (\mathbf{F} \oplus 1)^\top (\boldsymbol{\Phi}_i - \boldsymbol{\Phi}_j)(\mathbf{F} \oplus 1)$. Expanding,

$$\boldsymbol{\Phi} = \begin{bmatrix} \left(\mathbf{B}^\top \left(\boldsymbol{\mu}_i \boldsymbol{\mu}_i^\top + \boldsymbol{\Sigma}_i - \boldsymbol{\Sigma}_j - \boldsymbol{\mu}_j \boldsymbol{\mu}_j^\top\right) + \mathbf{A}^\top \boldsymbol{\mu}_c \left(\boldsymbol{\mu}_i^\top - \boldsymbol{\mu}_j^\top\right)\right) \mathbf{B} + \mathbf{B}^\top \left(\boldsymbol{\mu}_i - \boldsymbol{\mu}_j\right) \boldsymbol{\mu}_c^\top \mathbf{A} & \mathbf{B}^\top \left(\boldsymbol{\mu}_i - \boldsymbol{\mu}_j\right) \\ \left(\boldsymbol{\mu}_i^\top - \boldsymbol{\mu}_j^\top\right) \mathbf{B} & \mathbf{0} \end{bmatrix}.$$

By lemma 5

$$\boldsymbol{\Phi} = (\mathbf{B} \oplus 1)^\top \begin{bmatrix} \left(\left(\boldsymbol{\mu}_i \boldsymbol{\mu}_i^\top + \boldsymbol{\Sigma}_i - \boldsymbol{\Sigma}_j - \boldsymbol{\mu}_j \boldsymbol{\mu}_j^\top\right) + \mathbf{A}^\top \boldsymbol{\mu}_c \left(\boldsymbol{\mu}_i^\top - \boldsymbol{\mu}_j^\top\right)\right) + \left(\boldsymbol{\mu}_i - \boldsymbol{\mu}_j\right) \boldsymbol{\mu}_c^\top \mathbf{A} & \left(\boldsymbol{\mu}_i - \boldsymbol{\mu}_j\right) \\ \left(\boldsymbol{\mu}_i^\top - \boldsymbol{\mu}_j^\top\right) & \mathbf{0} \end{bmatrix} (\mathbf{B} \oplus 1).$$

Now $\mathbf{B} \in \mathcal{M}_{d_l \times d}(\mathbb{R})$, by equation (19) $d_l < d$. Hence, rank $\mathbf{B} \le d_l < d$ and rank($\mathbf{B} \oplus 1$) = rank $\mathbf{B} + 1 \le d_l + 1 < d + 1$. Thus rank($\boldsymbol{\Phi}_{\mathbf{x}|\mathbf{l}=i} - \boldsymbol{\Phi}_{\mathbf{x}|\mathbf{l}=j}$) $\le d_l + 1$ which implies $p(1) = \det \left[\boldsymbol{\Phi}_{\mathbf{x}|\mathbf{l}=i} - \boldsymbol{\Phi}_{\mathbf{x}|\mathbf{l}=j}\right] = 0$ and thus $1 \in \lambda(\boldsymbol{\Phi}_{\mathbf{x}|\mathbf{l}=i}, \boldsymbol{\Phi}_{\mathbf{x}|\mathbf{l}=j})$.

The (geometric) multiplicity of the eigenvalue 1 of $\boldsymbol{\Phi}$ is equal to the dimensionality of Ker($\boldsymbol{\Phi}$). $\boldsymbol{\Phi}$ being finite dimensional it is enough to look at its rank.

Now, by the injectivity assumption rank($\mathbf{F} \oplus 1$) = rank($\mathbf{F}$) + 1. $\mathbf{F} \oplus 1$ being non-singular; rank($\boldsymbol{\Phi}$) = rank($\boldsymbol{\Phi}_i - \boldsymbol{\Phi}_j$). It is enough to consider the rank of $\boldsymbol{\Phi}_i - \boldsymbol{\Phi}_j$.

$$\boldsymbol{\Phi}_i - \boldsymbol{\Phi}_j = \begin{bmatrix} \mathbf{0} & \boldsymbol{\mu}_c \left(\boldsymbol{\mu}_i^\top - \boldsymbol{\mu}_j^\top\right) & \mathbf{0} \\ \left(\boldsymbol{\mu}_i - \boldsymbol{\mu}_j\right) \boldsymbol{\mu}_c^\top & \boldsymbol{\mu}_i \boldsymbol{\mu}_i^\top + \boldsymbol{\Sigma}_i - \boldsymbol{\Sigma}_j - \boldsymbol{\mu}_j \boldsymbol{\mu}_j^\top & \boldsymbol{\mu}_i - \boldsymbol{\mu}_j \\ \mathbf{0} & \boldsymbol{\mu}_i^\top - \boldsymbol{\mu}_j^\top & \mathbf{0} \end{bmatrix}.$$

By block Gauss Jordan elimination $\boldsymbol{\Phi}_i - \boldsymbol{\Phi}_j$ is congruent to,

$$\begin{bmatrix} \left(\boldsymbol{\mu}_i - \boldsymbol{\mu}_j\right) \boldsymbol{\mu}_c^\top & \boldsymbol{\mu}_i \boldsymbol{\mu}_i^\top + \boldsymbol{\Sigma}_i - \boldsymbol{\Sigma}_j - \boldsymbol{\mu}_j \boldsymbol{\mu}_j^\top & \boldsymbol{\mu}_i - \boldsymbol{\mu}_j \\ \mathbf{0} & \boldsymbol{\mu}_i^\top - \boldsymbol{\mu}_j^\top & \mathbf{0} \\ \mathbf{0} & \mathbf{0} & \mathbf{0} \end{bmatrix}.$$

The rank of $\boldsymbol{\Phi}_i - \boldsymbol{\Phi}_j$ can then be read directly and hence rank($\boldsymbol{\Phi}_i - \boldsymbol{\Phi}_j$) = $d_l + 1$ = rank($\boldsymbol{\Phi}$). By the fundamental theorem of linear maps, dim Ker($\boldsymbol{\Phi}$) = $d_c$. $\qquad \square$

### D.7.4 RESTRICTION TO INDEPENDENT SUBSPACE

Next we show that the restriction of the $\boldsymbol{\Phi}$-embeddings to $\mathcal{E}_1$ are equal.

**Proposition 17.** *Let* $\left(\boldsymbol{\Phi}_{\mathbf{x}|\mathbf{l}=i}, \boldsymbol{\Phi}_{\mathbf{x}|\mathbf{l}=j}\right)$ *be a pencil of* $\boldsymbol{\Phi}$-*embeddings under* DGP-U. *Let* $\mathcal{E}_1$ *the pencils eigenspace associated to* $\lambda = 1$. *Then,*

$$\boldsymbol{\Phi}_{\mathbf{x}|\mathbf{l}=i}|_{\mathcal{E}_1} = \boldsymbol{\Phi}_{\mathbf{x}|\mathbf{l}=j}|_{\mathcal{E}_1}.$$

*Proof.* proposition 16 establishes that $\left(\boldsymbol{\Phi}_{\mathbf{x}|\mathbf{l}=i}, \boldsymbol{\Phi}_{\mathbf{x}|\mathbf{l}=j}\right)$ admits an eigenspace $\mathcal{E}_1$. Let $\left(\boldsymbol{\Lambda} = \mathbf{I} \oplus \boldsymbol{\Lambda}_2, \mathbf{V} = \begin{bmatrix} \mathbf{V}_1 & \mathbf{V}_2 \end{bmatrix}\right)$ be an eigensystem of $(\boldsymbol{\Phi}_{\mathbf{x}|\mathbf{l}=i}, \boldsymbol{\Phi}_{\mathbf{x}|\mathbf{l}=j})$. First the matrix $\mathbf{V}$ is non-singular. Let's assume that it is $\boldsymbol{\Phi}_{\mathbf{x}|=j}$ normalized. Consider the spectral resolutions,

$$\begin{bmatrix} \mathbf{V}_1^\top \\ \mathbf{V}_2^\top \end{bmatrix} \boldsymbol{\Phi}_{\mathbf{x}|\mathbf{l}=i} \begin{bmatrix} \mathbf{V}_1 & \mathbf{V}_2 \end{bmatrix} = \begin{bmatrix} \mathbf{I} & \mathbf{0} \\ \mathbf{0} & \boldsymbol{\Lambda}_2 \end{bmatrix},$$

$$\begin{bmatrix} \mathbf{V}_1^\top \\ \mathbf{V}_2^\top \end{bmatrix} \boldsymbol{\Phi}_{\mathbf{x}|\mathbf{l}=j} \begin{bmatrix} \mathbf{V}_1 & \mathbf{V}_2 \end{bmatrix} = \begin{bmatrix} \mathbf{I} & \mathbf{0} \\ \mathbf{0} & \mathbf{I} \end{bmatrix}.$$

From here we will deduce that $\mathbf{\Phi}_{\mathbf{x}|\mathbf{l}=i}|_{\mathscr{E}_1} = \mathbf{\Phi}_{\mathbf{x}|\mathbf{l}=j}|_{\mathscr{E}_1}$. Let's express the restrictions of the $\mathbf{\Phi}$-embeddings to $\mathscr{E}_1$. First we need an injection from $\mathbb{R}^{m+n}$ to $\mathscr{E}_1$. $\mathbf{V}_1 \in \text{Hom}(\mathbb{R}^{d_c}, \mathbb{R}^d)$. Hence, $J_{\mathscr{E}_1} = \mathbf{V}_1$. The restrictions to , $\mathbf{\Phi}_{\mathbf{x}|\mathbf{l}=i}|_{\mathscr{E}_1} = \mathbf{V}_1^\dagger \mathbf{\Phi}_{\mathbf{x}|\mathbf{l}=i} \mathbf{V}_1 = \mathbf{\Phi}_{\mathbf{x}|\mathbf{l}=j}|_{\mathscr{E}_1} = \mathbf{V}_1^\dagger \mathbf{\Phi}_{\mathbf{x}|\mathbf{l}=j} \mathbf{V}_1$ We have

$$\mathbf{\Phi}_{\mathbf{x}|\mathbf{l}=i}|_{\mathscr{E}_1} = \mathbf{\Phi}_{\mathbf{x}|\mathbf{l}=j}|_{\mathscr{E}_1} \iff \mathbf{V}_1^\dagger \mathbf{\Phi}_{\mathbf{x}|\mathbf{l}=i} \mathbf{V}_1 = \mathbf{V}_1^\dagger \mathbf{\Phi}_{\mathbf{x}|\mathbf{l}=j} \mathbf{V}_1.$$

Noting that the columns of $\mathbf{V}_1$ being independent, $\mathbf{V}_1^\dagger = (\mathbf{V}_1^\top \mathbf{V}_1)^{-1} \mathbf{V}_1^\top$. Hence,

$$\mathbf{\Phi}_{\mathbf{x}|\mathbf{l}=i}|_{\mathscr{E}_1} = \mathbf{\Phi}_{\mathbf{x}|\mathbf{l}=j}|_{\mathscr{E}_1} \iff (\mathbf{V}_1^\top \mathbf{V}_1)^{-1} \mathbf{V}_1^\top \mathbf{\Phi}_{\mathbf{x}|\mathbf{l}=i} \mathbf{V}_1 = (\mathbf{V}_1^\top \mathbf{V}_1)^{-1} \mathbf{V}_1^\top \mathbf{\Phi}_{\mathbf{x}|\mathbf{l}=j} \mathbf{V}_1 \iff \mathbf{V}_1^\top (\mathbf{\Phi}_{\mathbf{x}|\mathbf{l}=i} - \mathbf{\Phi}_{\mathbf{x}|\mathbf{l}=j}) \mathbf{V}_1 = \mathbf{0}.$$

$\square$

### D.7.5 Condition on DGP-U moments

Now that we have an equivalent condition for the equality of the restriction to $\mathscr{E}_1$. We will exploit the block spectral structure of pencils of $\mathbf{\Phi}$-embeddings in order to deduce from equality of the restrictions of the $\mathbf{\Phi}$-embedding on $\mathscr{E}_1$ constraints on the form of DGP-U.

**Proposition 18.** *Let $\left(\mathbf{\Phi}_{\mathbf{x}|\mathbf{l}=i}, \mathbf{\Phi}_{\mathbf{x}|\mathbf{l}=j}\right)$ be a pencil of $\mathbf{\Phi}$-embeddings under DGP-U. Let $\mathscr{E}_1$ the pencils eigenspace associated to $\lambda = 1$. Let $\mathbf{V}_1$ be a $\mathbf{\Phi}_{\mathbf{x}|\mathbf{l}=j}$-orthonormal matrix such that $\mathscr{R}(\mathbf{V}_1) = \mathscr{E}_1$. Partition $\mathbf{V}_1$ comformably, $\mathbf{V}_1 = \begin{bmatrix} \mathbf{V}_{11} \\ \mathbf{V}_{21} \end{bmatrix}$. Then,*

$$\begin{cases} \mathbf{V}_{11}^\top \mathbf{B}^\top \left(\mathbf{\Sigma}_i - \mathbf{\Sigma}_j\right) \mathbf{B} \mathbf{V}_{11} &= \mathbf{0}, \\ \mathbf{V}_{11}^\top \mathbf{B}^\top \left(\boldsymbol{\mu}_i - \boldsymbol{\mu}_j\right) &= \mathbf{0}. \end{cases} \tag{21}$$

*Proof.* Partition $\mathbf{V}_1$ comformably, $\mathbf{V}_1 = \begin{bmatrix} \mathbf{V}_{11} \\ \mathbf{V}_{21} \end{bmatrix}$. We have $\mathbf{\Phi}_{\mathbf{x}|\mathbf{l}=k} = \mathbf{\Phi}(\mathbf{A}_k, \mathbf{B}_k)$ for some $\mathbf{A}_k \in \mathcal{S}_\succ(m)$ and $\mathbf{B}_k \in \mathbb{R}^{m \times n}$. By proposition 17, $\mathbf{V}_1^\top \left(\mathbf{\Phi}(\mathbf{A}_i, \mathbf{B}_i) - \mathbf{\Phi}(\mathbf{A}_j, \mathbf{B}_j)\right) \mathbf{V}_1 = \mathbf{0}$, if and only,

$$\mathbf{V}_{11}^\top \left(\mathbf{B}_i \mathbf{B}_i^\top + \mathbf{A}_i - \mathbf{A}_j - \mathbf{B}_j \mathbf{B}_j^\top\right) \mathbf{V}_{11} + \mathbf{V}_{21}^\top \left(\mathbf{B}_i^\top - \mathbf{B}_j^\top\right) \mathbf{V}_{11} + \mathbf{V}_{11}^\top \left(\mathbf{B}_i - \mathbf{B}_j\right) \mathbf{V}_{21} = \mathbf{0},$$

$$\mathbf{V}_{11}^\top (\mathbf{A}_i - \mathbf{A}_j) \mathbf{V}_{11} + \mathbf{V}_{11}^\top (\mathbf{B}_i \mathbf{B}_i^\top - \mathbf{B}_j \mathbf{B}_j^\top) \mathbf{V}_{11} + \mathbf{V}_{21}^\top \left(\mathbf{B}_i^\top - \mathbf{B}_j^\top\right) \mathbf{V}_{11} + \mathbf{V}_{11}^\top \left(\mathbf{B}_i - \mathbf{B}_j\right) \mathbf{V}_{21} = \mathbf{0}.$$

By lemma 4,

$$\mathbf{B}_i \mathbf{B}_i^\top - \mathbf{B}_j \mathbf{B}_j^\top = \frac{1}{2} \left(\mathbf{B}_i - \mathbf{B}_j\right) \left(\mathbf{B}_i^\top + \mathbf{B}_j^\top\right) + \frac{1}{2} \left(\mathbf{B}_i + \mathbf{B}_j\right) \left(\mathbf{B}_i^\top - \mathbf{B}_j^\top\right).$$

Hence, $\mathbf{V}_1^\top \left(\mathbf{\Phi}(\mathbf{A}_i, \mathbf{B}_i) - \mathbf{\Phi}(\mathbf{A}_j, \mathbf{B}_j)\right) \mathbf{V}_1 = \mathbf{0}$, if and only,

$$\mathbf{V}_{11}^\top (\mathbf{A}_i - \mathbf{A}_j) \mathbf{V}_{11} +$$
$$\mathbf{V}_{11}^\top \left(\frac{1}{2} \left(\mathbf{B}_i - \mathbf{B}_j\right) \left(\mathbf{B}_i^\top + \mathbf{B}_j^\top\right) + \frac{1}{2} \left(\mathbf{B}_i + \mathbf{B}_j\right) \left(\mathbf{B}_i^\top - \mathbf{B}_j^\top\right)\right) \mathbf{V}_{11} +$$
$$\mathbf{V}_{21}^\top \left(\mathbf{B}_i^\top - \mathbf{B}_j^\top\right) \mathbf{V}_{11} +$$
$$\mathbf{V}_{11}^\top \left(\mathbf{B}_i - \mathbf{B}_j\right) \mathbf{V}_{21} = \mathbf{0}.$$

By proposition 18 , $\mathbf{V}_{11}^\top (\mathbf{B}_i - \mathbf{B}_j) = \mathbf{0}$. Hence, $\mathbf{V}_1^\top \left(\mathbf{\Phi}(\mathbf{A}_i, \mathbf{B}_i) - \mathbf{\Phi}(\mathbf{A}_j, \mathbf{B}_j)\right) \mathbf{V}_1 = \mathbf{0}$, if and only,

$$\begin{cases} \mathbf{V}_{11}^\top (\mathbf{A}_i - \mathbf{A}_j) \mathbf{V}_{11} &= \mathbf{0}, \\ \mathbf{V}_{11}^\top (\mathbf{B}_i - \mathbf{B}_j) &= \mathbf{0}. \end{cases} \tag{22}$$

Now in our setting, from DGP-U $\mathbf{A}_k = \mathbf{A}^\top \mathbf{\Sigma}_\mathbf{c} \mathbf{A} + \mathbf{B}^\top \mathbf{\Sigma}_k \mathbf{B}$, and $\mathbf{B}_k = \mathbf{A}^\top \boldsymbol{\mu}_\mathbf{c} + \mathbf{B}^\top \boldsymbol{\mu}_k$. Substituting into equation (22), we conclude

$$\begin{cases} \mathbf{V}_{11}^\top \mathbf{B}^\top \left(\mathbf{\Sigma}_i - \mathbf{\Sigma}_j\right) \mathbf{B} \mathbf{V}_{11} &= \mathbf{0}, \\ \mathbf{V}_{11}^\top \mathbf{B}^\top \left(\boldsymbol{\mu}_i - \boldsymbol{\mu}_j\right) &= \mathbf{0}. \end{cases} \tag{23}$$

$\square$

### D.7.6 RECOVERY MECHANISM

In this section we show explicitly how the spurious parts of DGP-U is annihilated by the eigenvectors spanning $\mathscr{E}_\perp$. Then we show in which sense is the common/invariant parts of DGP-U is recovered.

**Proposition 19.** *Under DGP-U,* $\mathbf{BV}_{11} = \mathbf{0}$.

*Proof.* By equation (21),

$$\mathbf{V}_{11}^\top \mathbf{B}^\top \left( \boldsymbol{\Sigma}_i - \boldsymbol{\Sigma}_j \right) \mathbf{BV}_{11} = \mathbf{0}. \tag{24}$$

By lemma 6 there exists non-singular matrix $\mathbf{Q}$ diagonalizing $(\boldsymbol{\Sigma}_i, \boldsymbol{\Sigma}_j)$. Take $\mathbf{Q}$ to be $\boldsymbol{\Sigma}_j$-orthonormal.Then $\mathbf{Q}^\top \boldsymbol{\Sigma}_i \mathbf{Q} = \mathbf{D}$ and $\mathbf{Q}^\top \boldsymbol{\Sigma}_j \mathbf{Q} = \mathbf{I}$. Substituting in equation (24),

$$\mathbf{V}_{11}^\top \mathbf{B}^\top \mathbf{Q}^{-\top} (\mathbf{D} - \mathbf{I}) \mathbf{Q}^{-1} \mathbf{BV}_{11} = \mathbf{0}. \tag{25}$$

Let $\mathbf{M}^\top := \mathbf{Q}^{-1} \mathbf{BV}_{11}$ and equation (25) reads,

$$\mathbf{MDM}^\top = \mathbf{MM}^\top. \tag{26}$$

Let see equation (26) as linear matrix equation in $\mathbf{D}$. By lemma 7 equation (26) admits a solution if and only if,

$$\mathbf{MM}^\dagger \mathbf{MM}^\top \mathbf{M}^{\dagger\top} \mathbf{M}^\top = \mathbf{MM}^\top, \iff$$
$$\mathbf{MM}^\dagger \mathbf{MM}^\top = \mathbf{MM}^\top, \iff$$
$$\mathbf{MM}^\top = \mathbf{MM}^\top.$$

The general solution is therefore given by,

$$\mathbf{D} = \mathbf{M}^\dagger \mathbf{MM}^\top \mathbf{M}^{\dagger\top} + \mathbf{Q} - \mathbf{M}^\dagger \mathbf{MQM}^\top \mathbf{M}^{\dagger\top},$$
$$\mathbf{X} = \mathbf{M}^\dagger \mathbf{M} + \mathbf{Q} - \mathbf{M}^\dagger \mathbf{MQM}^\top \mathbf{M}^{\dagger\top}.$$

For any arbitrary matrix $\mathbf{Q}$. In particular for $\mathbf{Q} = \mathbf{0}$, the solution $\mathbf{X} = \mathbf{M}^\dagger \mathbf{M}$ is the orthogonal projection onto $\mathscr{R}(\mathbf{M}^\top)$. It is now enough to show that $\mathbf{D}$ cannot satisfy this requirement. Indeed, for $\mathbf{D}$ to be an orthogonal projection it must verifie $\forall i \in \{1, \dots, d_l\} \mathbf{D}_{ii} \in \{0, 1\}$. But by assumption, $\mathbf{D}$ has at least one element different than 1. Moreover, since $\boldsymbol{\Sigma}_1$ and $\boldsymbol{\Sigma}_2$ are assumed to be positive definite the generalized eigenvalues of the pencil are strictly greater than zero. Thus, $\mathbf{D}$ cannot be a solution equation (26); The equation therefore requires that $\mathbf{M}^\top = \mathbf{0}$. $\mathbf{Q}$ being non-singular this is only possible this means that $\mathbf{BV}_{11} = \mathbf{0}$. $\qquad\square$

**Proposition 20.** *Under DGP-U. Let* $\mathbf{V}_1$ *be the eigenmatrix spanning the eigenspace* $\mathscr{E}_\perp$ *of* $(\boldsymbol{\Phi}_{\mathbf{x}|\mathbf{l}=i}, \boldsymbol{\Phi}_{\mathbf{x}|\mathbf{l}=j})$. *Then,*

$$\mathscr{R}(\mathbf{V}_{11}) = \mathscr{R}(\mathbf{A}^\top).$$

*Proof.* By proposition 4 $\boldsymbol{\Phi}_{\mathbf{x}|\mathbf{l}} > \mathbf{0}$ and hence $\left( \boldsymbol{\Phi}_{\mathbf{x}|\mathbf{l}}/\mathbf{I} \right) > \mathbf{0}$. Moreover, $\mathbb{V}[\mathbf{x} \mid \mathbf{l}] > \mathbf{0}$. Hence, $\mathrm{Ker}\left( \mathbb{V}[\mathbf{x} \mid \mathbf{l} = i] \right) \cap \mathrm{Ker}\left( \mathbb{V}[\mathbf{x} \mid \mathbf{l} = j] \right) = \{\mathbf{0}\}$. Under DGP-U, $\mathbb{V}[\mathbf{x} \mid \mathbf{l} = i] = \mathbf{A}^\top \boldsymbol{\Sigma}_\mathbf{c} \mathbf{A} + \mathbf{B}^\top \boldsymbol{\Sigma}_i \mathbf{B}$. Diagonalize $\mathbf{A}^\top \boldsymbol{\Sigma}_\mathbf{c} \mathbf{A}$ and $\mathbf{B}^\top \boldsymbol{\Sigma}_i \mathbf{B}$ by congruence. The there exists a an non-singular matrix $\mathbf{X}$ such that,

$$\mathbf{X}^\top \mathbb{V}[\mathbf{x} \mid \mathbf{l} = i] \mathbf{X} = \mathbf{Diag}(\boldsymbol{\alpha}) + \mathbf{Diag}(\boldsymbol{\beta}).$$

By Sylvester's theorem of inertia, $\alpha$ has exactly $\mathrm{rank}(\mathbf{A}^\top \mathbf{S}_\mathbf{c} \mathbf{A}) = d_c$ positive entries and $d_s$ zero entries. Similarly $\mathbf{B}^\top \boldsymbol{\Sigma}_i \mathbf{B}$ has exactly $\mathrm{rank}(\mathbf{B}^\top \boldsymbol{\Sigma}_i \mathbf{B}) = d_s$ positive entries and $d_c$ zero entries. $\mathbf{X}^\top \mathbb{V}[\mathbf{x} \mid \mathbf{l} = i] \mathbf{X}$ is also diagonal and by positive definitiveness of $\mathbb{V}[\mathbf{x} \mid \mathbf{l} = i]$ all of its entries are positive. By the pigeon-hole principle, there is no single index $i \in [d_c + d_s]$ such that $\alpha_i = \beta_i = 0$. Hence,

$$\mathrm{Ker}\left( \mathbf{A}^\top \boldsymbol{\Sigma}_\mathbf{c} \mathbf{A} \right) \cap \mathrm{Ker}\left( \mathbf{B}^\top \boldsymbol{\Sigma}_i \mathbf{B} \right) \cap \mathrm{Ker}\left( \mathbf{B}^\top \boldsymbol{\Sigma}_j \mathbf{B} \right) = \{\mathbf{0}\}. \tag{27}$$

Let $\mathbf{v}$ be any eigenvector of $\mathscr{E}_1$. proposition 19 implies that $\mathbf{v} \in \mathrm{Ker}(\mathbf{B})$ and hence $\mathbf{v} \in \mathrm{Ker}(\mathbf{B}^\top \boldsymbol{\Sigma}_j \mathbf{B})$. By equation (27), $\mathbf{v} \notin \mathrm{Ker}(\mathbf{A}^\top \boldsymbol{\Sigma}_\mathbf{c} \mathbf{A})$; $\mathbf{v} \in \mathrm{Ker}\left( \mathbf{A}^\top \boldsymbol{\Sigma}_\mathbf{c} \mathbf{A} \right)^\perp = \mathscr{R}\left( \mathbf{A}^\top \boldsymbol{\Sigma}_\mathbf{c} \mathbf{A} \right)$. $\boldsymbol{\Sigma}_c$ being positive definite,

$$\mathscr{R}\left(\mathbf{A}^\top\boldsymbol{\Sigma}_{\mathbf{c}}\mathbf{A}\right) = \mathscr{R}\left(\mathbf{A}^\top\boldsymbol{\Sigma}_c^{\frac{1}{2}}\boldsymbol{\Sigma}_c^{\frac{1}{2}}\mathbf{A}\right). \text{ Now, } \mathscr{R}(\mathbf{A}^\top\boldsymbol{\Sigma}_c^{\frac{1}{2}}\boldsymbol{\Sigma}_c^{\frac{1}{2}}\mathbf{A}) = \mathscr{R}\left(\mathbf{A}^\top\boldsymbol{\Sigma}_c^{\frac{1}{2}}\right). \text{ Hence, } \mathbf{v} \in \mathscr{R}(\mathbf{A}^\top\boldsymbol{\Sigma}_c^{\frac{1}{2}}).$$

Therefore, Writing $\mathbf{V}_{11}$ be which columns are $d_c$ eigenvectors of $\mathscr{E}_1$, $\mathscr{R}(\mathbf{V}_{11}) \subseteq \mathscr{R}(\mathbf{A}^\top\boldsymbol{\Sigma}_c^{\frac{1}{2}})$. By proposition 16, $\text{rank}(\mathscr{R}(\mathbf{V}_1)) = d_c = \text{rank}\left(\boldsymbol{\Sigma}_c^{\frac{1}{2}}\right) = \text{rank}(\mathbf{A}\boldsymbol{\Sigma}_c^{\frac{1}{2}})$. Thus, $\mathscr{R}(\mathbf{V}_{11}) = \mathscr{R}\left(\mathbf{A}^\top\boldsymbol{\Sigma}_c^{\frac{1}{2}}\right) =$

$\mathbf{A}^\top\mathscr{R}(\boldsymbol{\Sigma}_c^{\frac{1}{2}}) = \mathbf{A}^\top\mathbb{R}^{d_c} = \mathscr{R}(\mathbf{A}^\top)$. $\qquad\qquad\qquad\qquad\qquad\qquad\qquad\qquad\qquad\qquad \square$

### D.7.7 Proposition 2

It is now enough to take $\mathbf{P} = \mathbf{V}_{11}\mathbf{V}_{11}^\dagger$ the orthogonal projection onto $\mathscr{R}(\mathbf{V}_{11})$ and we conclude,

**Proposition 2.** *The expected ratio* $\mathbb{E}\left[\mathbb{E}[\boldsymbol{\Phi}_\mathbf{l}] : \boldsymbol{\Phi}_\mathbf{l}\right]$ *of both DGP-U and DGP-S admit a state independent subspace* $\mathscr{E}_\perp$ *of dimension* $d_c$. *Moreover, a matrix* $\mathbf{W}_1^\top = \begin{bmatrix} \mathbf{W}_{11}^\top & \mathbf{W}_{12}^\top \end{bmatrix}$ *such that* $\mathscr{R}(\mathbf{W}_{11}) = \mathscr{E}_\perp$. *Then the orthogonal projection* $\mathbf{P}$ *onto* $\mathscr{R}(\mathbf{W}_{11})$ *verifies,*

    (i) $\mathbf{BP} = \mathbf{0}$ *(Annihilation of state dependent component),*
    (ii) $\mathscr{R}(\mathbf{A}^\top) = \mathscr{R}(\mathbf{P})$ *(Preservation of oracle component),*
    (iii) $\mathbf{Px} = \mathbf{A}^\top\mathbf{c}$ *for DGP-U, and,* $\mathbf{Px} = y\mathbf{A}^\top\mathbf{c}$ *for DGP-S.*

## E Mathematical elements

### E.1 Schur complement

**Definition 8.** Let $\alpha$ and $\beta$ multi-indices of rows and columns of a given matrix $\mathbf{M} \in \mathscr{M}_{r \times c}(\mathbb{R})$. Let $\alpha^\complement, \beta^\complement$ denote their set complement in the row and column indices in $[r]$ and $[c]$ respectively. Define the generalized Schur-complement the matrix $\mathbf{M}[\alpha, \beta]$ in $\mathbf{M}$ is

$$\mathbf{M}/\mathbf{M}[\alpha, \beta] = \mathbf{M}[\alpha^\complement, \beta^\complement] - \mathbf{M}[\alpha^\complement, \beta]\mathbf{M}[\alpha, \beta]^\dagger\mathbf{M}[\alpha, \beta^\complement].$$

When $\alpha = \beta$, the $\beta$ multi-index is ommited, and the generalized schur-complement reads,

$$\mathbf{M}/\mathbf{M}[\alpha] := \mathbf{M}[\alpha^\complement, \alpha^\complement] - \mathbf{M}[\alpha^\complement, \alpha]\mathbf{M}[\alpha, \alpha]^\dagger\mathbf{M}[\alpha, \alpha^\complement].$$

### E.2 LDU and UDL Decomposition

**Proposition 21** (UDL/LDU decompositions)**.**

$$\begin{bmatrix} \mathbf{I} & \mathbf{0} \\ \mathbf{CA}^{-1} & \mathbf{I} \end{bmatrix}\begin{bmatrix} \mathbf{A} & \mathbf{0} \\ \mathbf{0} & \mathbf{D} - \mathbf{CA}^{-1}\mathbf{B} \end{bmatrix}\begin{bmatrix} \mathbf{I} & \mathbf{A}^{-1}\mathbf{B} \\ \mathbf{0} & \mathbf{I} \end{bmatrix} = \begin{bmatrix} \mathbf{A} & \mathbf{B} \\ \mathbf{C} & \mathbf{D} \end{bmatrix} = \begin{bmatrix} \mathbf{I} & \mathbf{BD}^{-1} \\ \mathbf{0} & \mathbf{I} \end{bmatrix}\begin{bmatrix} \mathbf{A} - \mathbf{BD}^{-1}\mathbf{C} & \mathbf{0} \\ \mathbf{0} & \mathbf{D} \end{bmatrix}\begin{bmatrix} \mathbf{I} & \mathbf{0} \\ \mathbf{D}^{-1}\mathbf{C} & \mathbf{I} \end{bmatrix}$$

*Proof.* Let $\mathbf{X} := \begin{bmatrix} \mathbf{A} & \mathbf{B} \\ \mathbf{C} & \mathbf{D} \end{bmatrix}$. First form the pivot Multiply the first row by $\mathbf{A}^{-1}$, $\mathbf{E}_1 = \begin{bmatrix} \mathbf{A}^{-1} & \mathbf{0} \\ \mathbf{0} & \mathbf{I} \end{bmatrix}$.

$$\mathbf{E}_1\mathbf{X} = \begin{bmatrix} \mathbf{I} & \mathbf{A}^{-1}\mathbf{B} \\ \mathbf{C} & \mathbf{D} \end{bmatrix}.$$

Substract $\mathbf{C}$ times the first row from the second, $\mathbf{E}_2 = \begin{bmatrix} \mathbf{I} & \mathbf{0} \\ -\mathbf{C} & \mathbf{I} \end{bmatrix}$.

$$\mathbf{E}_2\mathbf{E}_1\mathbf{X} = \begin{bmatrix} \mathbf{A} & \mathbf{B} \\ \mathbf{0} & \mathbf{D} - \mathbf{CA}^{-1}\mathbf{B} \end{bmatrix}.$$

Multiply the first row by $\mathbf{A}$, $\mathbf{E}_3 = \begin{bmatrix} \mathbf{A} & \mathbf{0} \\ \mathbf{0} & \mathbf{I} \end{bmatrix}$.

$$\mathbf{E}_3\mathbf{E}_2\mathbf{E}_1\mathbf{X} = \begin{bmatrix} \mathbf{A} & \mathbf{B} \\ \mathbf{0} & \mathbf{D} - \mathbf{CA}^{-1}\mathbf{B} \end{bmatrix}.$$

Now proceed similarly from the right and multiply the first column by $\mathbf{A}^{-1}$, $\mathbf{F}_1 = \begin{bmatrix} \mathbf{I} & \mathbf{A}^{-1} \\ \mathbf{0} & \mathbf{I} \end{bmatrix}$,

$$\mathbf{E}_3\mathbf{E}_2\mathbf{E}_1\mathbf{X}\mathbf{F}_1 = \begin{bmatrix} \mathbf{I} & \mathbf{B} \\ \mathbf{0} & \mathbf{D} - \mathbf{C}\mathbf{A}^{-1}\mathbf{B} \end{bmatrix}.$$

Then the second colum gets $-\mathbf{B}$ the first, $\mathbf{F}_2 = \begin{bmatrix} \mathbf{I} & \mathbf{0} \\ -\mathbf{B} & \mathbf{I} \end{bmatrix}$,

$$\mathbf{E}_3\mathbf{E}_2\mathbf{E}_1\mathbf{X}\mathbf{F}_1\mathbf{F}_2 = \begin{bmatrix} \mathbf{I} & \mathbf{0} \\ \mathbf{0} & \mathbf{D} - \mathbf{C}\mathbf{A}^{-1}\mathbf{B} \end{bmatrix}.$$

Finally the first column get multiplied by $\mathbf{A}$, $\mathbf{F}_3 = \begin{bmatrix} \mathbf{A} & \mathbf{0} \\ \mathbf{0} & \mathbf{I} \end{bmatrix}$.

$$\mathbf{E}_3\mathbf{E}_2\mathbf{E}_1\mathbf{X}\mathbf{F}_1\mathbf{F}_2\mathbf{F}_3 = \begin{bmatrix} \mathbf{A} & \mathbf{0} \\ \mathbf{0} & \mathbf{D} - \mathbf{C}\mathbf{A}^{-1}\mathbf{B} \end{bmatrix}.$$

Thus we have,

$$\mathbf{E} := \mathbf{E}_1\mathbf{E}_2\mathbf{E}_3 = \begin{bmatrix} \mathbf{I} & \mathbf{0} \\ -\mathbf{C}\mathbf{A} & \mathbf{I} \end{bmatrix}, \quad \mathbf{F} := \mathbf{F}_1\mathbf{F}_2\mathbf{F}_3 = \begin{bmatrix} \mathbf{I} & -\mathbf{A}^{-1}\mathbf{B} \\ \mathbf{0} & \mathbf{I} \end{bmatrix}.$$

Moreover, $\mathbf{E}^{-1} = \begin{bmatrix} \mathbf{I} & \mathbf{0} \\ \mathbf{C}\mathbf{A}^{-1} & \mathbf{I} \end{bmatrix}$ and $\mathbf{F}^{-1} = \begin{bmatrix} \mathbf{I} & \mathbf{A}^{-1}\mathbf{B} \\ \mathbf{0} & \mathbf{I} \end{bmatrix}$. Hence,

$$\begin{bmatrix} \mathbf{A} & \mathbf{B} \\ \mathbf{C} & \mathbf{D} \end{bmatrix} = \begin{bmatrix} \mathbf{I} & \mathbf{0} \\ \mathbf{C}\mathbf{A}^{-1} & \mathbf{I} \end{bmatrix} \begin{bmatrix} \mathbf{A} & \mathbf{0} \\ \mathbf{0} & \mathbf{D} - \mathbf{C}\mathbf{A}^{-1}\mathbf{B} \end{bmatrix} \begin{bmatrix} \mathbf{I} & \mathbf{A}^{-1}\mathbf{B} \\ \mathbf{0} & \mathbf{I} \end{bmatrix}.$$

To get a UDL decomposition start by using the second row as a pivot, $\mathbf{E}_1 = \begin{bmatrix} \mathbf{I} & \mathbf{0} \\ \mathbf{0} & \mathbf{D}^{-1} \end{bmatrix}$,

$$\mathbf{E}_1\mathbf{X} = \begin{bmatrix} \mathbf{A} & \mathbf{B} \\ \mathbf{D}^{-1}\mathbf{C} & \mathbf{I} \end{bmatrix}.$$

Then the first row gets $-\mathbf{B}$ the second, $\mathbf{E}_2 = \begin{bmatrix} \mathbf{I} & -\mathbf{B} \\ \mathbf{0} & \mathbf{I} \end{bmatrix}$.

$$\mathbf{E}_2\mathbf{E}_1\mathbf{X} = \begin{bmatrix} \mathbf{A} - \mathbf{B}\mathbf{D}^{-1}\mathbf{C} & \mathbf{0} \\ \mathbf{D}^{-1}\mathbf{C} & \mathbf{I} \end{bmatrix}$$

The second row gets multiplied by $\mathbf{D}$, $\mathbf{E}_3 = \begin{bmatrix} \mathbf{I} & \mathbf{0} \\ \mathbf{0} & \mathbf{D} \end{bmatrix}$.

$$\mathbf{E}_3\mathbf{E}_2\mathbf{E}_1\mathbf{X} = \begin{bmatrix} \mathbf{A} - \mathbf{B}\mathbf{D}^{-1}\mathbf{C} & \mathbf{0} \\ \mathbf{C} & \mathbf{D} \end{bmatrix}.$$

Next we start operating on the columns. The second column gets multiplied by $\mathbf{D}^{-1}$, $\mathbf{F}_1 = \begin{bmatrix} \mathbf{I} & \mathbf{0} \\ \mathbf{0} & \mathbf{D}^{-1} \end{bmatrix}$.

$$\mathbf{E}_3\mathbf{E}_2\mathbf{E}_1\mathbf{X}\mathbf{F}_1 = \begin{bmatrix} \mathbf{A} - \mathbf{B}\mathbf{D}^{-1}\mathbf{C} & \mathbf{0} \\ \mathbf{C} & \mathbf{I} \end{bmatrix}.$$

The first column gets $-\mathbf{C}$ the second, $\mathbf{F}_2 = \begin{bmatrix} \mathbf{I} & \mathbf{0} \\ -\mathbf{C} & \mathbf{I} \end{bmatrix}$.

$$\mathbf{E}_3\mathbf{E}_2\mathbf{E}_1\mathbf{X}\mathbf{F}_1\mathbf{F}_2 = \begin{bmatrix} \mathbf{A} - \mathbf{B}\mathbf{D}^{-1}\mathbf{C} & \mathbf{0} \\ \mathbf{0} & \mathbf{I} \end{bmatrix}.$$

Finally the second column gets multiplied by $\mathbf{D}$, $\mathbf{F}_3 = \begin{bmatrix} \mathbf{I} & \mathbf{0} \\ \mathbf{0} & \mathbf{D} \end{bmatrix}$.

$$\mathbf{E}_3\mathbf{E}_2\mathbf{E}_1\mathbf{X}\mathbf{F}_1\mathbf{F}_2\mathbf{F}_3 = \begin{bmatrix} \mathbf{A} - \mathbf{B}\mathbf{D}^{-1}\mathbf{C} & \mathbf{0} \\ \mathbf{0} & \mathbf{D} \end{bmatrix}.$$

$$\mathbf{E} := \mathbf{E}_1\mathbf{E}_2\mathbf{E}_3 = \begin{bmatrix} \mathbf{I} & -\mathbf{B}\mathbf{D} \\ \mathbf{0} & \mathbf{I} \end{bmatrix}, \quad \mathbf{F} := \mathbf{F}_1\mathbf{F}_2\mathbf{F}_3 = \begin{bmatrix} \mathbf{I} & \mathbf{0} \\ -\mathbf{D}^{-1}\mathbf{C} & \mathbf{I} \end{bmatrix}.$$

Hence,

$$\begin{bmatrix} \mathbf{A} & \mathbf{B} \\ \mathbf{C} & \mathbf{D} \end{bmatrix} = \begin{bmatrix} \mathbf{I} & \mathbf{B}\mathbf{D}^{-1} \\ \mathbf{0} & \mathbf{I} \end{bmatrix} \begin{bmatrix} \mathbf{A} - \mathbf{B}\mathbf{D}^{-1}\mathbf{C} & \mathbf{0} \\ \mathbf{0} & \mathbf{D} \end{bmatrix} \begin{bmatrix} \mathbf{I} & \mathbf{0} \\ \mathbf{D}^{-1}\mathbf{C} & \mathbf{I} \end{bmatrix}.$$

$\square$

### E.3 Simultaneous diagonalization of SPD matrices by congruence

**Lemma 6.** *Let* $\mathbf{S}_1, \mathbf{S}_2 \in \mathcal{S}_{++}(d)$. *Then there exists a matrix* $\mathbf{V}$ *diagonalizing both* $\mathbf{S}_1$ *and* $\mathbf{S}_2$ *by congruence while also diagonalizing their quotient,* $\mathbf{S}_2^{-1}\mathbf{S}_1$, *by similarity.*

*Proof.* By the spectral theorem, $\mathbf{D}_1^{-1/2}\mathbf{V}_1^\top\mathbf{S}_1\mathbf{V}_1\mathbf{D}_1^{-1/2} = \mathbf{I}$ and $\mathbf{D}_1^{-1/2}\mathbf{V}_1^\top\mathbf{S}_2\mathbf{V}_1\mathbf{D}_1^{-1/2} = \mathbf{V}\boldsymbol{\Lambda}\mathbf{V}^\top$. The matrix $\mathbf{V}_1\boldsymbol{\Lambda}_1^{-1/2}\mathbf{V}$ simultaneously diagonalize $\mathbf{S}_1$ and $\mathbf{S}_2$ by congruence. $\square$

### E.4 Spectral equivalences

**Proposition 22** (Eigenpairs of the inverse). *Let* $\mathbf{A} \in \mathcal{S}_\succ$. *Then,*

   (i) $\mathbf{A}$ *and* $\mathbf{A}^{-1}$ *share the same set of eigenvectors,*
   (ii) $\lambda(\mathbf{A}^{-1}) = \lambda(\mathbf{A})^{-1}$, *and,*
   (iii) $\lambda^\downarrow(\mathbf{A}^{-1}) = \lambda^\uparrow(\mathbf{A})^{-1}$.

**Proposition 23.** *Let* $\mathbf{A}, \mathbf{B} \in \mathcal{S}_\succ(d)$. *Then,*

   (i) *If* $\mathbf{V}$ *is an eigenmatrix of* $(\mathbf{A}, \mathbf{B})$ *then* $\mathbf{B}^{\frac{1}{2}}\mathbf{V}$ *is one for* $\mathbf{B}^{-\frac{1}{2}}\mathbf{A}\mathbf{B}^{-\frac{1}{2}}$,
   (ii) $\lambda(\mathbf{A}, \mathbf{B}) = \lambda(\mathbf{B}^{-\frac{1}{2}}\mathbf{A}\mathbf{B}^{-\frac{1}{2}})$, *and,*
   (iii) $\lambda^\downarrow(\mathbf{A}, \mathbf{B}) = \lambda^\downarrow(\mathbf{B}^{-\frac{1}{2}}\mathbf{A}\mathbf{B}^{-\frac{1}{2}})$.

*Proof.* $\mathbf{B}$ being positive definite it has a unique non singular square root. Let $(\lambda, \mathbf{v})$ be an eigenpair of $(\mathbf{A}, \mathbf{B})$ if and only, $\mathbf{A}\mathbf{V} = \lambda\mathbf{B}\mathbf{v} \iff \mathbf{B}^{-\frac{1}{2}}\mathbf{A}\mathbf{v} = \lambda\mathbf{B}^{\frac{1}{2}}\mathbf{v}$ (iii). Writing $\mathbf{w} = \mathbf{B}^{\frac{1}{2}}\mathbf{v}$, then $(\lambda, \mathbf{v})$ if and only if $\mathbf{B}^{-\frac{1}{2}}\mathbf{A}\mathbf{B}^{-\frac{1}{2}}\mathbf{w} = \lambda\mathbf{w}$, which holds if and only if $(\lambda, \mathbf{w})$ is an eigenpair of $\mathbf{B}^{-\frac{1}{2}}\mathbf{A}\mathbf{B}^{-\frac{1}{2}}$, this proves (i), (ii), and (iii). $\square$

**Proposition 24.** *Let* $\mathbf{A}, \mathbf{B} \in \mathcal{S}_\succ(d)$. *Then,*

   (i) $(\mathbf{A}, \mathbf{B})$ *and* $(\mathbf{B}, \mathbf{A})$ *share the same eigenvectors,*
   (ii) $\lambda(\mathbf{B}, \mathbf{A}) = \lambda(\mathbf{A}, \mathbf{B})^{-1}$,
   (iii) $\lambda^\downarrow(\mathbf{B}, \mathbf{A}) = \lambda^\uparrow(\mathbf{A}, \mathbf{B})$.

*Proof.* By proposition 23 $(\lambda, \mathbf{v})$ is an eigenpair of $(\mathbf{A}, \mathbf{B})$ if and only if it verifies $\mathbf{A}\mathbf{v} = \lambda\mathbf{B}\mathbf{v}$, if and only if, $\mathbf{B}\mathbf{v} = \frac{1}{\lambda}\mathbf{A}\mathbf{v}$ if and only if $(\frac{1}{\lambda}, \mathbf{v})$ is an eigenpair of $(\mathbf{B}, \mathbf{A})$ (i)(ii). The map $t \mapsto \frac{1}{t}$ being order reversing, $\lambda^\downarrow(\mathbf{B}, \mathbf{A}) = \lambda^\uparrow(\mathbf{A}, \mathbf{B})$. $\square$

**Proposition 25.** *Let* $\mathbf{A}, \mathbf{B} \in \mathcal{S}_\succ(d)$. *Let* $\mathbf{L}$ *be block lower triangular and* $\mathbf{D}$ *be block diagonal with positive definite blocks such that* $\mathbf{B} = \mathbf{L}^\top\mathbf{D}\mathbf{L}$. *Then,*

   (i) *If* $\mathbf{V}$ *is an eigenmatrix of* $(\mathbf{A}, \mathbf{B})$ *then* $\mathbf{D}^{\frac{1}{2}}\mathbf{L}\mathbf{V}$ *is one for* $\mathbf{D}^{-\frac{1}{2}}\mathbf{L}^{-1}\mathbf{A}\mathbf{L}^{-\top}\mathbf{D}^{-\frac{1}{2}}$,

*(ii)* $\lambda(\mathbf{A}, \mathbf{B}) = \lambda(\mathbf{D}^{-\frac{1}{2}}\mathbf{L}^{-1}\mathbf{A}\mathbf{L}^{-\top}\mathbf{D}^{-\frac{1}{2}})$, *and,*

*(iii)* $\lambda^{\downarrow}(\mathbf{A}, \mathbf{B}) = \lambda^{\uparrow}(\mathbf{L}^{-1}\mathbf{D}^{-\frac{1}{2}}\mathbf{A}\mathbf{L}^{-\top}\mathbf{D}^{-\frac{1}{2}})$.

*Proof.* Write, $\mathbf{C} := \mathbf{L}^{\top}\mathbf{D}^{\frac{1}{2}}$ then $\mathbf{B} = \mathbf{C}\mathbf{C}^{\top}$ $(\lambda, \mathbf{v})$ is an eigenpair of $(\mathbf{A}, \mathbf{B})$ if and only, $\mathbf{A}\mathbf{V} = \lambda\mathbf{B}\mathbf{v} \iff \mathbf{C}^{-1}\mathbf{A}\mathbf{v} = \lambda\mathbf{C}^{\top}\mathbf{v}$ *(iii)*. Writing $\mathbf{w} := \mathbf{C}^{\top}\mathbf{v}$, $(\lambda, \mathbf{v})$ is an eigenpair of $(\mathbf{A}, \mathbf{B})$ if and only if $\mathbf{C}^{-1}\mathbf{A}\mathbf{C}^{-\top}\mathbf{w} = \lambda\mathbf{w}$, which holds if and only if $(\lambda, \mathbf{w})$ is an eigenpair of $\mathbf{C}^{-1}\mathbf{A}\mathbf{C}^{-\top}$, this proves *(i)*, *(ii)*, and *(iii)*. $\quad\square$

**Proposition 26** (Spectral equivalance ∶). *Let* $\mathbf{\Phi}_1, \mathbf{\Phi}_2 \in \mathcal{Q}$. *Then,*

*(i)* $\lambda(\mathbf{\Phi}_1 : \mathbf{\Phi}_2) = \lambda(\mathbf{\Phi}_1, \mathbf{\Phi}_2)$, *and,*

*(ii)* $\lambda^{\downarrow}(\mathbf{\Phi}_1 : \mathbf{\Phi}_2) = \lambda^{\downarrow}(\mathbf{\Phi}_1, \mathbf{\Phi}_2)$.

*Proof.* *(i)* and *(ii)* follow by the definition of ∶ and proposition 25. $\quad\square$

**Proposition 27.** *Let* $\mathbf{\Phi}_1, \mathbf{\Phi}_2 \in \mathcal{Q}$. *Then,*

*(i)* $\lambda(\mathbf{\Phi}_2 : \mathbf{\Phi}_1) = \lambda(\mathbf{\Phi}_1 : \mathbf{\Phi}_2)^{-1}$, *and,*

*(ii)* $\lambda^{\downarrow}(\mathbf{\Phi}_2 : \mathbf{\Phi}_1) = \lambda^{\uparrow}(\mathbf{\Phi}_1 : \mathbf{\Phi}_2)^{-1}$.

*Proof.* *(i)* By proposition 26 $\lambda(\mathbf{\Phi}_2 : \mathbf{\Phi}_2) = \lambda(\mathbf{\Phi}_2, \mathbf{\Phi}_1)$. By proposition 24 $\lambda(\mathbf{\Phi}_2, \mathbf{\Phi}_1) = \lambda(\mathbf{\Phi}_1, \mathbf{\Phi}_2)^{-1}$, this proves *(ii)*. The map $t \mapsto \frac{1}{t}$ being order reversing, $\lambda^{\downarrow}(\mathbf{\Phi}_2 : \mathbf{\Phi}_1 = \lambda^{\uparrow}(\mathbf{\Phi}_1 : \mathbf{\Phi}_2)^{-1}$. This proves *(iii)*. $\quad\square$

## E.5 CONCAVITY OF THE LOG-DETERMINANT

Several proofs are possible. We give one leveraging Gaussian integrals and Holder's inequality.

**Proposition 28.** *The function,* $\mathbf{S} \mapsto \log\det[\mathbf{S}]$ *is, non-decreaing and concave on the cone of symmetric positive definite matrices.*

*Proof.* For any $\mathbf{C} \in \mathcal{S}_{\succ}(d)$, we have, $\int e^{-\frac{1}{2}\langle\mathbf{x},\mathbf{C}\mathbf{x}\rangle}\, d\mathbf{x} = \frac{\pi^{\frac{d}{2}}}{\sqrt{\det[\mathbf{C}]}}$. Take $\mathbf{A}, \mathbf{B} \in \mathcal{S}_{\succ}(d)$, and, $p, q > 0$, such that $\{t\}$, $\frac{1}{p} + \frac{1}{q} = 1$. We have,

$$\frac{\pi^{\frac{d}{2}}}{\sqrt{\det\left[\frac{1}{p}\mathbf{A} + \frac{1}{q}\mathbf{B}\right]}} = \int e^{-\left\langle\mathbf{x},(\frac{\mathbf{A}}{p}+\frac{\mathbf{B}}{q})\mathbf{x}\right\rangle}\, d\mathbf{x}.$$

By Holder's inequality,

$$\int e^{-\left\langle\mathbf{x},\frac{\mathbf{A}}{p}\mathbf{x}\right\rangle}e^{-\left\langle\mathbf{x},\frac{\mathbf{B}}{p}\mathbf{x}\right\rangle}\, d\mathbf{x} \leq \left(\int e^{-\langle\mathbf{x},\mathbf{A}\mathbf{x}\rangle}\, d\mathbf{x}\right)^{\frac{1}{p}}\left(\int e^{-\langle\mathbf{x},\mathbf{B}\mathbf{x}\rangle}\, d\mathbf{x}\right)^{\frac{1}{q}},$$

$$= \left(\frac{\pi^{\frac{d}{2}}}{\sqrt{\det[\mathbf{A}]}}\right)^{\frac{1}{p}} + \left(\frac{\pi^{\frac{d}{2}}}{\sqrt{\det[\mathbf{B}]}}\right)^{\frac{1}{q}}.$$

Taking the logarithm and simplifying, we conclude,

$$\log\det\left[\frac{1}{p}\mathbf{A} + \frac{1}{q}\mathbf{B}\right] \geq \frac{1}{p}\log\det[\mathbf{A}] + \frac{1}{q}\log\det[\mathbf{B}].$$

$\quad\square$

### E.6    Necessary and sufficient solutions for a Pseudo-Inverse solution

**Lemma 7** (Penrose 1955). *A necessary and sufficient condition for the equation $\alpha\omega\beta = \gamma$ to have solution is*

$$\alpha\alpha^{\dagger}\gamma\beta\beta^{\dagger} = \gamma,$$

*in which case the general solution is,*

$$\omega = \alpha^{\dagger}\gamma\beta^{\dagger} + \tau - \alpha^{\dagger}\alpha\tau\beta\beta^{\dagger},$$

### E.7    f-divergence

**Definition 9.** Let $\mathbb{P}$ and $\mathbb{Q}$ be two probability measures on a measurable space $(\Omega, \mathscr{F})$, such that $\mathbb{P} \ll \mathbb{Q}$. For any convex function $f : (0, \infty) \mapsto \mathbb{R}$, strictly convex at 1 and satisfying $f(1) = 0$, the $f$-divergence between $\mathbb{P}$ and $\mathbb{Q}$ is defined as,

$$D_f(\mathbb{P} \,||\, \mathbb{Q}) = \mathbb{E}_{\mathbb{Q}}\left[ f\left( \frac{\mathrm{d}\mathbb{P}}{\mathrm{d}\mathbb{Q}} \right) \right].$$

**Theorem 5** (Elementary properties of f-divergences). *Let $\mathbb{P}$ and $\mathbb{Q}$ be two probability measures on the measurable space $(\mathscr{X}, \mathscr{B}(\mathscr{X}))$.*

(i) *$D_f(\mathbb{P} \,||\, \mathbb{Q}) \geq 0$ with equality if and only if $\mathbb{P} = \mathbb{Q}$.*
(ii) *The function $(\mathbb{P}, \mathbb{Q}) \mapsto D_f(\mathbb{P} \,||\, \mathbb{Q})$ is jointly convex.*
(iii) *Let $(\mathscr{X}, \mathscr{B}(\mathscr{X}))$ and $(\mathscr{Y}, \mathscr{B}(\mathscr{Y}))$ be two measurable spaces. Let $\mathbb{P}_X$ be a probability measure on $(\mathscr{X}, \mathscr{B}(\mathscr{X}))$. Let $\mathbb{P}_{Y|X} : (\mathscr{X}, \mathscr{B}(X)) \mapsto$ and $\mathbb{Q}_{Y|X}$ be two conditional probability operators (Markov kernels). Define, $\mathbb{P}_Y = \mathbb{E}_{\mathbb{P}_X}[\mathbb{P}_{Y|X}]$ and $\mathbb{Q}_Y = \mathbb{E}_{\mathbb{P}_X}[\mathbb{Q}_{Y|X}]$. Then,*

$$D_f\left( \mathbb{P}_Y \,||\, \mathbb{Q}_Y \right) \leq D_f\left( \mathbb{P}_{Y|X} \,||\, \mathbb{Q}_{Y|X} \right).$$

*Proof.* (Polyanskiy & Wu, 2016) $\qquad\qquad\qquad\qquad\qquad\qquad\qquad\qquad\qquad\qquad\qquad\qquad\qquad\qquad$ □

