# OpenReview forum: "Linearly Independent Feature Extraction"
_ICLR.cc/2026/Conference — Submitted to ICLR 2026_

### Official Review · Reviewer_1aVu · 2025-10-26

**Soundness:** 3
**Presentation:** 2
**Contribution:** 2
**Rating:** 2
**Confidence:** 3

**Summary:**

The paper introduces Linearly Independent Feature Extraction (LIFE), a spectral framework for out-of-distribution generalization that models a continuous spectrum of state dependence instead of seeking strict domain invariance. By formulating state dependence through an operator algebra equivalent to information-theoretic measures and solving a simple eigenvalue problem, LIFE learns representations with controllable invariance. Empirical results show that LIFE outperforms or matches existing OOD methods such as IRM and GroupDRO, achieving near-oracle performance on some benchmarks, however, the experimental comparison is very limited.

**Strengths:**

(1) LIFE offers a theoretically grounded ynd computationally simple framework that unifies information theory and spectral analysis to learn state-dependent representations through a single eigensolver, making it both interpretable and efficient.

(2) The evaluation shows superior or near oracle performance on some OOD datasets while mitigating shortcut learning, however, those experiments are limited.

**Weaknesses:**

(1) The paper is theoretically dense, and while its algebraic formalism and operator-theoretic exposition are elegant, the underlying conceptual intuition is not clearly conveyed. The connections among the theoretical components in sec. 2 are difficult to follow, and the practical implementation details of the proposed method remain insufficiently explained.

(2) The evaluations are primarily conducted on linear models and some of the standard OOD benchmarks, which effectively demonstrate the core theoretical claims but limit insights into the method's broader applicability. It also lacks comprehensive comparisons with recent deep invariant or causal OOD frameworks. A detailed list of benchmarks could be found in [A]. Also, scalability to deep nonlinear architectures remains underexplored.

(3) The paper lacks ablation studies and sensitivity analyses. For example, the influence of hyperparameters (e.g., rank-k subspace, priors) and robustness to noise or imperfect priors could have been systematically studied.

**Questions:**

Please see the weaknesses

---

### Official Review · Reviewer_WGnP · 2025-10-28

**Soundness:** 3
**Presentation:** 2
**Contribution:** 3
**Rating:** 4
**Confidence:** 2

**Summary:**

Domain invariance is too crude for OOD; This paper models how features depend on the world's state across a whole spectrum.
They build a spectral theory—equivalent to information-theoretic dependence—inside a new operator algebra to make this goal concrete.
The result is LIFE, an eigensolver-based algorithm that extracts linearly independent representations whose state-dependence can be dialed at will.  On analytic benchmarks LIFE recovers oracle features; on linear tasks it beats invariant baselines and sometimes even deep non-linear models.  Together, the work may furnishe a general, dynamic theory of controllable state-dependence for robust generalization.

**Strengths:**

1. The theory seems quite comprehensive.

**Weaknesses:**

1. No title

2. Lack of strict definition of problems, many theories do not know what problems to solve.

3. There are very few experiments, and there is a lack of theoretical analysis and experimental evidence on the domain generalization aspect of this method.

4. Why can this method be applied to the problem of out of distribution generalization, and why can the effect be improved.

**Questions:**

1. Why can this method be applied to the problem of out of distribution generalization, and why can the effect be improved.

2. The theory is complex, but the problem analysis is not thorough. I saw something about Robust PCA.

3. There are too few experiments, and I am concerned about practicality and the significance of the problem.

4. How is out of distribution reflected in your system?  The feature extraction process is complex and difficult to understand.

---

### Official Review · Reviewer_K6fh · 2025-10-30

**Soundness:** 3
**Presentation:** 3
**Contribution:** 2
**Rating:** 4
**Confidence:** 3

**Summary:**

This paper proposes a novel algebraic framework to study the dependence of learned features on the "state" of the world, a concept referred to as the "environment" in related works on invariant learning. The authors develop a spectral theory, grounded in an operator algebra, to quantify this dependence and introduce an algorithm, Linearly Independent Feature Extraction (LIFE), to learn representations with controllable state-dependence.

**Strengths:**

1. The core idea and the accompanying mathematical formalism are interesting and are, overall, presented neatly and clearly.

2. The mathematical contribution, specifically the development of a novel operator algebra to formally connect a spectral theory with information-theoretic measures of dependence is noteworthy, even if its immediate relevance for complex, real-world machine learning applications is still arguable.

Hence I think the paper has upsides in terms of originality, quality and clarity.

**Weaknesses:**

1. A central claim of the paper is that "perfect invariance" is not a desirable target for OOD generalization. However, the experiments are then conducted on tasks where perfect invariance is required, and the paper shows that the proposed method successfully achieves this invariance. This makes the criticism of perfect invariance difficult to fully understand.
    * The paper would be far more convincing if it demonstrated its method's effectiveness on a real-world dataset, such as one from the WILDS benchmark, where perfect invariance is not built into the data-generating process.
   * Alternatively, given the theoretical nature of the paper, a clear toy linear setting describing an intuitive problem where LIFE is guaranteed to generalize while standard invariant learning methods (like IRM) or ERM do not would be a very helpful addition.
2. A minor correction: The paper claims that IRM (Invariant Risk Minimization) looks for "causal" features. This is inaccurate, since for example works like [1] discuss problems involving both causal and anti-causal but stable features, which IRM (or similarly multi-domain calibration) can learn. The claim about causal features seems more directly related to earlier work like ICP [2]. While I agree with the general point that perfect invariance is a strong and potentially rigid requirement, it's not clear that the proposed solution is a superior alternative, especially since optimizing a loss function (like in variants of IRM) is already a form of relaxation.
3. The experimental evaluation has several points that need clarification and could be strengthened.
    * The experiments do not seem to ablate the problem of training a deep network for invariance vs. the problem of the objective itself not being desirable for OOD generalization.
    * The proposed LIFE method operates on the features of the last layer of a network (a two-stage approach), while other baseline methods appear to train networks from scratch. It has been argued from both theoretical (e.g. [3]) and empirical (e.g., DFR [4]; DARE [5]") standpoints that two-stage training is highly beneficial when training large models with losses that promote robustness to spurious correlations.
    * These two-stage baselines were not incorporated into the evaluation. Therefore, it is unclear whether the proposed LIFE algorithm is a fundamentally better objective for these cases, or if its success stems from it being a two-stage method that, like DFR, circumvents the geometric or optimization-related problems (like malign overfitting) associated with end-to-end training of large models with invariance-promoting losses.
    * It is not specified how the rank-$k$ parameter for the LIFE method was chosen for the experiments.

Finally, the main title of the paper is missing from the header of the page. I disregarded that while performing the review, but this should be amended.

[1] Wald, Yoav, et al. "On calibration and out-of-domain generalization." Advances in neural information processing systems 34 (2021): 2215-2227.

[2] Peters, Jonas, Peter Bühlmann, and Nicolai Meinshausen. "Causal inference by using invariant prediction: identification and confidence intervals." Journal of the Royal Statistical Society Series B: Statistical Methodology 78.5 (2016): 947-1012.

[3] Wald, Yoav, et al. "Malign Overfitting: Interpolation can Provably Preclude Invariance." The Eleventh International Conference on Learning Representations.

[4] Kirichenko, Polina, Pavel Izmailov, and Andrew Gordon Wilson. "Last Layer Re-Training is Sufficient for Robustness to Spurious Correlations." The Eleventh International Conference on Learning Representations.

[5] Rosenfeld, Elan, Pradeep Ravikumar, and Andrej Risteski. "Domain-adjusted regression or: Erm may already learn features sufficient for out-of-distribution generalization." arXiv preprint arXiv:2202.06856 (2022).

**Questions:**

1. Could the authors clarify the apparent contradiction between criticizing "perfect invariance" as undesirable, yet evaluating the method on tasks where perfect invariance is the goal and showing that LIFE achieves it?
2. Given that LIFE is implemented as a two-stage method, how does its performance compare to other two-stage baselines designed to mitigate malign overfitting, such as DFR or Domain-Adjusted ERM? This would help isolate whether the benefit comes from the novel spectral objective or from the two-stage training procedure itself.
3. How was the rank-$k$ parameter for LIFE selected in the experiments? Was it tuned per dataset, and if so, how?

---

### Official Review · Reviewer_et3t · 2025-10-31

**Soundness:** 2
**Presentation:** 3
**Contribution:** 2
**Rating:** 2
**Confidence:** 4

**Summary:**

The authors observe that the idea of invariant risk minimization is too restrictive for OOD generalization. To this end, they argue that generalization across domains should be achieved by performing representation learning in a more nuanced way through modelling the dependence of a learned feature on the state of the world it is operating in. With this view, they propose, Linearly Independent Feature Extraction (LIFE) based on spectral theory and operator algebra, and study the same both analytically and empirically.

**Strengths:**

1. The motivation of this work is sound and resonates with currently prevalent sentiments about invariant features in the domain generalization community that hard invariance is too strict a constraint for generalization and in practical scenarios, something more soft / adaptable to the target domain is needed.

2. The authors are rigorous in their theoretical analysis and provide a detailed account for the development of their spectral theory of state dependence, which naturally leads to their proposed algorithm.

3. The analytical treatment of their method presented in Section 5.2 is quite useful and adds completeness to the theoretical foundations, as the authors clearly illustrate that their method is able to recover invariant / oracle features corresponding to the state-independent subspace.

4. The proposed method is simple to implement and the algorithm is stated clearly.

**Weaknesses:**

1. The authors argue that they aim to learn representations whose dependence on the state is stable across domain shifts. However, what exactly they mean by this stability of dependence is not exactly clear.

2. The proposed algorithm depends on the assumption that the state of the environment is a recoverable object (Section 2.4). However, this assumption may not necessarily hold under well-known fundamental limitations of identifiability [a]. Furthermore, if this is the case, then the proposed approach is not different, in essence to invariance learning algorithms, which are governed by the same limitations.

3. In Table 2, although the proposed method, LIFE, achieves better worst group accuracy compared with the baselines, the average accuracy is lower than all other baselines. However, no comment or discussion on why this happens is provided in the text. Is there a trade-off that life strikes between average and worst group accuracies? In that case, the behaviour of LIFE can be thought of as optimizing for the worst group while sacrificing average performance, which is not an ideal characteristic of a domain generalization algorithm.

4. Although the authors theoretically argue the limits of capturing invariant features and present LIFE as an alternative, their caption in Table 1 states otherwise. It is stated their that "LIFE improves worst-group accuracy on the
Waterbirds dataset by effectively isolating bird features from spurious background signals", which is essentially equivalent to capturing invariant features, the invariant being the bird.

5. The evaluation for the robustness to spurious correlations is quite insufficient. Several methods such as LFF [b], JTT [c], Logit Correction [d], DeNetDM [e], etc., provide performance significantly superior to the considered GroupDRO, none of which have been compared against.

6. Methods like GroupDRO requires group information during training, however, their have been several future developments such as LFF, JTT, etc, which do not, while providing much superior performance. Was the evaluation of the proposed LIFE also conducted with group information being used during training?

7. Overall, the authors' statement, "to ask only if a feature is invariant is to mistake a practical problem of degree for a philosophical problem of kind", nicely summarizes the spirit of this work. It illustrates its foundational view that simply seeing features as invariant and spurious is too simplistic and an algorithm should allow for a more soft/probabilistic treatment of features based on the domain, which the authors call dependence of the representations on the state of the world. However, this idea of soft invariance is being explored in the community since several years [e, f], (Zhou et al., 2022), so claiming this work to be wholly complementary to the IRM literature is an overstatement.

Formatting:

The title is not present in the paper.
Line 430: "Annhilation" -> "Annihilation"
Line 211: "as trace the trace" -> "as the trace"

References:

[a] Gulrajani and Hashimoto, "Identifiability Conditions for Domain Adaptation", ICML 2022.\
[a] Nam et al., "Learning from Failure: Training Debiased Classifier from Biased Classifier", NeurIPS 2020.\
[b] Liu et al., "Just Train Twice: Improving Group Robustness without Training Group Information", ICML 2021.\
[c] Liu et al., "Avoiding spurious correlations via logit correction", ICLR 2023.\
[d] Sreelatha et al., "DeNetDM: Debiasing by Network Depth Modulation", NeurIPS 2024.\
[e] Zhang et al., "Free Lunch for Domain Adversarial Training: Environment Label Smoothing", ICLR 2023.\
[f] Huh et al., "The Missing Invariance Principle Found – the Reciprocal Twin of Invariant Risk Minimization", NeurIPS 2022.

**Questions:**

Please refer to the Weaknesses section.

---

### Author Response · Authors · 2025-12-04
**General response**

We thank the reviewers for their time, efforts and insightful comments.

The "quality, originality and clarity" and "rigour" of the work, the "soundness" behind its motivation, and the mathematical contributions have been kindly noted.

LIFE does not necessarily present itself as complementary to invariance. Rather, the paper shows both analytically and empirically that domain invariance corresponds to a single point of the spectrum LIFE defines.

Our aim is to show that question out-of-domain generalization can be naturally expressed in terms of dependence on the state.

General changes to the draft

- Improved formatting and presentations.
- Improved writing by adding motivations to important section.
- Improved the readability of the appendix by adding a table of contents.
- Added a didact analytical example revealing the inner working of the algorithm while showing how it can be applied in situation of absence of invariance.

---

### Meta-Review · Area_Chair_39Bu · 2025-12-30

**Summary:**

The central claim of the submission is that enforcing invariance of features across domains is an overly strict objective for domain generalisation. This is analysed primarily from a theoretical viewpoint and supplemented with experimental investigations. The reviewers appreciated the theoretical depth of the paper and the simplicity of the proposed approach. However, several concerns were raised by multiple reviewers. The most prevalent concern was the limited scope of the empirical validation of the proposed approach, including missing baselines, ablations, and flaws in the experimental setup. There were also concerns raised that indirectly relate to the relevance of the submission to deep networks: reviewer 1aVu points out the evaluation is limited to linear settings and reviewer et3t raises the issue of identifiability, which is exacerbated in deep learning contexts.

**Reviewer Concerns:**

The rebuttal was very brief, so it is unlikely that any of the major concerns were fully addressed. Some of the minor presentations issues have been fixed, and a concern related to the experimental setup relying too much on invariances is likely partially addressed.

**Reviewer Scores:**

I believe it is unlikely that reviewers K6fh, WGnP, and 1aVu would have changed their score. There is a small chance that reviewer et3t would have raised their score slightly due to the additional example focused on non-invariance preserving domain shifts.

---

### Decision · Program_Chairs · 2026-01-26

Reject